# Parameterization of Wind Evolution using Lidar

Yiyin Chen[1], David Schlipf[2], and Po Wen Cheng[1]

[1]Stuttgart Wind Energy (SWE) at Institute of Aircraft Design, University of Stuttgart, Allmandring 5b, 70569 Stuttgart, Germany
[2]Wind Energy Technology Institute, Flensburg University of Applied Sciences, Kanzleistraße 91-93, 24943 Flensburg, Germany

**Correspondence:** Yiyin Chen (chen@ifb.uni-stuttgart.de)

**Abstract.** Wind evolution, i.e. the evolution of turbulence structures over time, has become an increasingly interesting topic in recent years, mainly due to the development of lidar-assisted wind turbine control, which requires accurate prediction of wind evolution to avoid unnecessary or even harmful control actions. Moreover, four-dimensional stochastic wind field simulations can be made possible by integrating wind evolution into standard three-dimensional simulations, to provide a more realistic simulation environment for this control concept. Motivated by these factors, this research aims to investigate the potential of Gaussian process regression in the parameterization of wind evolution. Wind evolution is commonly quantified using magnitude-squared coherence of wind speed and is estimated with lidar data measured by two nacelle-mounted lidars in this research. A two-parameter wind evolution model modified from a previous study is used to model the estimated coherence. A statistical analysis is done for the wind evolution model parameters determined from the estimated coherence, to provide some insights into the characteristics of wind evolution. Gaussian process regression models are trained with the wind evolution model parameters and different combinations of wind-field–related variables acquired from the lidars and a meteorological mast. The automatic relevance determination squared exponential kernel function is applied to select suitable variables for the models. The performance of the Gaussian process regression models is analyzed with respect to different variable combinations, and the selected variables are discussed to shed light on the correlation between wind evolution and these variables.

## 1   Introduction

*Wind evolution* refers to the physical phenomenon of turbulence structures (eddies) changing over time, and is defined, in this study, as magnitude-squared coherence dependent on evolution time. Magnitude-squared coherence (hereafter referred to as coherence) is a common statistical measure of turbulence structure properties (see e.g. Panofsky and McCormick, 1954; Davenport, 1961; Panofsky et al., 1974). In general, coherence describes the correlation between spectral components of two signals or data sets, taking values between zero, for no correlation, to unity, for perfect correlation. Because turbulent eddies are advected by the mean flow while evolving, the longitudinal coherence, i.e. coherence of turbulent velocity at locations separated in the mean direction of the flow, is used to measure wind evolution in practice (see e.g. Schlipf et al., 2015; Simley and Pao, 2015). And when estimating the coherence, the data measured at the downstream location should be shifted by the travel time, corresponding to the evolution time, to match the data measured at the upstream location. Taylor's (1938) hypothesis is a

special case that assumes all turbulent motions remain unchanged, while eddies move with the mean flow. In other words, it assumes no wind evolution, which means the coherence is unity for all frequencies. The validity of Taylor's (1938) hypothesis was researched in some studies (see e.g. Willis and Deardorff, 1976; Schlipf et al., 2011) and this hypothesis is widely used in data analysis and wind field modeling for the sake of simplification (see e.g. Kelberlau and Mann, 2019; Veers, 1988).

The research on wind evolution dates back to the 1970s. Pielke and Panofsky (1970) attempted to generalize some of the mathematical descriptions for horizontal variation of turbulence characteristics. The final goal at that time was to figure out an empirical model of the four-dimensional (space–time) structure of turbulence. In Pielke and Panofsky's (1970) work, the coherence model suggested by Davenport (1961) to describe the correlation between horizontal wind components at different heights, also known as Davenport Geometric Similarity, was extended into other wind components and separation directions. Pielke and Panofsky's (1970) model also followed Davenport's idea to approximate the coherence with a simple exponential function using a single decay parameter. The decay parameters were assumed to be constants. After that, Ropelewski et al. (1973) systematically studied the coherence for streamwise and cross-stream wind components with horizontal separations. Based on their theoretical discussion, the decay parameter for longitudinal separation is supposed to be a function of turbulence intensity, which is a function of roughness length and Richardson number (a measure of atmospheric stability) (Lumley and Panofsky, 1964). Extending the study, Panofsky and Mizuno (1975) found that the relationships between coherence and other parameters were rather complicated. A model for the decay parameter was proposed based on its empirical properties. This decay parameter model involves turbulence intensity accounting for the influence of terrain roughness, standard deviation of the lateral wind component, lateral integral length scale of the longitudinal wind component (which shows a relationship with Richardson number), separation of two observations, and the angle between the wind direction and the measurement line. This model can be regarded as the first parameterization of Pielke and Panofsky's (1970) model. However, the model was developed using only very few observations taken on meteorological towers, and the dependence of coherence on separation and atmospheric stability was not thoroughly researched in that study.

It is worth mentioning that the longitudinal coherence differs from the lateral and vertical coherence because the former is coupled with time-dependent variations in turbulence, while the latter measures the decay of correlation due to spatial separations in their respective directions. However, in the above-mentioned studies the longitudinal coherence was not clearly distinguished. Kristensen (1979) proposed that the longitudinal coherence should behave differently and deduced an alternative expression for it, which we refer to Kristensen's (1979) model. This model assumes that the coherence can be modeled with the probability that an eddy observed at the first point can also be observed at the second point, given that: the eddy has not completely faded out during the travel time; and the eddy has been taken towards the second point.

Wind evolution has become interesting again because of the new concept of lidar-assisted wind turbine control (see e.g. Schlipf, 2015; Simley, 2015; Simley et al., 2018). Lidar — more specifically, Doppler wind lidar — is a remote sensing technology which can be used to measure wind speed in a certain spatial range (Weitkamp, 2005). The main idea of lidar-assisted wind turbine control is to enable a feedforward control of wind turbines by using a nacelle-mounted lidar to measure the approaching wind field at some distance upwind. The control system should react only to the changes in the wind field which can be predicted accurately, to avoid harmful and unnecessary control actions. This is made possible by applying an

adaptive filter to remove the uncorrelated part of the lidar signal. An accurate prediction of the wind evolution will thus benefit the filter design. Moreover, the application of Taylor's hypothesis in the wind field simulation is no longer appropriate for modeling the lidar-assisted control system. To solve this problem, different approaches (see e.g. Bossanyi, 2013; Laks et al., 2013) have been proposed to integrate the wind evolution model within the wind field simulation method of Veers (1988), to make it possible to simulate a four-dimensional wind field.

Some attempts were made to further promote the modeling of wind evolution. Schlipf et al. (2015) suggested an approach to determine the decay parameter in Pielke and Panofsky's (1970) model with data measured by a nacelle-mounted lidar, taking into account the influence of lidar measurement on coherence. However, the limitation of this study is that only four one-hour data blocks were examined. Simley and Pao (2015) attempted to validate the models of Pielke and Panofsky (1970) and Kristensen (1979) with data from LES wind fields, but found that neither model can always correctly model the coherence as frequency approaches zero. To improve this issue, Simley and Pao (2015) tried to apply the coherence model for transverse and vertical separations suggested by Thresher et al. (1981) to the longitudinal coherence. This model has a form similar to Pielke and Panofsky's (1970) model but includes an additional parameter to allow coherence less than unity at very low frequency. Davoust and von Terzi (2016) examined Simley and Pao's (2015) model with data from nacelle-mounted lidars on three sites. To enable a direct comparison with Simley and Pao's (2015) work, a correction method was applied to compensate the influence of lidar measurement on coherence. However, the linear dependence of the decay parameter on turbulence intensity suggested by Simley and Pao (2015) was not clearly observed. The relationship between the offset parameter and integral length scale shows a good match with that suggested in Simley and Pao's (2015) work, but the agreement decreases after the correction of coherence. At the same time, de Maré and Mann (2016) developed a four-dimensional model to describe the space-time structure of turbulence by combining the Mann (1994) spectral velocity tensor and Kristensen's (1979) longitudinal coherence model.

Motivated by the above-mentioned research, this study aims to achieve parameterization models for a wind evolution model modified from Simley and Pao's (2015) model. In addition, it is desired to gain some insights into the complex relationships between wind evolution and wind-field–related variables such as wind statistics, atmospheric stability, and relative positions of measurement points. For these purposes, a previous study (Chen, 2019) was done to explore different supervised machine learning algorithms on a simple level, including stepwise linear regression (see e.g. Hocking, 1976), regression tree (see e.g. Breiman et al., 1984), support vector regression (see e.g. Vapnik, 1995), and Gaussian process regression (see e.g. Rasmussen and Williams, 2006). It was found that Gaussian process regression, overall, performs the best for prediction of wind evolution model parameters, and thus its potential is further analyzed in this study with more extensive data.

This research is mainly done using lidar measurement because lidar can provide large amounts of spatially separated measuring points simultaneously, which is of great advantage for studying the dependence of wind evolution on separation in comparison to data from a meteorological tower. Lidar data from two measurement campaigns undertaken in different terrain types are available. In one of the measurement campaigns, data taken on a meteorological tower is also involved in the analysis to provide a comparison.

The present paper is organized as follows: Section 2 briefly explains the theoretical basis of wind evolution and its prediction concept as well as the principles of the methods applied in this work; Section 3 introduces the measurement campaigns and the data processing; Section 4 presents the results of the statistical analysis of the wind evolution model parameters; Section 5 illustrates the process of model training and the evaluation of the parameterization models; Section 6 summarizes the results and gives the conclusions and an outlook.

## 2  Methodology

This section first explains the mathematical expression of wind evolution in Sect. 2.1. Then, our concept of wind evolution prediction and a corresponding workflow are presented in Sect. 2.2. After that, the wind evolution model applied in this work is introduced in Sect. 2.3. Finally, the details of the workflow are introduced and discussed in Sect. 2.4–2.7.

### 2.1  Wind Evolution

As mentioned in the introduction, wind evolution is mathematically defined as the magnitude-squared coherence between two wind speed signals $i$ and $j$ measured at two points separated in the longitudinal direction, with $i$ for the signal measured at the upstream point and $j$ at the downstream point:

$$\gamma_{ij}^2(f) = \frac{|S_{ij}(f)|^2}{S_{ii}(f)S_{jj}(f)}, \tag{1}$$

where $S_{ii}(f)$ and $S_{jj}(f)$ represent the power-spectral densities (PSDs) of signals $i$ and $j$, respectively, and $S_{ij}(f)$ represents the cross-spectral density between $i$ and $j$. It must be emphasized that the coherence corresponds to a lagged correlation, which means the signal $j$ should be shifted by the travel time $\Delta t$ after which the signal $i$ is expected to arrive at the downstream point for calculation of the coherence.

### 2.2  Concept and Workflow

A supervised learning algorithm aims to find the mapping function from *predictors* (i.e. input variables) to a *target* (i.e output variable) through known data about the predictors and the target without relying on a predefined equation as a model. The key to using supervised learning is to identify suitable predictors and targets, which is in fact a process of abstracting and condensing information.

In this study, we aim to develop a predictive model for wind evolution of the longitudinal wind component. It is worth noting the different meanings of wind evolution and wind evolution model. Wind evolution, i.e. the coherence estimated from measured data in practice, is not predictable because the estimated coherence consists of approximately infinite data points. Therefore, a model with a limited number of parameters is needed to approximate the estimated coherence; this is a wind evolution model. From the perspective of machine learning, using a wind evolution model is essentially condensing the information in the estimated coherence into several model parameters which are predictable. These model parameters are targets of predictive models, and thus the predictive model is deemed a *parameterization model* in this study.

Wind-field–related variables such as wind statistics, atmospheric stability, and relative positions of measurement points are considered as *potential* predictors, based on the theoretical and experimental studies mentioned in the introduction. A discussion about the potential predictors is provided in Sect. 2.5. Further analysis needs to be done to determine which of the potential predictors should be selected for model training, i.e. *feature selection*. The principle of feature selection is to figure out which variables provide the best predictive power (accounting for most of the variation in the target values) and, ideally, these variables should be independent from each other to prevent over-fitting in model training. To investigate the necessary predictors under different data availability, different combinations of predictors are discussed in Sect. 5.

Figure 1 illustrates our concept and workflow of wind evolution prediction. For model training, the essential steps are determination of observed values of predictors and targets from measured data and training parameterization models using a machine learning algorithm, more specifically: 1) to estimate the coherence using lidar data; 2) to determine the observed target values, i.e. the wind evolution model parameters, by fitting the estimated coherence to a wind evolution model; 3) to calculate observed predictor values from measured data (mainly lidar data; sonic data could be used if available); 4) to train parameterization models using a machine learning algorithm. The prediction process goes in the opposite direction: Firstly, the wind evolution model parameters are predicted by the trained parameterization models using new predictor values calculated from new measured data; then, the predicted coherence is reconstructed by the wind evolution model using the predicted model parameters.

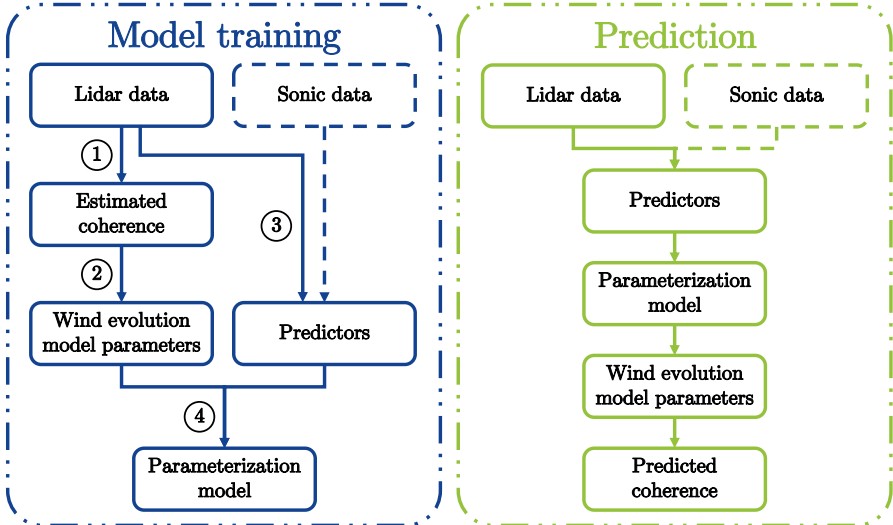

**Figure 1.** Concept and workflow of wind evolution prediction. The workflow of model training is: 1 – Estimation of coherence using lidar data; 2 – Determination of wind evolution model parameters by fitting the estimated coherence to a wind evolution model; 3 – Calculation of potential predictors from measured data (mainly lidar data; sonic data could be involved if available); 4 – Training parameterization models using a machine learning algorithm.

To demonstrate our concept and workflow: Sect.2.3 explains the wind evolution model used in this study; Sect. 2.4 discusses special issues about coherence estimation using lidar data; Sect. 2.5 discusses the potential predictors of the parameterization models; Sect. 2.6 and Sect. 2.7 briefly introduce the principle of Gaussian process regression (the machine learning algorithm applied in this study) and the method of model validation, respectively; Section 3.2 shows the fitting process of the estimated coherence in detail; Section 5 demonstrates the training of parameterization models, predictor selection, and model validation in the respective subsections.

## 2.3 Wind Evolution Model

Following the theoretical considerations by Ropelewski et al. (1973), the coherence decreases exponentially with increasing evolution time $\Delta t$ of the signal with respect to 'eddy turnover time' $\tau$

$$\gamma_{\text{model}}^2(f) = \exp\left(-C \cdot \frac{\Delta t}{\tau}\right). \tag{2}$$

The term $C$ represents the decay behaviour of the coherence depending on the time ratio. $C$ could be a constant, a linear function, or a more complicated term. $\tau$ is a time scale associated with the characteristic eddy size $\lambda$ and characteristic velocity of turbulence which is approximated by the standard deviation of wind speed $\sigma$ as follows

$$\tau \sim \frac{\lambda}{\sigma}. \tag{3}$$

This expression implies that eddies are supposed to decay faster under strong turbulence. Given the same degree of turbulence, large eddies are supposed to take longer time to decay. The eddy size $\lambda$ is linked to the frequency of horizontal wind velocity fluctuations $f$ and the flow mean wind speed $U$ with this relation

$$\lambda \sim \frac{U}{f}. \tag{4}$$

Combining Eq. (2)–(4), the coherence model becomes

$$\gamma_{\text{model}}^2(f) = \exp\left(-C \cdot \frac{\sigma}{U} \cdot f \cdot \Delta t\right). \tag{5}$$

This equation is essentially the same as the model proposed by Pielke and Panofsky (1970), except that, in their model, $\Delta t$ is approximated by $d/U$ ($d$ is separation) (Taylor, 1938; Willis and Deardorff, 1976), indicated as $\Delta t_{\text{T}}$.

Simley and Pao (2015) noted a limitation of this one-parameter model form: the intercept (coherence for 0 frequency) of the modeled coherence is forced to be unity, which is not always realistic. To overcome this issue, Simley and Pao (2015) introduced a second parameter in the coherence model, taking a model form similar to the coherence model for transverse and vertical separations suggested by Thresher et al. (1981)

$$\gamma_{\text{model}}^2(f, d) = \exp\left(-a'\sqrt{\left(\frac{fd}{U}\right)^2 + (b'd)^2}\right), \tag{6}$$

where $a'$ and $b'$ are tuning parameters. A comparison between the fitting quality of a one-parameter model and a two-parameter model is given in Sect. 3.2 to confirm the necessity of using a two-parameter wind evolution model.

We have made two modifications to Simley and Pao's (2015) model. Firstly, $d/U$ is restored to the travel time $\Delta t$ to avoid coupling the approximation of $\Delta t = d/U$ in the wind evolution model, considering the effect of the wind turbine's induction zone. In fitting the estimated coherence to the wind evolution model, $\Delta t$ is determined by the time lag of the peak of the cross-correlation between two wind speed signals, indicated as $\Delta t_{\mathrm{M}}$. Secondly, $a'b'd$ is replaced with $b$. The reasons for that are: 1) With the original form $a'b'd$, $a'b'$ is essentially the fitted term (given that $d$ is known) in the curve fitting. Thus, $b'$ shows a strong dependence on $a'$, which is generally undesirable for machine learning algorithms. And, 2) the form $a'b'd$ implies that this term is proportional to $d$, but we found that $d$ is still an important predictor for $b'$, indicating that the assumption of a linear relationship might be not proper. Therefore, we decided to directly use $b$ to represent the intercept and take $d$ as a predictor instead (see Sect. 2.5).

The modified wind evolution model is

$$\gamma^2_{\mathrm{model}}(f) = \exp\left(-\sqrt{a^2 \cdot (f \cdot \Delta t)^2 + b^2}\right), \tag{7}$$

where the *decay parameter* $a$ represents the decay effect of coherence, and the *offset parameter* $b$ is used to adjust the intercept (coherence for 0 frequency) of the modeled coherence curve. The intercept equals $\exp(-|b|)$. Both parameters are dimensionless. The term $f \cdot \Delta t$ is dimensionless, and thus is defined as *dimensionless frequency* $f_{\mathrm{dless}}$. In the end, our wind evolution model is defined as

$$\gamma^2_{\mathrm{model}}(f_{\mathrm{dless}}) = \exp\left(-\sqrt{a^2 \cdot f^2_{\mathrm{dless}} + b^2}\right). \tag{8}$$

In some studies (see e.g. Schlipf et al., 2015), the wind evolution model is defined as a function of wavenumber $k$, with $k = 2\pi f/U$. The relationship between $k$ and $f_{\mathrm{dless}}$ is $k = 2\pi f_{\mathrm{dless}}/d$, applying the approximation of $\Delta t = d/U$. To give an intuitive impression of the wind evolution model, Fig. 2 shows the theoretical curves calculated with different values of $a$ and $b$ as examples.

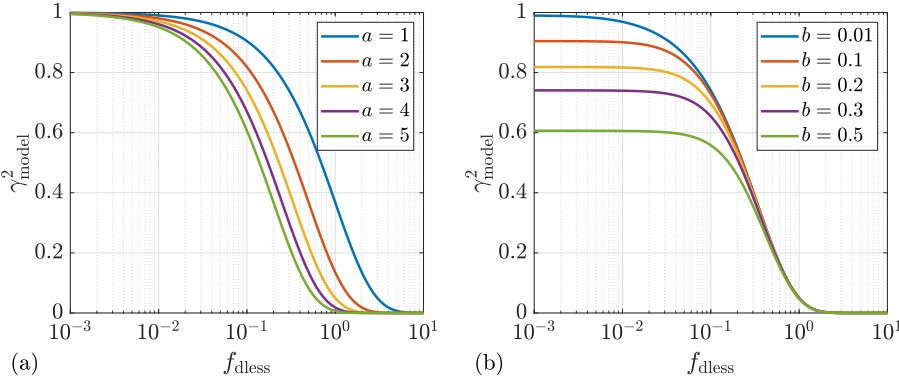

**Figure 2.** Impact of the model parameters $a$ and $b$ on the wind evolution model. (a) $b = 0$. (b) $a = 3$.

## 2.4 Estimating Coherence using Lidar Data

In this work, the coherence is estimated with lidar data because lidar can provide more data with respect to different spatial separations. This is not easy to obtain when using meteorological towers because multiple towers would be needed and only when the wind direction is aligned with the tower locations would the data be usable. Further, the prediction of the coherence is mainly expected to be applied when coupled with the deployment of a lidar, e.g. in lidar-assisted wind turbine control.

A Doppler wind lidar is a remote sensing device that measures wind speed based on the optical Doppler effect. Lidar emits laser pulses and detects the Doppler shift in backscattered light from aerosol particles in the atmosphere that are entrained with the wind. The Doppler shift is proportional to the line-of-sight wind speed, i.e. the wind speed projected onto the laser beam, and thus can be used to estimate the line-of-sight wind speed. The measurement principle of Doppler wind lidar is explained in many publications (e.g. Weitkamp, 2005; Peña et al., 2013; Liu et al., 2019) and thus is not introduced here in detail.

However, it must be emphasized that the coherence estimated with lidar data deviates from that estimated with data taken from ultrasonic anemometers. The reasons for that are: 1) The sampling rate of lidars is generally much lower than that of ultrasonic anemometers, and thus lidars cannot measure high-frequency fluctuations in wind speed; 2) the measuring volume of lidars is generally much longer than that of ultrasonic anemometers because of its measurement principle, and thus for lidars, the spatial averaging effect within the measuring volume needs to be considered; 3) lidars can only measure the wind speed projected onto the emitted laser beams, i.e. the light-of-sight wind speed. The influence of these three aspects is discussed following, specifically considering lidar in staring mode:

*Low sampling rate of lidar.* According to the Nyquist–Shannon sampling theorem (Shannon, 1949), the upper frequency limit of a signal transformed from the time domain into the frequency domain is the half of the sampling frequency. As long as the lidar sampling rate is sufficiently high to acquire a complete coherence curve covering the range from the highest coherence (e.g. 0.9 to 1.0) to the lowest coherence (e.g. 0 to 0.1), it would probably not have a large impact on studying the coherence. To obtain as high a sampling rate as possible, it is decided to select staring-mode data to calculate the coherence. Staring mode generally means that the lidar measures the wind speed with a single laser beam pointing in a fixed direction. Specifically in this work, the laser beam points horizontally upstream of the wind turbine.

*Spatial averaging effect of lidar.* Consider a pulsed lidar (only pulsed lidars are involved in this work). The spatial averaging effect can be modeled with a moving average weighted by a Gaussian-like shape function (see e.g. Carious, 2013) or a triangular function (see e.g. Sathe and Mann, 2012) centered at a measurement point. Following Carious (2013), the weighting function $w(x)$ is an even function centered at every measurement point along the laser beam. The lidar-measured wind speed at the measurement point $x_0$ for any instant can be modeled with

$$u_l(x_0) = \int_{-\infty}^{\infty} w(x_0 - x) u_p(x) dx = (w * u_p)(x_0), \tag{9}$$

where $u_p(x)$ is a wind speed function of spatial points on the $x$-axis aligned with the lidar's laser beam. According to the convolution theorem (Oppenheim et al., 1997), the following relationship is valid for the Fourier transformation between space

and wavenumber domain

$$\mathcal{F}\{u_l\} = \mathcal{F}\{w * u_p\} = \mathcal{F}\{w\} \cdot \mathcal{F}\{u_p\}, \tag{10}$$

where $\mathcal{F}\{\ \}$ is the Fourier transform operator.

Following Eq. (1), the coherence estimated with lidar data, indicated with the subscript 'l', is

$$225 \quad \gamma_{ij,l}^2(f) = \frac{|S_{ij,l}(f)|^2}{S_{ii,l}(f) \cdot S_{jj,l}(f)}, \tag{11}$$

where $S_{ii,l}(f)$ and $S_{jj,l}(f)$ are the auto-spectrum at the point $i$ and $j$, respectively, $S_{ij,l}(f)$ is the cross-spectrum between $i$ and $j$, and $f$ is the frequency in Hz. They are all estimated from lidar data. The auto-spectrum is

$$S_{ii,l}(f) = \mathcal{F}\{u_{i,l}(t)\} \cdot \mathcal{F}^*\{u_{i,l}(t)\}, \tag{12}$$

where $u_{i,l}(t)$ is the time series of the wind speed at $i$, and the symbol * means conjugate. And the cross-spectrum is

$$230 \quad S_{ij,l}(f) = \mathcal{F}\{u_{i,l}(t)\} \cdot \mathcal{F}^*\{u_{j,l}(t)\}. \tag{13}$$

Assume that the laser beam is aligned with the wind direction and Taylor's (1938) hypothesis applies within the measurement volume, and that Eq. (10) is also valid for the Fourier transformation between the time and frequency domains. Taylor's (1938) hypothesis is considered valid within the measurement volume because, in principle, wind evolution depends on the evolution time of turbulence (see Eq. (2)), and the measurement volume corresponds to a temporal length on the order of magnitude of
235 $10^{-7}$ s (typical length of a laser pulse). Now, Eq. (11) can be written as (with $t$ and $f$ omitted for clarity):

$$
\begin{aligned}
\gamma_{ij,l}^2 &= \frac{|\mathcal{F}\{u_{i,l}\} \cdot \mathcal{F}^*\{u_{j,l}\}|^2}{\mathcal{F}\{u_{i,l}\} \cdot \mathcal{F}^*\{u_{i,l}\} \cdot \mathcal{F}\{u_{j,l}\} \cdot \mathcal{F}^*\{u_{j,l}\}} \\
&= \frac{|\mathcal{F}\{w\} \cdot \mathcal{F}\{u_{i,p}\} \cdot \mathcal{F}^*\{w\} \cdot \mathcal{F}^*\{u_{j,p}\}|^2}{\mathcal{F}\{w\} \cdot \mathcal{F}\{u_{i,p}\} \cdot \mathcal{F}^*\{w\} \cdot \mathcal{F}^*\{u_{i,p}\} \cdot \mathcal{F}\{w\} \cdot \mathcal{F}\{u_{j,p}\} \cdot \mathcal{F}^*\{w\} \cdot \mathcal{F}^*\{u_{j,p}\}}.
\end{aligned}
\tag{14}
$$

Because the function $w(x)$ is real and even, according to the conjugate symmetry of the Fourier transformation (Oppenheim et al., 1997), $\mathcal{F}\{w\} = \mathcal{F}^*\{w\}$ and $\mathcal{F}\{w\}$ is real and even as well. As a result, all $\mathcal{F}\{w\}$ in the denominator and the numerator are cancelled out. And thus Eq. (14) becomes:

$$240 \quad \gamma_{ij,l}^2 = \frac{|\mathcal{F}\{u_{i,p}\} \cdot \mathcal{F}^*\{u_{j,p}\}|^2}{\mathcal{F}\{u_{i,p}\} \cdot \mathcal{F}^*\{u_{i,p}\} \cdot \mathcal{F}\{u_{j,p}\} \cdot \mathcal{F}^*\{u_{j,p}\}} = \gamma_{ij,p}^2. \tag{15}$$

This means that the spatial averaging effect does not influence the coherence under the above-mentioned ideal assumptions.

*Misalignment of wind direction and lidar measurement.* The above derivation is based on an important assumption that the laser beam is aligned with the wind direction. This will not always be fulfilled in reality, even for a nacelle-mounted lidar operating in staring mode. Figure 3 shows a misalignment between wind direction and lidar measurement direction, at an
245 angle $\alpha$. The coherence of the line-of-sight wind speed is $\gamma_{12}^2$, which is no longer the longitudinal coherence but the horizontal coherence as defined by Panofsky and Mizuno (1975). $\gamma_{13}^2$ and $\gamma_{23}^2$ are the longitudinal and lateral coherence, respectively.

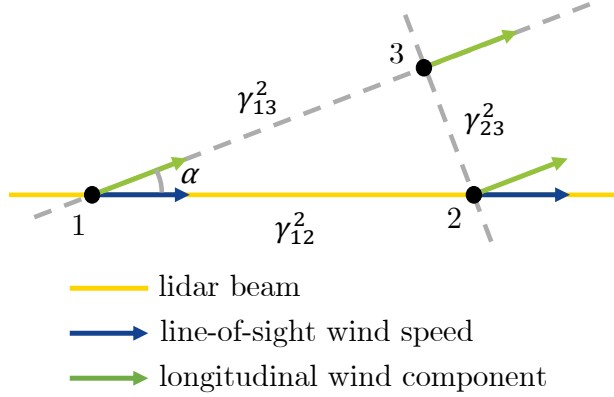

**Figure 3.** Misalignment of wind direction and lidar measurement. $\alpha$ is the misalignment angle. $\gamma_{12}^2$ is the coherence of the line-of-sight wind speed. $\gamma_{13}^2$ and $\gamma_{23}^2$ are the longitudinal and lateral coherence, respectively.

Schlipf et al. (2015) suggested a model for the horizontal coherence (magnitude coherence) based on the assumption of point-measurement for simplification

$$\gamma_{ij,\text{losP}} = \frac{\cos^2(\alpha)\gamma_{ij,\text{ux}}\gamma_{ij,\text{uy}}S_{ii,\text{u}}}{\cos^2(\alpha)S_{ii,\text{u}} + \sin^2(\alpha)S_{ii,\text{v}}}, \tag{16}$$

where $\gamma_{ij,\text{losP}}$ is the horizontal coherence of line-of-sight wind speed point-measurements, $\gamma_{ij,\text{ux}}$ and $\gamma_{ij,\text{uy}}$ are the longitudinal and lateral coherence of the longitudinal wind component, $S_{ii,\text{u}}$ and $S_{ii,\text{v}}$ are the auto-spectra of the longitudinal and lateral wind components. Based on this equation, determining the longitudinal coherence $\gamma_{ij,\text{ux}}$ is possible only given a specific turbulence model (knowing $S_{ii,\text{u}}$, $S_{ii,\text{v}}$ and $\gamma_{ij,\text{uy}}$) and knowing the misalignment angle $\alpha$. Moreover, the above discussed spatial averaging effect must be coupled to the horizontal coherence, considering that the lateral coherence for the point at $x$ depends

on the lateral separation $\Delta y$ associated with its distance from the center point of the range gate $x_0$, i.e. $\Delta y = \cos(\alpha)(|x - x_0|)$. Therefore, the longitudinal coherence is implicitly included in the integration of horizontal coherence weighted by the range weighting function of lidars.

In this study, we decide to develop a parameterization model based on horizontal coherence for the following reasons. Firstly, consider the case for a nacelle-mounted lidar. The misalignment of the lidar measurement means that the wind turbine

is misaligned as well. In this case, it makes sense to predict the corresponding horizontal coherence. Secondly, a standalone parameterization model, independent from any turbulence model, is desired for more flexibility in application. Thirdly, determining the parameters in an implicit wind evolution model is complicated when using measured data. And it is necessary to acquire the misalignment angle $\alpha$, which is not always possible in application, especially when lidar is the only data source, though deployment of lidars with multiple beams might help in this case . Moreover, the requirement for the accuracy of $\alpha$

is very high because $\alpha$ is included in the most basic step — fitting the estimated coherence to the wind evolution model. The uncertainties contained in $\alpha$ will propagate through the whole model and affect the further analysis radically. Since the prediction concept needs to be applicable under different data availabilities, it is not desired to make the fitting process depend so

critically on a variable whose availability and accuracy are not always guaranteed. It is thus helpful to consider $\alpha$ as a predictor (see Sect. 2.5) to account for variations in the horizontal coherence caused by the direction misalignment. The benefit of doing
so is to make $\alpha$ more standalone and to prevent its errors from affecting everything else, while reasonably taking its influences into account. In addition, Gaussian process regression inherently assumes imperfect training data (containing noisy terms; see Sect. 2.6), so it is better to keep uncertainties in predictors.

Certainly, if the direction misalignment is available and sufficiently accurate in a given application scenario, the prediction concept can be easily adjusted by changing the wind evolution model to which the estimated coherence is supposed to fit.

**2.5 Potential Predictors**

In the literature reviewed in the introduction, the variables considered relevant to wind evolution are as listed below:

- Ropelewski et al. (1973): turbulence intensity (a function of roughness length and Richardson number (Lumley and Panofsky, 1964))

- Panofsky and Mizuno (1975): mean wind speed, turbulence intensity, standard deviation of the lateral wind component,
lateral integral length scale of the longitudinal wind component, longitudinal separation, and the angle between the wind direction and the measurement line (if misalignment exists)

- Kristensen (1979): turbulence intensity, longitudinal integral length scale of the longitudinal wind component, and longitudinal separation

- Simley and Pao (2015): turbulence intensity, longitudinal integral length scale of the longitudinal wind component, and
285 longitudinal separation

The above-mentioned variables can be categorized into three groups: wind statistics, atmospheric stability, and relative positions of measurement points. We follow this train of thought to discuss potential predictors of the parameterization models. It is worth mentioning, in advance, that not all of these predictors will be used in the final models. Useful features will be selected using the automatic relevance determination squared exponential kernel function (Duvenaud, 2014). The goal of this
initial step is to collect all possible predictors even though some of them will turn out to be redundant and can be converted to each other.

*Wind statistics.* Following prior research, turbulence intensity $I_T$ is considered as a predictor. The turbulence intensity is defined as

$$I_T = \frac{\sigma}{U}. \tag{17}$$

In addition, mean wind speed $U$ and its standard deviation $\sigma$ are also included because they are the fundamental variables of turbulence intensity. Apparently, $I_T$ and $\sigma$ are equivalent (given $U$), so only one of them will be selected according to the result of feature selection.

Moreover, integral length scale $L$ is considered as a predictor, and approximated with (Pope, 2000; Simley and Pao, 2015)

$$L = U \cdot \int\limits_0^\infty \rho(s)\mathrm{d}s = U \cdot T, \tag{18}$$

where $\rho(s)$ is the autocorrelation function. Indeed, integrating the autocorrelation gives the integral time scale $T$. The approximation of $L$ is essentially based on assuming the turbulent eddies advected by the mean flow at $U$. Please note that this is not necessarily equivalent to assuming 'frozen' turbulence. Turbulent eddies can evolve when preserving the same mean wind speed and statistical properties (including autocorrelation). The multiplication of $U$ can be understood as translating the integration domain from time lag $s$ to spatial separation by approximating the spatial separation with $U \cdot s$. This approximation might contain uncertainties, but we have no alternatives for calculation of $L$ from measured data. The integration of autocorrelation is computed up to the first zero-crossing instead of infinity in practice (Simley and Pao, 2015). Considering the correlation between $L$ and $T$ shown in Eq. (18), $T$ is also considered as a predictor, and thus $L$ and $T$ constitute another pair of redundant predictors from which only one will be selected.

Besides the variables already considered in prior studies, it is interesting to explore whether high-order wind statistics such as skewness and kurtosis of wind speed could play a role in wind evolution prediction. Skewness (i.e. the third standardized central moment) and kurtosis (i.e. the fourth standardized central moment) are measures of the asymmetry and flatness of the wind speed distribution, respectively. The sample skewness $G_1$, with bias correction, is defined as (Joanes and Gill, 1998)

$$G_1 = \frac{\sqrt{n(n-1)}}{n-2} \cdot \frac{\frac{1}{n}\sum_{i=1}^n u_i^3}{\left(\frac{1}{n}\sum_{i=1}^n u_i^2\right)^{3/2}}, \tag{19}$$

and the sample kurtosis $G_2$ (not substracting 3), with bias correction, is defined as (Joanes and Gill, 1998)

$$G_2 = \frac{n-1}{(n-2)(n-3)} \cdot \left[ (n+1) \cdot \frac{\frac{1}{n}\sum_{i=1}^n u_i^4}{\left(\frac{1}{n}\sum_{i=1}^n u_i^2\right)^2} - 3(n-1) \right] + 3, \tag{20}$$

where $u_i$ is wind speed fluctuations, and $n$ is the number of data points. According to Lenschow et al. (1994), statistical moments estimated using time series data with limited length show a systematic deviation from the true moments and also contain random errors. Both are decreasing functions of the averaging time. Compared to the sample standard deviation, the sample skewness and kurtosis would probably contain larger uncertainties. Nevertheless, we still want to test, on a simple level, whether these two high-order wind statistics could be useful for prediction.

*Atmospheric stability.* The atmospheric stability represents a global effect of the surface layer in the boundary layer on a wind field. It is believed to affect wind evolution being an influence factor on turbulence stability (Ropelewski et al., 1973; Lumley and Panofsky, 1964). A dimensionless height $\zeta$, built with Obukhov length $L_{\mathrm{MO}}$ (Obukhov, 1971), is considered as a predictor (Businger et al., 1971)

$$\zeta = \frac{z}{L_{\mathrm{MO}}} = -\frac{\kappa g \overline{w'\theta'_{\mathrm{v}}} z}{\overline{\theta} u_*^3}, \tag{21}$$

where $\kappa$ is the von Kármán constant, $g$ is gravitational acceleration, $z$ is the measurement height, $\overline{\theta}$ is the mean potential temperature, $u_*$ is the friction velocity, and $\overline{w'\theta'_\mathrm{v}}$ is the covariance of vertical velocity perturbations and virtual potential temperature.

***Relative positions of measurement points.*** Based on our modifications to Simley and Pao's (2015) model (see Sect. 2.3), measurement separation $d$ has been removed from the wind evolution model and is now considered as a predictor. As discussed in Sect. 2.4, the misalignment angle $\alpha$ is not involved in fitting the wind evolution model but is considered as a predictor to account for the influence of the lateral coherence on the horizontal coherence. In fact, $d$ is associated with two different effects. On the one hand, $d$ corresponds to travel time or, rather, to evolution time $\Delta t$, which is believed to play an important role in wind evolution. On the other hand, $d$ together with $\alpha$ account for the decay of the lateral coherence. The travel time determined with the maximum cross-correlation $\Delta t_\mathrm{M}$ is a more accurate variable. However, considering that calculating $\Delta t_\mathrm{M}$ might not always be feasible due to its computational complexity, the travel time approximated using Taylor's (1938) translation hypothesis $\Delta t_\mathrm{T}$ is included as well.

The notations of the above-mentioned potential predictors are summarized in Table 1. These variables are derived from both lidar data and data measured with ultrasonic anemometers (hereafter referred to as sonic data) according to their availability in each measurement campaign. The measurement instrument is indicated with a subscript: "l" for lidar and "s" for sonic (i.e. ultrasonic anemometer). For example, $U_\mathrm{l}$ represents the mean wind speed calculated from lidar data. Regarding sonic data, it is more reasonable for the analysis of wind evolution to use a wind coordinate system with the $x$-axis aligned to the mean wind direction instead of the meteorological coordinate system. The mean wind direction is determined with the mean wind direction for each data block. The high-resolution longitudinal (indicated with the subscript "x") and lateral (indicated with the subscript "y") wind speeds are obtained by projecting the high-resolution wind components measured with ultrasonic anemometers on the wind coordinate system. Then, the above-mentioned variables are derived from the data based on the wind coordinate system. For example, $U_\mathrm{x,s}$ represents the mean wind speed calculated from the longitudinal wind component measured with ultrasonic anemometers.

## 2.6 Gaussian Process Regression

This section briefly introduces the principle of the Gaussian process regression (GPR) and the hyperparameters that modify the behavior of a GPR model. The model training is done using the MATLAB Statistics and Machine Learning Toolbox[1].

***The principle of GPR.*** Consider making a regression model from some data. A very intuitive approach is to fit certain functions, e.g. linear or polynominal. However, this requires an initial guess about the functional relationship(s) behind the data, which is very difficult in this case because the wind evolution model parameters do not indicate any clear dependence on the potential predictors. The reasons for that could be multiple: 1) the data could be noisy; and 2) the dependence could exist in multidimensional space not observable in a single dimension, etc. Under this circumstance, GPR turns out to be a good choice because it is non-parametric probabilistic model, which means the model is not a specific function, but a probability distribution over functions. The principle underlying GPR is Bayesian inference. The prior distribution over functions, which

---

[1]https://de.mathworks.com/products/statistics.html

**Table 1.** Notations of potential predictors.

| Notation | Variable | Unit |
|---|---|---|
| $U$ | mean wind speed | $[\mathrm{m\,s^{-1}}]$ |
| $\sigma$ | standard deviation of wind speed | $[\mathrm{m\,s^{-1}}]$ |
| $G_1$ | skewness of wind speed | [-] |
| $G_2$ | kurtosis of wind speed | [-] |
| $I_\mathrm{T}$ | turbulence intensity | [-] |
| $T$ | integral time scale | [s] |
| $L$ | integral length scale | [m] |
| $\zeta$ | dimensionless Obukhov length | [-] |
| $d$ | measurement separation | [m] |
| $\alpha$ | angle between wind direction and lidar measurement | [°] |
| $\Delta t_\mathrm{M}$ | travel time determined by the maximum cross-correlation | [s] |
| $\Delta t_\mathrm{T}$ | travel time approximated by $d/U$ | [s] |

can be understood as a guess about what kinds of function could be present without knowing the data, is specified by a particular
Gaussian process (GP) which favors smooth functions. In the training process, as adding the data, the probabilities associated
with the functions which do not agree with the observations will be decreased, which gives the posterior distribution over the
functions (Rasmussen and Williams, 2006).

*Hyperparameters of GPR*. The behavior of a GPR model is defined by its hyperparameters. To introduce the hyperparameters, a basic explanation is given following Rasmussen and Williams (2006). Please note that the complete deduction is not
displayed here because it is beyond the scope of this paper. For further details, please refer to Chapter 2 of Rasmussen and
Williams' (2006) book.

GPR is based on Bayesian inference. First, consider a single observation. The Bayesian linear regression model with Gaussian noise is defined as

$$f(\boldsymbol{x}) = \phi(\boldsymbol{x})^\top \boldsymbol{w}, \quad y = f(\boldsymbol{x}) + \varepsilon, \tag{22}$$

where $\boldsymbol{x}$ is an input vector containing $D$ different predictors of a single observation, $\phi(\boldsymbol{x})$ is the function which maps the input
vector onto a higher dimensional space where the Bayesian linear model is applicable, $\boldsymbol{w}$ is a vector of weights of the linear
model, $f(\boldsymbol{x})$ is the function value, $y$ is the observed target value, and $\varepsilon$ is independent identically distributed Gaussian noise
with zero mean and variance $\sigma_\mathrm{n}^2$

$$\varepsilon \sim \mathcal{N}(0, \sigma_\mathrm{n}^2). \tag{23}$$

The Bayesian linear model is a GP given that the prior distribution of $\boldsymbol{w}$ is normally distributed with zero mean. Since a GP is fully specified by its mean and covariance, the Bayesian linear model is written as

$$f(\mathbf{X}) \sim \mathcal{GP}(\mathbf{0}, \mathrm{cov}(f(\mathbf{X}))), \tag{24}$$

where $\mathbf{X}$ is the aggregation of all input vectors of $n$ observations. This is the prior distribution over functions. The presence of $\varepsilon$ shows another advantage of GPR, viz. that it is able to inherently assume noisy observations and take this effect into account

in the model. $\sigma_\mathrm{n}$ is one of the hyperparameters.

   It is common, but not necessary, to assume GPs with a zero mean function. The mean function can be modeled with a set of basis functions $\mathbf{h}(\boldsymbol{x})$ and a corresponding coefficient vector $\boldsymbol{\beta}$. So, GPs with a non-zero mean function can be assumed as

$$g(\boldsymbol{x}) = f(\boldsymbol{x}) + \mathbf{h}(\boldsymbol{x})^\top \boldsymbol{\beta}. \tag{25}$$

The basis function is one of the hyperparameters. MATLAB provides four types of basis function: zero (assuming no basis

function), constant, linear, and pure quadratic. The coefficient vector $\boldsymbol{\beta}$ can also be understood as the weight vector of $\mathbf{h}(\boldsymbol{x})$. But we have defined $\boldsymbol{w}$ as a weight vector in Eq. (22), we want to avoid using the same word here in case reader might confuse these two different processes. $\boldsymbol{\beta}$ is estimated from training data.

   The covariance of the function values is not specified explicitly but estimated using a kernel function

$$\mathrm{cov}(f(\mathbf{X})) = \mathrm{K}(\mathbf{X}, \mathbf{X}), \tag{26}$$

which is the so-called *kernel trick*. There are two types of kernel functions: one is kernel functions with the same characteristic length scale for all predictors; the other has separate characteristic length scales. The latter are called automatic relevance determination kernel functions and can be used to select predictors. The kernel function and its characteristic length scale(s) are hyperparameters of the GPR model.

   In this work, Automatic Relevance Determination Squared Exponential kernel function (ARD-SE kernel) (Duvenaud, 2014)

is applied. The ARD-SE kernel function is basically a squared exponential kernel function (SE kernel) with a separate characteristic length scale $\sigma_\mathrm{m}$ for each predictor $m$ ($m$ is the index of predictors). For any pairs of observations $i, j$, the ARD-SE kernel function is defined as

$$\mathrm{K}(\boldsymbol{x_i}, \boldsymbol{x_j}) = \sigma_f^2 \exp\left[ -\frac{1}{2} \sum_{m=1}^{D} \frac{(x_{im} - x_{jm})^2}{\sigma_m^2} \right], \tag{27}$$

where $\sigma_f^2$ denotes the signal variance, which determines the variation of function values from their mean. In the context of

machine learning, the characteristic length scale $\sigma_m$ is not a 'length' in the physical sense; it is a characteristic magnitude for the predictor $m$ which implies the sensitivity of the function being modeled to the predictor $m$. A relatively large $\sigma_m$ indicates a relatively small variation along the corresponding dimensions in the function, which means these predictors are less relevant than the others (Duvenaud, 2014).

   In the end, the key predictive equation for GPR can be derived by conditioning the joint Gaussian prior distribution on the

observations, and it is normally distributed.

$$\boldsymbol{f_*}|\mathbf{X}, \boldsymbol{y}, \mathbf{X}_* \sim \mathcal{N}(\overline{\boldsymbol{f}}_*, \mathrm{cov}(\boldsymbol{f_*})), \tag{28}$$

where $\mathbf{X}_*$ denotes new input data used in prediction. $\boldsymbol{f}_*$ represents $\boldsymbol{f}(\boldsymbol{X}_*)$ for convenience, which is the predicted function value.

To summarize, the hyperparameters defining a GPR model are the basis function $\mathbf{h}(\boldsymbol{x})$, the noise standard deviation of the Gaussian process model $\sigma_\mathrm{n}$, the kernel function $\mathrm{K}(\boldsymbol{x_i}, \boldsymbol{x_j})$, the standard deviation of the function values $\sigma_\mathrm{f}$, and the characteristic length scale in the kernel function $\sigma_\mathrm{m}$. These hyperparameters can be tuned in the training process to achieve a better model.

## 2.7 Model Validation

The trained model is evaluated with a $k$-fold cross-validation in which the data is divided into $k$ disjoint, equally sized subsets. The model validation is done with one subset (also called in-fold observations) and the training is done with the remaining $(k-1)$ subsets (also called out-of-fold observations). This procedure is repeated $k$ times, each time with a different subset for validation. The predicted target values and the goodness-of-fit measures of the regression models are computed for in-fold observations using a model trained on out-of-fold observations.

Theoretically, $k$ can be any integer between two and the number of observations (a special case called 'leave-one-out'). When $k$ is very small, the sample size of training data ($\frac{k-1}{k}$ of the total observations) could be insufficiently large. However, considering that the training process must be repeated $k$ times, it would take a very long time when $k$ is very large. As a compromise between these two factors, $k$ is commonly set to 5 to 10 in machine learning. In this study, 5-fold cross-validation is applied.

The model performance is evaluated with two goodness-of-fit measures: root-mean-square error (RMSE)

$$\mathrm{RMSE} = \sqrt{\frac{1}{N} \sum_i^N (y_i - y_{\mathrm{pred},i})^2} \tag{29}$$

and the coefficient of determination ($R^2$)

$$R^2 = 1 - \frac{\sum_i^N (y_i - y_{\mathrm{pred},i})^2}{\sum_i (y_i - \overline{y})^2}, \tag{30}$$

where $y$ and $y_{\mathrm{pred}}$ denote the observed and predicted target values respectively, $\overline{y}$ denotes the average of the observed target values, $N$ denotes the number of observations. It is worth mentioning that, according to this definition, $R^2$ can be understood as taking prediction with the mean value of the observations as a reference by which to evaluate the model performance. In this case, $R^2$ ranges from $-\infty$ to one, for perfect prediction. $R^2$ equals zero if the prediction is made simply with the mean value of the observations. The higher $R^2$ is, the better the model performs. A negative value of $R^2$ indicates that the selected model performs even worse than prediction using just the mean value of the observations.

## 3 Data Processing

This section first introduces the data sources in Sect. 3.1 and then explains the procedure for the determination of the wind evolution model parameters in Sect. 3.2.

### 3.1 Data Source

This study involves measured data from two research projects. The reasons for using two different data sources are, on the one hand, to find commonality between two different measurements and avoid accidental conclusions, and, on the other hand, to study whether there are differences or what kind of differences in the wind evolution can be observed. The relevant research projects as well as the measurement campaigns are (briefly) as follows:

*LidarComplex.* The research project LidarComplex was funded by the German Federal Ministry for Economic Affairs and Energy (BMWi). In this project, a lidar measurement campaign was carried out in Grevesmühlen, Germany. The measurement site is basically flat, mainly farmland with hedges and few large trees. More details about the measurement campaign can be found in Schlipf et al. (2015). The lidar deployed in this measurement campaign was the SWE Scanner 1.0, which was adapted from a WindCube V1 from Leosphere (Schlipf et al., 2015). This lidar has five measurement range gates focusing at distances of $54.5\,\mathrm{m}$, $81.75\,\mathrm{m}$, $109\,\mathrm{m}$, $136.25\,\mathrm{m}$, and $163.5\,\mathrm{m}$, respectively. The full width at half maximum (FWHM) of the measurement range gates is $30\,\mathrm{m}$ (Carious, 2013). The lidar was installed on the nacelle of a wind turbine (rotor diameter of $109\,\mathrm{m}$) at $95\,\mathrm{m}$. In addition, a meteorological mast is located $295\,\mathrm{m}$ southwest of the wind turbine; data from an ultrasonic anemometer installed at $93\,\mathrm{m}$ on the meteorological mast is also involved in this study. SCADA data of the wind turbine is also available. Recorded yaw positions are used to estimate the misalignment angle $\alpha$, assuming that the mean wind direction at the turbine can be approximated with the mean wind direction measured on the meteorological mast.

*ParkCast.* The ParkCast[2] project is an ongoing project funded by the German Federal Ministry for Economic Affairs and Energy (BMWi). While this paper is in preparation, a lidar measurement campaign is being conducted on the offshore wind farm *alpha ventus*[3]. Two long-range lidars (StreamlineXR) have been deployed in the measurement campaign. The data used here is from the lidar installed on the nacelle of wind turbine AV4 (rotor diameter of $126\,\mathrm{m}$) at $92\,\mathrm{m}$, measuring the inflow. The measurement distances were set to $30\,\mathrm{m}$ to $990\,\mathrm{m}$ with an increment of $60\,\mathrm{m}$. The FWHM of the measurement range gates is $60\,\mathrm{m}$. Unfortunately, neither data from the meteorological mast on FINO1[4] nor SCADA data of AV4 for the observed period was available when the analysis was done. Therefore, the misalignment angle $\alpha$ is not available for ParkCast.

Compared to ultrasonic anemometers, lidar systems have much lower sampling rates. To obtain the highest possible sampling rate, we select the measurement periods where the staring mode was used, for both campaigns.

Essential information about the measurements is summarized in Table 2. Figure A1 gives an overview of the wind statistics of these two selected measurement periods by illustrating the relative frequency distribution of lidar-measured wind speed and turbulence intensity. For brevity, 'LidarComplex' and 'ParkCast' are used to refer to the selected measurements throughout the paper.

---

[2]https://www.rave-offshore.de/en/parkcast.html
[3]https://www.alpha-ventus.de/english
[4]https://www.fino1.de/en/

**Table 2.** Summary of measurement setups.

| Measurement campaign | LidarComplex | ParkCast |
|---|---|---|
| Selected period | 02 Dec 2013–20 Dec 2013 | 04 Jun 2019–14 Jun 2019 |
| Location | Grevesmühlen, Germany | alpha ventus |
| Terrain type | onshore, flat | offshore |
| Device | nacelle based lidar + met mast | nacelle based lidar |
| Measurement height [m] | 95 (lidar), 93 (sonic) | 92 |
| Range gate [m] | 54.5, 81.75, ..., 163.5 | 30, 90, ..., 990 |
| Number of range gates | 5 | 17 |
| Full width at half maximum [m] | 30 | 60 |
| Sampling rate [Hz] | 0.99 | 0.27 |
| Valid samples[*] | 3285 | 10112 |

[*]After lidar data filtering, data pairing, and outlier filtering. For details see Sect.3.2

### 3.2 Determination of Wind Evolution Model Parameters

To obtain the wind evolution model parameters $a$ and $b$, the wind evolution is estimated with lidar data and then fitted to the wind evolution model (Eq. (8)). The processing procedure is described as follows:

*Step 1: Filtering of the lidar data.* The lidar data from LidarComplex is filtered according to the carrier-to-noise ratio (CNR) of the lidar signals (*CNR filter*). The valid range of the CNR filter is $-24\,\mathrm{dB}$ to $-5\,\mathrm{dB}$, determined from the plot of CNR values and wind speed.

A CNR filter is not, however, suitable for lidar data from ParkCast because, for a long-range lidar, the backscattered signals from distant range gates could be very weak, and thus the CNR values could be low even when the measured wind speed is plausible. Würth et al. (2018) suggested an approach to filter the data based on the value range (*range filter*) and the standard deviation (*standard deviation filter*) within a certain number of adjacent data points defined as a window, which can keep more valid data than a CNR filter. A range filter detects the maximum value difference within a window and filters the data points for which the maximum value difference exceeds a threshold. A standard deviation filter calculates the standard deviation within a window and filters the data points for which the standard deviation exceeds a threshold. Both filters are applied to check the line-of-sight wind speed with thresholds of $6\,\mathrm{m\,s^{-1}}$ and $3\,\mathrm{m\,s^{-1}}$, respectively. The window size is set to three data points.

*Step 2: Estimation of coherence.* The lidar data is divided into 30-minute blocks. This is consistent with the commonly used period for calculating the Obukhov length. Only the data blocks with more than $80\,\%$ valid data points are used to estimate the coherence. The missing values are estimated by shape-preserving piecewise cubic interpolation (Fritsch and Carlson, 1980). The missing end values are each replaced with their nearest value. Data measured at different range gates (i.e. measurement distances) is paired in the way shown in Fig. 4 to obtain as many samples (i.e. data blocks) as possible. The pairing has $\binom{N}{2}$ possibilities ($N$ is the number of the lidar range gates). The travel time of the wind field is approximated with the time lag

at the maximum of the cross-correlation $\Delta t_{\mathrm{M}}$ between these two wind speed signals. The upstream point is always regarded as the reference point. The data measured at the downstream point is shifted by $\Delta t_{\mathrm{M}}$ to match the reference wind speed data. The magnitude-squared coherence is estimated using Welch's overlapped averaged periodogram method using a Hamming window, 24 segments, and $50\,\%$ overlap. The data of the reference point is used to calculate lidar-measured wind statistics.

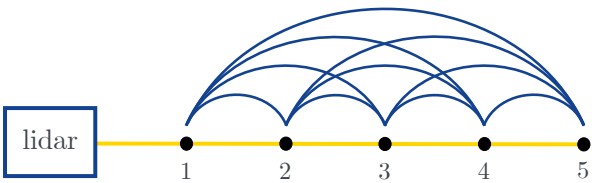

**Figure 4.** Pairing of different measurement points for estimating coherence for LidarComplex, given as an example.

***Step 3: Fitting to the wind evolution model.*** Before fitting the model, we must consider two issues that might introduce noise into the coherence estimate. Firstly, because both lidars are installed on the nacelle of a wind turbine which is actually in motion, the focus points of the laser beams are moving as well. This motion causes excitation at certain frequencies in the estimated coherence. Figure A2 shows a comparison between an example coherence curve and the power spectral density (PSD) of the fore-aft and in-plane tower top acceleration of LidarComplex. The excitation in the coherence conforms to that
in both PSDs and occurs mainly at frequencies above $0.2\,\mathrm{Hz}$. To avoid negative effects on the fitting quality caused by this excitation, the cut-off frequency is hence set at $0.2\,\mathrm{Hz}$, and the coherence is fitted only up to this cut-off frequency.

    Secondly, according to Schlipf (2015), critical wavenumbers where the lidar signals would be only determined by noises must be checked. The critical wavenumbers are $2\pi/W_{\mathrm{L}}$ ($W_{\mathrm{L}}$ is the full width at half maximum of the range gate) and its harmonics. As mentioned in Sect. 2.3, the relationship between wavenumber $k$ and dimensionless frequency $f_{\mathrm{dless}}$ is $f_{\mathrm{dless}} =$
$kd/2\pi$. Thus, the smallest critical value of $f_{\mathrm{dless}}$ is $d/W_{\mathrm{L}}$. Considering LidarComplex as an example, $W_{\mathrm{L}} = 30\,\mathrm{m}$ and $d = 27.25\,\mathrm{m}$ for the smallest separation, which is the most critical case. $d/W_{\mathrm{L}} \approx 0.91$, which is already located in the filtered part (see the grey area in Fig. 5 (a)).

    The fitting is done by a nonlinear least-squares method using the Levenberg-Marquardt algorithm (Levenberg, 1944; Marquardt, 1963; Moré, 1978). Only the data blocks with $R^2 > 0.8$ are considered as valid samples.

***Step 4: Outlier filtering.*** The final filtering was done by checking the value distribution of every relevant variable to omit outliers. It is emphasized that outliers are not necessarily false data. In some cases, the outlier is from a value range in which not enough samples were collected. It is very important to filter outliers properly because it is difficult for a regression model to capture the relationship for those value ranges with too few samples. Because the distributions of the variables all have a long right tail, the outliers are chosen as all data exceeding the 99th percentile of the data. Figure 5 is an example plot of the data
block from 07 Dec. 2013, 12:00–12:30, from LidarComplex. This data block is selected here for two reasons: data integrity and representative wind statistics. In this data block, the lidar-measured mean wind speed is $7.3\,\mathrm{m\,s}^{-1}$ to $7.7\,\mathrm{m\,s}^{-1}$, and the lidar-measured turbulence intensity is $0.10$ to $0.12$, for different range gates. These values appeared frequently in the selected

period according to Fig.A1. Hence, this data block is regarded as a representative case-study example for LidarComplex and is referred to throughout the paper. The figure illustrates the estimated coherence between different range gates and the corresponding fitted curves. The shaded areas show that the selected cut-off frequency of $0.2\,\text{Hz}$ is reasonable for this case. A similar plot from ParkCast is found in Fig. A3. Because the sampling rate of ParkCast is lower, the excitation by the nacelle's movement is not observed in the coherence, and thus no cut-off frequency was set for ParkCast data.

In Fig. 5 (c) and (d), the intercept of the coherence is much lower than 1 even though the separation is not very large. This confirms the necessity of choosing a wind evolution model which is able to define different offset values depending on the conditions. Indeed, compared with the fitting quality of Pielke and Panofsky's model which contains merely a single parameter — the decay parameter $a$ — the fitting quality of the wind evolution model (Eq. (8)) is overall better (see Fig. A4). The value of $R^2$ for the fitting of Eq. (8) is almost always higher than for the fitting of Pielke and Panofsky's (1970) model. The wind evolution model used in this work (Eq. (8)) is thus proven able to model the coherence better.

## 4  Statistical Analysis of Wind Evolution

This section presents a statistical analysis of wind evolution, including the distributions of the wind evolution model parameters (Sect. 4.1) and their dependence on measurement separation (Sect. 4.2).

### 4.1  Distribution of the Wind Evolution Model Parameters

To study the overall characteristics of wind evolution, the distributions of the wind evolution model parameters for both measurements are displayed in Fig.6.

As listed in Table 2, there are two main differences between the lidar settings in both measurements: sampling rate and measurement range, which might affect the distributions of the wind evolution parameters. To enhance the comparability of both distributions, two special post-processings are executed correspondingly. Firstly, because the lidar sampling rate of LidarComplex is approximately three times that of ParkCast, an artificial data set is made for LidarComplex by averaging every three data points of the original lidar data to simulate measurement at a sampling rate similar to that of ParkCast, so that the distributions of both measurements can be compared. The fitted probability density function (PDF) of the wind evolution model parameters determined with this data set are plotted as yellow dashed lines in Fig. 6 (a) and (b). The comparison between the fitted PDF of the original data and that of the data with reduced sampling rate indicates that the lidar sampling rate only very slightly affects the wind evolution model parameters, or, perhaps more accurately, the estimated coherence. Hence, the different sampling rates do not account for the differences between the cases observed in Fig. 6. Secondly, because of the limited measurement range of LidarComplex, the maximum separation between two range gates reaches only $109\,\text{m}$, while that of ParkCast reaches more than $700\,\text{m}$. To make them comparable, Fig. 6 (c) and (d) show only the wind evolution parameters calculated from the coherence with separation below $120\,\text{m}$ of ParkCast.

Apart from that, the measurements were carried out in different environments (onshore and offshore), at different times of the year (which impacts atmospheric stability), and have different wind speed and turbulence intensity distributions (see Fig.

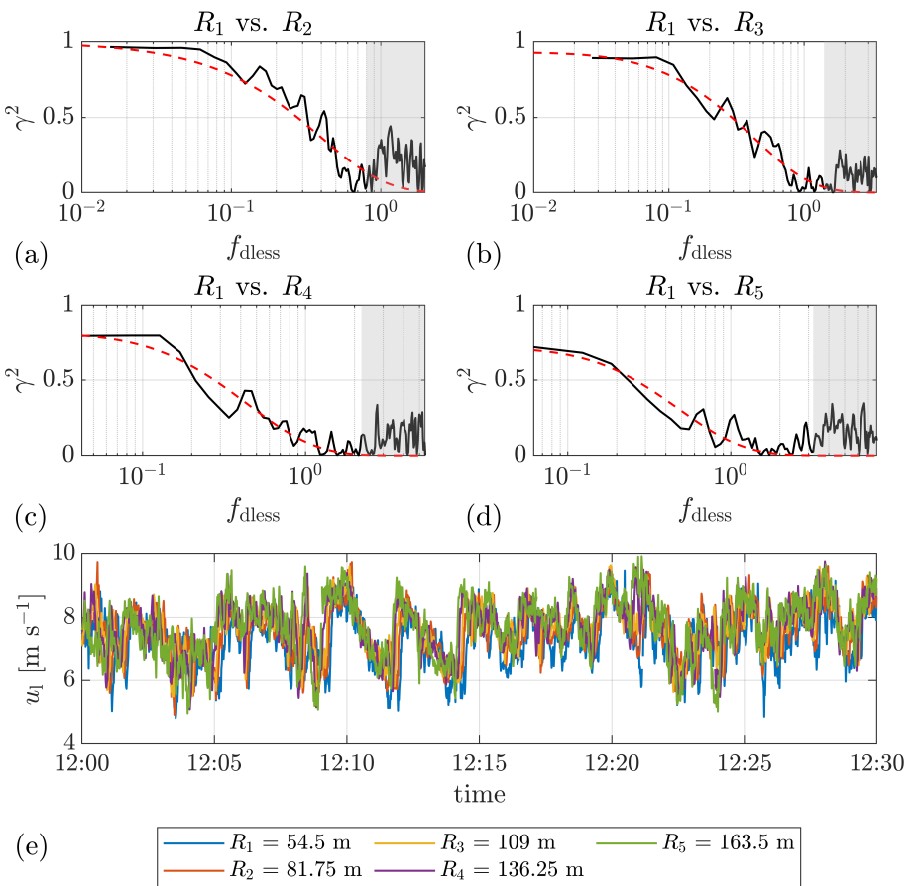

**Figure 5.** (a) – (d) Example plots of the estimated coherence between the lidar wind speeds measured at different range gates and the corresponding fitted curves. The separations between the corresponding range gates are $27.25\,\mathrm{m}$, $54.5\,\mathrm{m}$, $81.75\,\mathrm{m}$, and $109\,\mathrm{m}$, respectively. The shaded areas indicate the data filtered by the cut-off frequency $0.2\,\mathrm{Hz}$. (e) Time series of the lidar wind speed. The mean lidar wind speed $U_l$ ranges from $7.3\,\mathrm{m\,s^{-1}}$ to $7.7\,\mathrm{m\,s^{-1}}$ and the lidar measured turbulence intensity $I_{T,l}$ ranges from $0.10$ to $0.12$, for different range gates. Date: 07 Dec. 2013. Data source: LidarComplex.

A1). Despite these differences, the distributions of the wind evolution model parameters do have some common characteristics. First of all, the value ranges of both wind evolution model parameters for both measurements are similar; $a$ ranges mostly from $0$ to $6$ and $b$ from $0$ to $0.5$. Values out of these ranges are less likely to happen, according to the measurements. Second, the values of $a$ and $b$ are found to follow an inverse Gaussian distribution and a Gamma distribution, respectively. These two PDFs

are determined by fitting the histograms to all the PDFs supported by the MATLAB Statistics and Machine Learning Toolbox and searching for the one with the maximum likelihood. This is done using a tool called *fitmethis*[5].

The corresponding fitted parameters of the PDFs (orange curves) are displayed in Table 3. It is interesting to observe that the peak of the probability density is located around $a = 1.8$ for the onshore LidarComplex, while around $a = 0.8$ for the offshore ParkCast. Moreover, the medians of $a$ are approximately 2.0 and 1.5 for LidarComplex and ParkCast, respectively. The mean (see $\mu$ in Table 3) and median of $a$ as well as its value of the peak location of the PDF of LidarComplex are all higher than that of ParkCast. This indicates that the coherence under similar separation generally decays faster in an onshore location than an offshore location. In terms of $b$, most of the values are near 0, and values higher than 0.1 are not often observed. Therefore, the $y$-axes in Fig. 6 (b) and (d) are plotted logarithmically to make the higher-value part of $b$ visible. However, $b$ shows no significant difference between the two cases observed in the figure.

It is not yet possible to explain the physical relationship between the wind evolution model parameters and the above-mentioned PDFs and the physical meaning of the corresponding PDF parameters. To verify whether the above-discussed phenomena commonly occur in wind evolution, further research involving more different measurement campaigns is necessary. At this point, a hypothesis is made that the values of $a$ and $b$ might follow an inverse Gaussian distribution and a Gamma distribution, respectively. The corresponding PDF parameters might depend on the terrain types, on the one hand. It is not clear if the roughness length would be a suitable parameter to quantify the influence of the terrain type on the value distribution of wind evolution model parameters. To figure out a concrete relationship between the PDF parameters and the terrain types, again, it is necessary to involve more measured data gathered from different terrain types. On the other hand, unfortunately, it is not yet possible to estimate to what extent the atmospheric stability would affect the distribution of the wind evolution model parameters because there was no sonic data available for ParkCast to inform the associated investigation until this work was finished.

**Table 3.** Parameters of the fitted probability density functions.

| wind evolution model parameters | PDF | LidarComplex | ParkCast |
|---|---|---|---|
| $a$ | inverse Gaussian distribution $\mathrm{f}(x;\mu,\lambda) = \sqrt{\frac{\lambda}{2\pi x^3}}\exp\left[-\frac{\lambda(x-\mu)^2}{2\mu^2 x}\right]$ | $\mu = 2.07$ $\lambda = 17.23$ | $\mu = 1.86$ $\lambda = 2.38$ |
| $b$ | Gamma distribution $\mathrm{f}(x;k,\theta) = \frac{1}{\Gamma(k)\theta^k}x^{k-1}e^{-\frac{x}{\theta}}$ | $k = 0.42$ $\theta = 0.18$ | $k = 0.24$ $\theta = 0.16$ |

Note: the notations $\mu, \lambda, k, \theta$ are independent from the other notations in the table.

---

[5]Francisco de Castro (2020). fitmethis (https://www.mathworks.com/matlabcentral/fileexchange/40167-fitmethis), MATLAB Central File Exchange. Retrieved Jan 13, 2020.

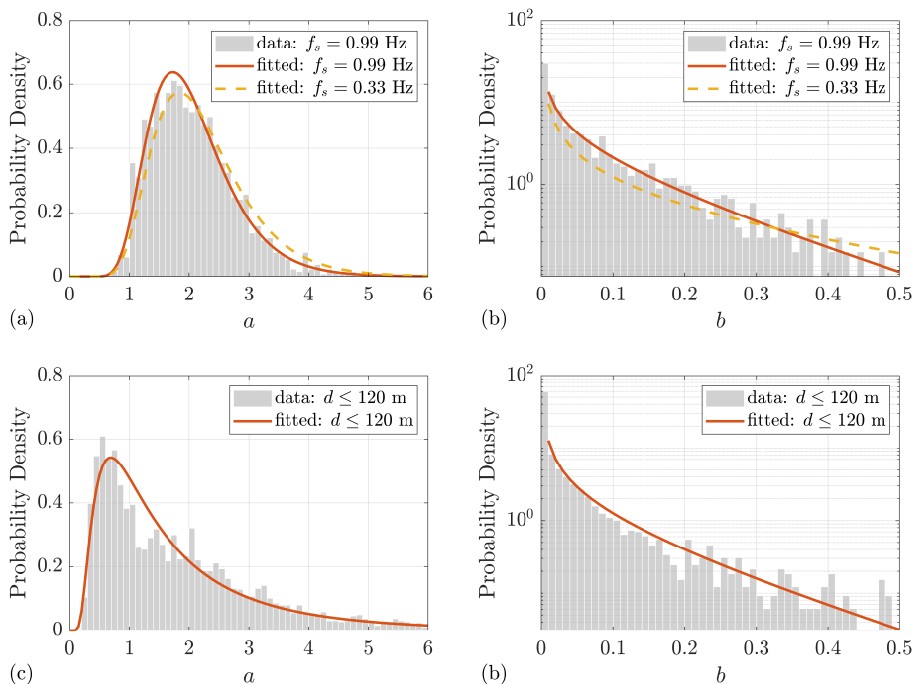

**Figure 6.** Distribution of wind evolution model parameters. (a) and (b): LidarComplex. (c) and (d): ParkCast. The curves show the corresponding fitted probability density function.

## 4.2 Dependence of the Wind Evolution Model Parameters on measurement separation

Figure 7 shows the fitted curves of the estimated coherence of all pairings of the above-mentioned LidarComplex case-study example. Each color indicates a particular range gate, while each marker indicates a particular measurement separation. The figure shows a very clear dependence of the fitted curve form on the measurement separation — the curves with the same marker overlap despite having different range gates. This confirms that the coherence depends on the separation of the measurement points but not on their positions, even though in the wind turbine's induction zone (defined as within $2.5$ rotor diameters on the inflow side of the wind turbine). Since the curve offset is related only to the offset parameter $b$, obviously, $b$ must strongly depend on the measurement separation. In addition, that all the fitted curves of the coherence are grouped together suggests it is reasonable to model the wind evolution based on the dimensionless frequency. Similar conclusions can be drawn from the example plot of ParkCast (see Fig. A5), which proves that these conclusions are not accidental.

To further study the dependence of the wind evolution model parameters on the measurement separation, the box plots of the wind evolution parameters, grouped by the measurement separations, are given in Fig. 8. Although the ranges of the measurement separation from the two measurement campaigns are very different, the box plots still show similar trends. The decay parameter $a$ shows a decreasing trend with increasing measurement separation. This decreasing trend of $a$ gradually stops at a separation of about $300\,\mathrm{m}$, as observed in Fig. 8 (c). The offset parameter $b$ shows an increasing trend with separation. An

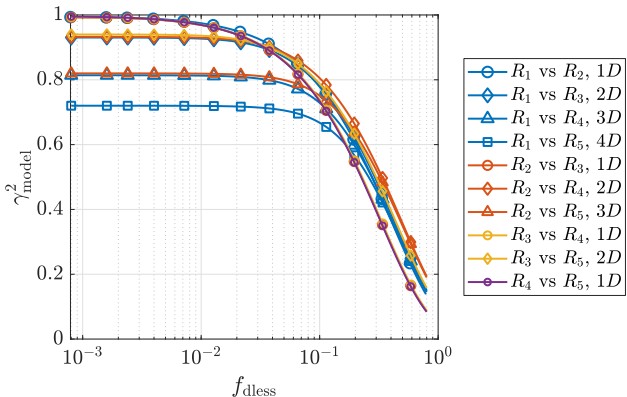

**Figure 7.** Fitted curves of the estimated coherence between the lidar wind speeds measured at different range gates. The range gates $R_1$ to $R_5$ are located at $54.5\,\mathrm{m}$, $81.75\,\mathrm{m}$, $109\,\mathrm{m}$, $136.25\,\mathrm{m}$, and $163.5\,\mathrm{m}$, respectively. $1D = 27.25\,\mathrm{m}$. The mean lidar wind speed $U_1$ ranges from $7.3\,\mathrm{m\,s^{-1}}$ to $7.7\,\mathrm{m\,s^{-1}}$ and the lidar measured turbulence intensity $I_{\mathrm{T,l}}$ ranges from 0.10 to 0.12, for different range gates. Date and time: 07 Dec. 2013, 12:00-12:30. Data source: LidarComplex.

increase in $b$ implies a decreased offset of the coherence curve. This is consistent with the phenomena observed from Fig. 7 and Fig. A5.

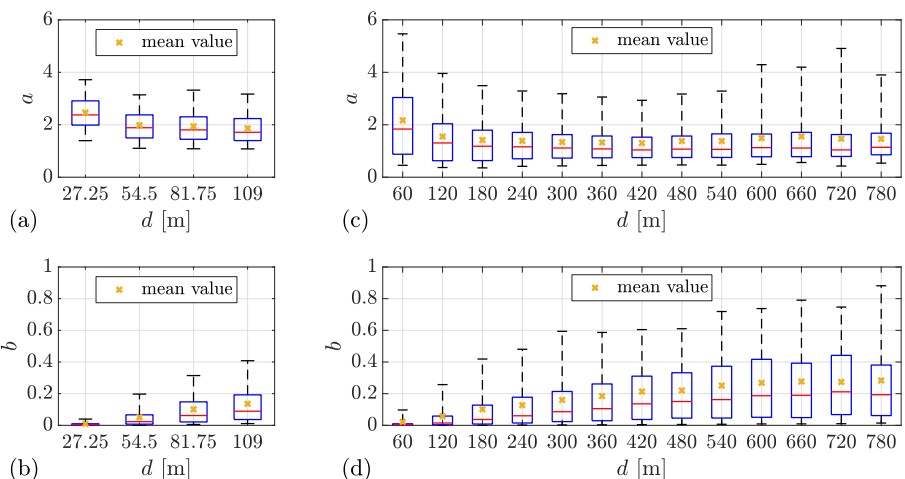

**Figure 8.** Box plots of the wind evolution model parameters grouped by the measurement separations $d$. (a) and (b): LidarComplex. (c) and (d): ParkCast. The bottom and top of the boxes indicate the first ($25^{\mathrm{th}}$ percentile) and third ($75^{\mathrm{th}}$ percentile) quartiles. The lower and upper whiskers show $5^{\mathrm{th}}$ (bottom) and $95^{\mathrm{th}}$ (top) percentiles. The red line in the middle indicates the median value. Minimum sample size is 50.

The decay of coherence is supposed to result from the evolution of turbulence eddies depending on travel time. The dependence of the decay parameter $a$ on the measurement separation, or rather the travel distances, actually reveals the dependence of $a$ on the travel time. Figure 9 shows the correlation between $a$ and the travel time approximated by $\Delta t_{\mathrm{M}}$ of ParkCast. The

590 fitted curve represents a negative correlation trend between them. This implies that the decay rate of the coherence decreases with increasing travel time. The nonlinear least-squares fitting is done using the Levenberg-Marquardt algorithm (Levenberg, 1944; Marquardt, 1963; Moré, 1978).

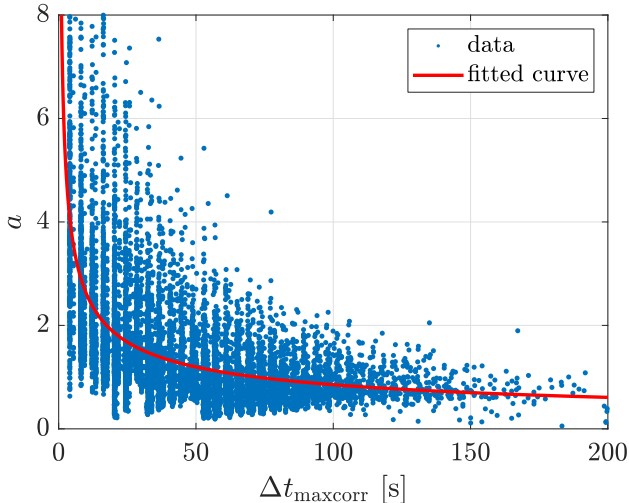

**Figure 9.** Correlation between the decay parameter $a$ and the travel time approximated by $\Delta t_{\mathrm{M}}$. The fitted curve: $a = 8.06 \cdot \Delta t_{\mathrm{M}}^{-0.49}$. Data source: ParkCast.

## 5 Parameterization Model

This section first presents the training procedure of GPR models with application of the ARD-SE kernel to select the suitable

predictors in Sect. 5.1. Following that is a discussion of the selected predictors in Sect. 5.2, and an evaluation of the model performance in Sect. 5.3.

### 5.1 Model Training

The initial settings for GPR model training are listed in Table 4. The 'exact GPR' setting means that a standard GPR is applied in the fitting and prediction process; otherwise GPR can be approximated using different methods to reduce the computation

time for large amounts of training data. The initial values of $\sigma_{\mathrm{n}}$, $\sigma_{\mathrm{f}}$, and $\sigma_m$ listed in the table are just used to initiate the training process, and their final values will be estimated from the training data by the GPR algorithm. The training data is standardized by centering and scaling the data of each predictor by its mean and standard deviation, respectively, which gives the standard scores (also called $z$-scores) (Kreyszig, 1979; Mendenhall and Sincich, 2007) of the predictor data.

Training the model is a two-step process. In the first step, all the potential predictors are included in a preliminary training

to determine the characteristic length scale $\sigma_m$ for each predictor (see Eq. 27). Figure 10 illustrates a comparison among the $\log(\sigma_m^{-2})$ of all potential predictors. As explained in Sect. 2.6, the larger $\log(\sigma_m^{-2})$ is, the more important and useful the

**Table 4.** Initial settings of GPR model training.

| Hyperparameter | Setting |
| --- | --- |
| Basis function | constant |
| Kernel function | ARD-SE |
| Fitting method | exact GPR |
| Prediction method | exact GPR |
| Initial value of $\sigma_n$ | standard deviation of observed target values |
| Initial value of $\sigma_f$ | standard deviation of observed target values |
| Initial value of $\sigma_m$ | 10 |
| Standardization | true |

corresponding predictor is for a GPR model, and thus this predictor should be selected. In the second step, new GPR models are trained only with the selected predictors, applying a 5-fold cross-validation to evaluate the model performance, using RMSE (see Eq. 29) and $R^2$ (see Eq. 30) as criteria.

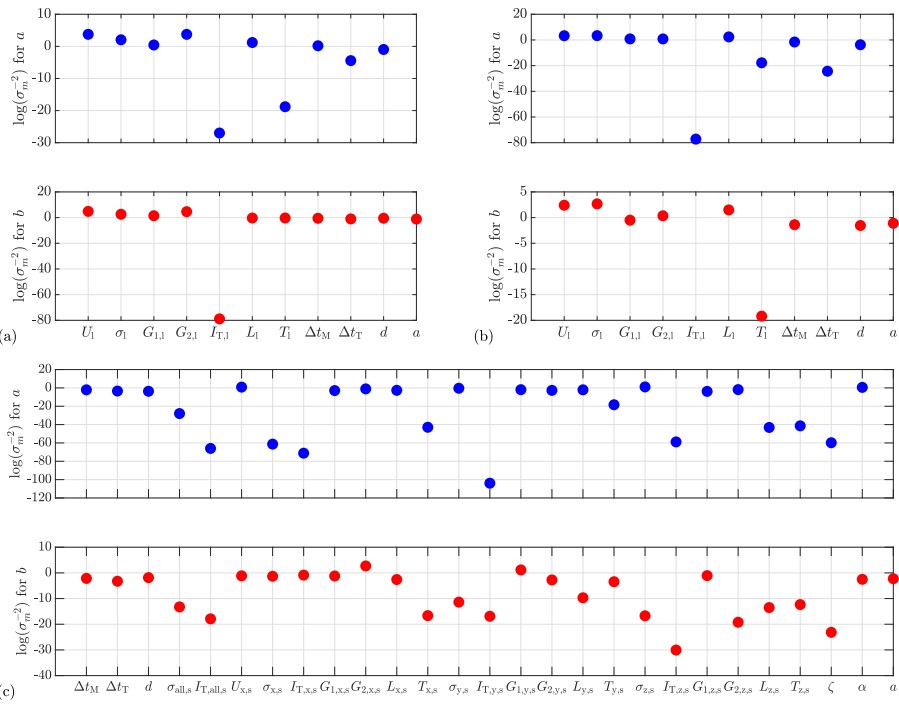

**Figure 10.** Comparison of the relative importance of predictors. (a) ParkCast, lidar data. (b) LidarComplex, lidar data. (c) LidarComplex, sonic data. $\log(\sigma_m^{-2}) = -\infty$ is not displayed.

Table 5 displays the predictors selected according to different lower limits of $\log(\sigma_m^{-2})$ under different measurement campaigns (LidarComplex or ParkCast), different data availability (whether sonic data is available), and different targets ($a$ or $b$). $R^2$ and the RMSE of the 5-fold cross-validation for the model trained with the respective combination of predictors are shown in the table as well.

In general, the more relevant predictors are involved in the model, the more accurate predictions the model can make. However, using more predictors entails a larger training data set and thus a longer model training time. On the other hand, it might also reduce the applicability of the model because predictions can only be made when all predictors are consistently available and reliable. The trade-off between these factors must be considered in predictor selection, and it is aimed to achieve relatively high model performance with as few predictors as possible. The bold text in Table 5 indicates the recommended predictor combinations for each situation based on these considerations. The predictors with $\log(\sigma_m^{-2}) > -2$ are generally essential for the model.

Let us take the situation of using lidar data from LidarComplex to predict $a$ as an example to explain the process of predictor selection (see Fig. 10 (a) top and the first block in Table 5). Firstly, since $\log(\sigma_m^{-2})$ of $I_{T,l}$ and $T_l$ are much smaller than the others, it is not necessary to consider these two predictors, and the lower limit of $\log(\sigma_m^{-2})$ can be initially set to $-4$ (see Table 5: Case 1). Then, try to increase the lower limit of $\log(\sigma_m^{-2})$ step by step, e.g. first to $-2$ (see Table 5: Case 2) and then to 0 (see Table 5: Case 3), to further reduce the number of predictors. The resulting models are evaluated to determine whether it is appropriate to remove these predictors. For example, the comparison between Case 1 and Case 2 shows that removing $d$ almost does not affect the model performance in this situation, with $R^2$ decreasing only slightly from 0.70 to 0.69. However, further abandonment of $\Delta t_M$ significantly reduces the prediction accuracy, reducing $R^2$ more substantially from 0.69 in Case 2, to 0.59 in Case 3. Therefore, it is no longer proper to remove further predictors, and the predictor combination in Case 2 is recommended.

## 5.2 Discussion of Selected Predictors

Feature selection is not only a tool to select suitable predictors for a machine learning model, but also could shed some light on intrinsic relationships among data. Here are some discussions about the selected predictors, to provide some insights into possible correlations between wind evolution and these predictors.

***Selection between two related variables.*** In the preliminary training, two pairs of related variables are intentionally involved at the same time: $\sigma$ and $I_T$, $T$ and $L$. It is only necessary to select one of the two related variables (if determined to be relevant), because they can be converted into each other (given $U$). In terms of $\sigma$ and $I_T$, it is surprising to notice that the GPR models show a preference for $\sigma$ rather than $I_T$, although $I_T$ is more commonly used in data analysis and simulation in wind energy. The only exception is the situation of using sonic data from LidarComplex to predict $b$ (Cases 16–18). It is possible that GPR generally tends to select fundamental variables (directly calculated from measured data) instead of derived variables (calculated from other variables). However, the selection becomes complicated for $T$ and $L$. In some situations, $L$ is clearly more preferred, e.g. $\log(\sigma_m^{-2})$ of $L$ is obviously higher than $\log(\sigma_m^{-2})$ of $T$ in Fig. 10 (a) top and (b). In the other situations,

**Table 5.** Summary of the predictors selected according to different lower limits of $\log(\sigma_m^{-2})$ under different measurement campaigns, different data availability, and different targets. $R^2$ and RMSE are obtained from a 5-fold cross-validation of the model trained with the respective combination of predictors. The bold text indicates the recommended predictor combinations.

| Measurement | Target | Case | $\log(\sigma_m^{-2})$ | Predictors | RMSE | $R^2$ |
|---|---|---|---|---|---|---|
| LidarComplex lidar data | $a$ | 1 | $> -4$ | $U_\mathrm{l}, \sigma_\mathrm{l}, G_{1,\mathrm{l}}, G_{2,\mathrm{l}}, L_\mathrm{l}, \Delta t_\mathrm{M}, d$ | 0.39 | 0.70 |
| | | **2** | $> -2$ | $U_\mathrm{l}, \sigma_\mathrm{l}, G_{1,\mathrm{l}}, G_{2,\mathrm{l}}, L_\mathrm{l}, \Delta t_\mathrm{M}$ | 0.40 | 0.69 |
| | | 3 | $> 0$ | $U_\mathrm{l}, \sigma_\mathrm{l}, G_{1,\mathrm{l}}, G_{2,\mathrm{l}}, L_\mathrm{l}$ | 0.46 | 0.59 |
| | | 4 | - | $U_\mathrm{l}, \sigma_\mathrm{l}, L_\mathrm{l}, \Delta t_\mathrm{M}, d$ | 0.49 | 0.53 |
| | | 5 | - | $U_\mathrm{l}, \sigma_\mathrm{l}, G_{1,\mathrm{l}}, G_{2,\mathrm{l}}, L_\mathrm{l}, \Delta t_\mathrm{T}, d$ | 0.41 | 0.67 |
| | | **6** | - | $U_\mathrm{l}, \sigma_\mathrm{l}, G_{1,\mathrm{l}}, G_{2,\mathrm{l}}, L_\mathrm{l}, \Delta t_\mathrm{M}, d, \alpha$ | 0.35 | 0.76 |
| LidarComplex lidar data | $b$ | **7** | $> -2$ | $U_\mathrm{l}, \sigma_\mathrm{l}, G_{1,\mathrm{l}}, G_{2,\mathrm{l}}, L_\mathrm{l}, \Delta t_\mathrm{M}, d$ | 0.047 | 0.70 |
| | | 8 | $> -1$ | $U_\mathrm{l}, \sigma_\mathrm{l}, G_{1,\mathrm{l}}, G_{2,\mathrm{l}}, L_\mathrm{l}$ | 0.081 | 0.12 |
| | | 9 | - | $U_\mathrm{l}, \sigma_\mathrm{l}, L_\mathrm{l}, \Delta t_\mathrm{M}, d$ | 0.063 | 0.46 |
| | | 10 | - | $U_\mathrm{l}, \sigma_\mathrm{l}, G_{1,\mathrm{l}}, G_{2,\mathrm{l}}, L_\mathrm{l}, \Delta t_\mathrm{T}, d$ | 0.048 | 0.69 |
| | | **11** | - | $U_\mathrm{l}, \sigma_\mathrm{l}, G_{1,\mathrm{l}}, G_{2,\mathrm{l}}, L_\mathrm{l}, \Delta t_\mathrm{M}, d, \alpha$ | 0.038 | 0.81 |
| | | 12 | - | $U_\mathrm{l}, \sigma_\mathrm{l}, G_{1,\mathrm{l}}, G_{2,\mathrm{l}}, L_\mathrm{l}, \Delta t_\mathrm{M}, d, a$ | 0.044 | 0.74 |
| LidarComplex sonic data | $a$ | **13** | $> -4$ | $U_\mathrm{x,s}, G_{1,\mathrm{x,s}}, G_{2,\mathrm{x,s}}, L_\mathrm{x,s}, \sigma_\mathrm{y,s}, G_{1,\mathrm{y,s}}, G_{2,\mathrm{y,s}}$ $L_\mathrm{y,s}, \sigma_\mathrm{z,s}, G_{1,\mathrm{z,s}}, G_{2,\mathrm{z,s}}, \Delta t_\mathrm{M}, d, \alpha$ | 0.30 | 0.83 |
| | | 14 | $> -2$ | $U_\mathrm{x,s}, G_{2,\mathrm{x,s}}, \sigma_\mathrm{y,s}, G_{1,\mathrm{y,s}}, \sigma_\mathrm{z,s}, G_{2,\mathrm{z,s}}, \alpha$ | 0.39 | 0.70 |
| | | **15** | $> 0$ | $U_\mathrm{x,s}, \sigma_\mathrm{z,s}, \alpha$ | 0.39 | 0.70 |
| LidarComplex sonic data | $b$ | 16 | $> -3$ | $U_\mathrm{x,s}, I_{\mathrm{T},x,s}, G_{1,\mathrm{x,s}}, G_{2,\mathrm{x,s}}, L_\mathrm{x,s}, G_{1,\mathrm{y,s}}, G_{2,\mathrm{y,s}}$ $G_{1,\mathrm{z,s}}, \Delta t_\mathrm{M}, d, \alpha, a$ | 0.039 | 0.80 |
| | | **17** | $> -2$ | $U_\mathrm{x,s}, I_{\mathrm{T},x,s}, G_{1,\mathrm{x,s}}, G_{2,\mathrm{x,s}}, G_{1,\mathrm{y,s}}, G_{1,\mathrm{z,s}}, d$ | 0.040 | 0.78 |
| | | 18 | $> -1$ | $I_{\mathrm{T},x,s}, G_{2,\mathrm{x,s}}, G_{1,\mathrm{y,s}}$ | 0.081 | 0.13 |
| ParkCast lidar data | $a$ | **19** | $> -1$ | $U_\mathrm{l}, \sigma_\mathrm{l}, G_{1,\mathrm{l}}, G_{2,\mathrm{l}}, L_\mathrm{l}, \Delta t_\mathrm{M}, d$ | 0.53 | 0.81 |
| | | 20 | $> 0$ | $U_\mathrm{l}, \sigma_\mathrm{l}, G_{1,\mathrm{l}}, G_{2,\mathrm{l}}, L_\mathrm{l}, \Delta t_\mathrm{M}$ | 0.67 | 0.69 |
| ParkCast lidar data | $b$ | **21** | $> -1$ | $U_\mathrm{l}, \sigma_\mathrm{l}, G_{1,\mathrm{l}}, G_{2,\mathrm{l}}, L_\mathrm{l}, \Delta t_\mathrm{M}, d$ | 0.11 | 0.67 |
| | | 22 | $> 0$ | $U_\mathrm{l}, \sigma_\mathrm{l}, G_{1,\mathrm{l}}, G_{2,\mathrm{l}}$ | 0.16 | 0.31 |

Notes: Cases 4–6 and Cases 9–12 are selected for comparison with Case 1 and Case 7 for different purposes, respectively. Case 4 and Case 9: to examine the effect of introducing $G_1$ and $G_2$ as predictors. Case 5 and Case 10: to examine the different effects of $\Delta t_\mathrm{M}$ and $\Delta t_\mathrm{T}$. Case 6 and Case 11: to examine the effect of having $\alpha$ available. Case 12: to examine the effect of introducing $a$ as a predictor for $b$.

$\log(\sigma_m^{-2})$ of $L$ and $\log(\sigma_m^{-2})$ of $T$ show similar values. For consistency, we decided to select $L$ for all cases whenever $L$ is determined to be relevant.

***Introducing higher-order wind statistics as predictors.*** So far, skewness $G_1$ and kurtosis $G_2$ of wind speed have not been considered in wind evolution research. However, it is worth noting that both are selected as predictors in all cases except Case 15, despite different measurement sites and devices. Case 4 and Case 9 are aimed at examining the effects of $G_1$ and $G_2$ on the prediction of $a$ and $b$, respectively, with $G_1$ and $G_2$ removed in comparison to Cases 1 and 7. Case 4 and Case 9 show much worse prediction accuracy, with $R^2 = 0.53$ in Case 4 compared to $R^2 = 0.70$ in Case 1 and $R^2 = 0.46$ in Case 9 compared to

$R^2 = 0.70$ in Case 7. This comparison confirms that $G_1$ and $G_2$ are essential for predicting wind evolution when using lidar data, and introducing $G_1$ and $G_2$ as predictors can significantly improve the models, despite uncertainties contained in their estimated values from measured data (see Sect. 2.5). This implies that $G_1$ and $G_2$ might contain additional information which could distinguish different states of turbulence given a particular mean wind speed and turbulence intensity, and this 'different state' might be relevant to wind evolution.

***Different approximations of travel time.*** $\Delta t_{\mathrm{M}}$ and $\Delta t_{\mathrm{T}}$ are two different approximations of travel time. Although $\Delta t_{\mathrm{M}}$ is expected to be more predictive than $\Delta t_{\mathrm{T}}$, $\Delta t_{\mathrm{T}}$ is still involved in the model training because, in application, it is easier to calculate $\Delta t_{\mathrm{T}}$ than $\Delta t_{\mathrm{M}}$. Cases 5 and 10 are selected to compare with Cases 1 and 7, respectively, to examine the different effects of $\Delta t_{\mathrm{T}}$ and $\Delta t_{\mathrm{M}}$ on the GPR models. The respective values of $R^2$ show that replacing $\Delta t_{\mathrm{M}}$ with $\Delta t_{\mathrm{T}}$ only slightly decreases the prediction accuracy. Therefore, for a simpler calculation of travel time, $\Delta t_{\mathrm{T}}$ can be used as a predictor instead.

***Effect of misalignment angle.*** As discussed in Sect. 2.5, misalignment angle $\alpha$ is supposed to be an important predictor for the prediction of the horizontal coherence. In Cases 13–15, where sonic data from LidarComplex is used to predict $a$, $\alpha$ shows a high relevance with $\log(\sigma_m^{-2}) > 0$. However, for the prediction of $b$ using sonic data (Cases 16–18), removing $\alpha$ from predictors does not influence the prediction accuracy much, especially when comparing Case 16 and Case 17, with $R^2 = 0.80$ and $R^2 = 0.78$, respectively. These results indicate that $\alpha$ is essential for the prediction of $a$ but not relevant for predicting $b$.

In addition, $\alpha$ is introduced in the prediction using lidar data (Case 6 and Case 11) as well to examine its effect, although $\alpha$ is actually not available when only using a lidar in staring mode. As mentioned in Sect. 3.1, $\alpha$ is approximated by the deviation between the yaw position of the turbine and mean wind direction taken on the meteorological mast. Cases 6 and 11 both show better prediction accuracy than Cases 1 and 7, with $R^2 = 0.76$ and $R^2 = 0.81$, respectively, despite the uncertainties in the approximation of $\alpha$. This means that if $\alpha$ were available, the prediction accuracy of the models trained with lidar data could be

further improved. As mentioned earlier, $\alpha$ could be made available e.g. by deploying a multi-beam lidar.

***Introducing one of the targets as a predictor for the other.*** According to the wind evolution model (Eq. (8)), $a$ and $b$ jointly determine the shape and the position of the modeled coherence, and thus they have a certain correlation with each other. Introducing one of them as a predictor for the other may improve its prediction accuracy. Case 12, with $R^2 = 0.74$, compared to $R^2 = 0.70$ in Case 7, confirms that introducing $a$ as a predictor for $b$ can help with the prediction of $b$. This means it could

be a good idea to predict the wind evolution model parameters successively rather than in parallel. This concept is not yet fully studied in this work, and thus Case 12 is not presented as a recommendation. To prove its applicability, it is necessary to investigate which wind evolution model parameter should be first predicted, and how the prediction uncertainty in the first parameter would propagate to the second.

*Prediction using sonic data.* Additional research on using sonic data as predictors aims to provide some insights into
whether it is worth involving sonic data in wind evolution prediction when available. When comparing the model performance
of using lidar data and sonic data from LidarComplex, Case 13 — the best case of using sonic data to predict $a$ — shows
a higher prediction accuracy ($R^2 = 0.83$ ) than Case 6 — the best case of using lidar data given $\alpha$ available ($R^2 = 0.76$).
However, Case 13 needs many more predictors than Case 6, whereas Case 14 and Case 15 , with fewer predictors, do not show
any advantage in prediction accuracy. For predicting $b$, Case 16 — the best case of using sonic data ($R^2 = 0.80$), does not
outperform Case 11, the best case of using lidar data given $\alpha$ available ($R^2 = 0.81$). It must be emphasized that the ultrasonic
anemometer is installed on a meteorological mast located $295\,\mathrm{m}$ away from the lidar. There must be a deviation between the
sonic data and the true values in the wind field where the coherence is estimated, which reduces the prediction accuracy when
using sonic data. Figure 11 illustrates a comparison between the model performance of the recommended cases of using lidar
data and sonic data.

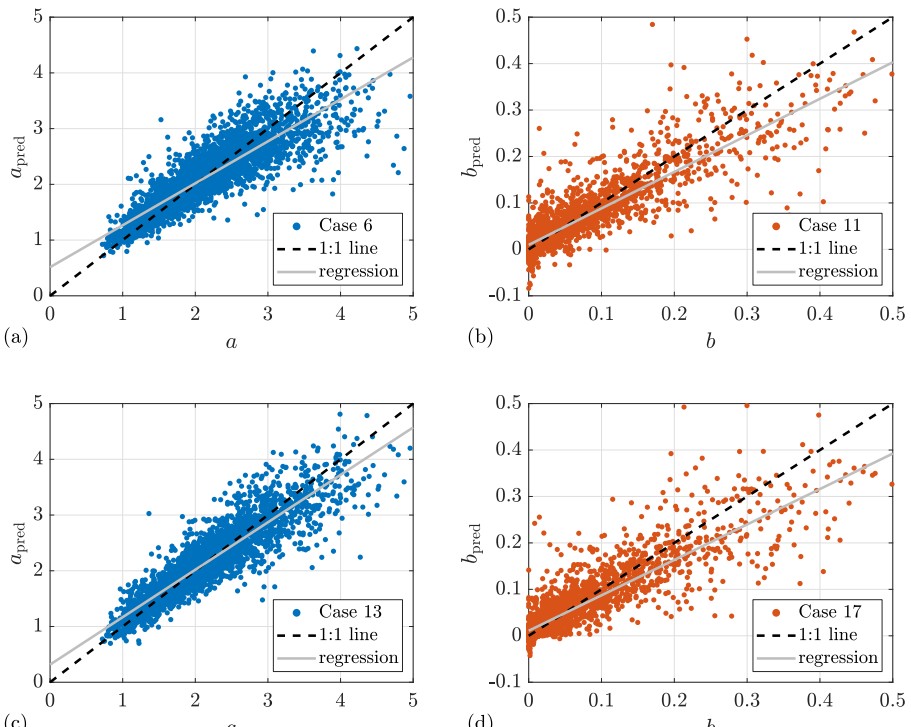

**Figure 11.** Comparison of prediction performance of models using lidar data and sonic data from LidarComplex. (a) and (b): lidar data. (c)
and (d): sonic data. The subscript 'pred' indicates the predicted values.

Interestingly, Case 15 can achieve the same predictive accuracy as Case 1, with only three predictors: mean wind speed $U_{\mathrm{x,s}}$,
standard deviation of the vertical wind component $\sigma_{\mathrm{z,s}}$, and the misalignment angle $\alpha$. In fact, $\sigma_{\mathrm{z,s}}$ is determined to be the most
important predictor by the ARD-SE kernel, having the maximum value of $\log(\sigma_m^{-2})$. This might imply a possible correlation
between wind evolution and vertical convection.

*Influence of atmospheric stability.* We initially intended to study the influence of atmospheric stability using a dimensionless
height $\zeta$ as the stability parameter (see Sect. 2.5). However, very surprisingly, $\zeta$ is not selected as a relevant predictor in any
cases, and $\log(\sigma_m^{-2})$ is quite small compared to the others (see Fig. 10 (c)). In the end, we found that the stability happens
to be mostly neutral during the chosen measurement in LidarComplex. This could be the reason for $\zeta$ not being selected as a
predictor. Therefore, it is not possible to analyze the influence of atmospheric stability on wind evolution in this study.

## 5.3 Model Evaluation

As shown in Table 5, $R^2$ of all recommended cases range from 0.67 to 0.83. These results are much better than that of the
preliminary study (Chen, 2019); in particular, the prediction accuracy of the offset parameter $b$ has been significantly improved.
This is mainly owing to the use of the ARD-SE kernel, which can help to select predictors reasonably and give different weights
to predictors according to their relevant importance for the prediction, whereas kernel functions with a common length scale
for predictors were applied in the preliminary study.

The prediction errors of $a$ and $b$ are quantified with the respective RMSE between their predicted and observed values. But
in fact, the shape and position of the predicted coherence determined by both parameters together is the final prediction goal.
And the corresponding prediction errors will eventually appear as the deviation between the predicted curve and its estimated
curve due to the prediction errors of $a$ and $b$.

To intuitively display how the prediction errors affect the shape and the position of the predicted coherence in the frequency
domain, Fig. 12 shows the predicted coherence and the corresponding $95\,\%$ confidence interval for the example case from
LidarComplex. For the example prediction with lidar data in Fig. 12 (a), the prediction of $a$ and $b$ is made by the GPR models
in Cases 6 and 11, respectively. And for the example prediction with sonic data in Fig. 12 (b), the prediction of $a$ and $b$ is
made by the GPR models in Cases 13 and 17, respectively. The predicted coherence and the $95\,\%$ confidence interval are
reconstructed by putting the predicted values of $a$ and $b$ and their lower and upper bounds of the $95\,\%$ confidence interval into
the wind evolution model (Eq. (8)). It can be observed that the prediction is very good for this example because the predicted
coherence is almost overlapped with the one estimated from the measured data, and the $95\,\%$ confidence interval is quite
narrow.

To show the prediction errors in a more general sense, the RMSE interval is additionally indicated as shaded areas in Fig.
12. The lower and upper bounds of the RMSE interval are determined with

$$\gamma^2_{\mathrm{model,lb}}(f_{\mathrm{dless}}) = \exp\left[-\sqrt{(a_{\mathrm{pred}}+\Delta a)^2 \cdot f^2_{\mathrm{dless}} + (b_{\mathrm{pred}}+\Delta b)^2}\right] \tag{31}$$

and

$$\gamma^2_{\mathrm{model,ub}}(f_{\mathrm{dless}}) = \exp\left[-\sqrt{(a_{\mathrm{pred}}-\Delta a)^2 \cdot f^2_{\mathrm{dless}} + (b_{\mathrm{pred}}-\Delta b)^2}\right], \tag{32}$$

respectively, where $a_{\mathrm{pred}}$ and $b_{\mathrm{pred}}$ are the predicted values of $a$ and $b$, and $\Delta a$ and $\Delta b$ are the respective RMSE. The narrow
RMSE interval shows that the GPR models perform overall well in the prediction of wind evolution.

Moreover, it is important to check if the prediction errors of the models are relevant to the values of the predictors. Taking
the models trained with the lidar data from LidarComplex (Case 6 and Case 11) as an example, Fig. 13–16 show the box plots

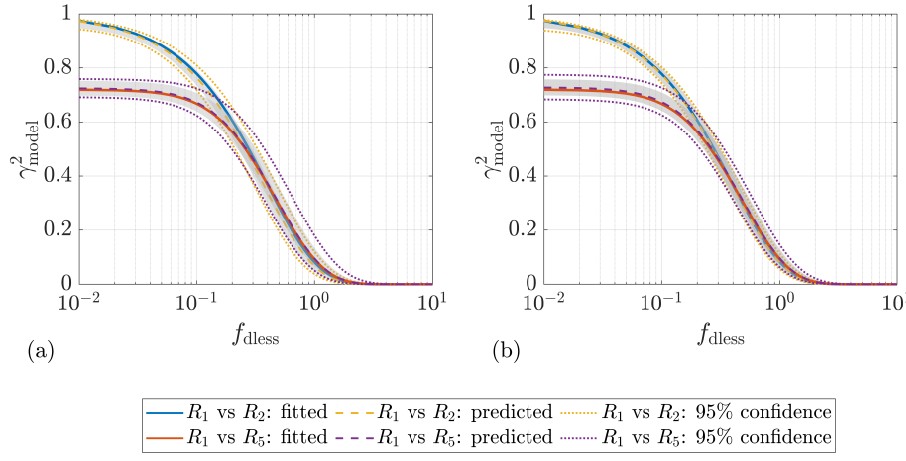

**Figure 12.** Example predicted coherence with 95% confidence interval for two different measurement separations of LidarComplex. (a) Prediction with lidar data: $a$ – Case 6, $b$ – Case 11. (b) Prediction with sonic data: $a$ – Case 13, $b$ – Case 17. The shaded areas indicate the RMSE interval. The input predictor data and the estimated coherence are from the case study example of LidarComplex: 07 Dec. 2013, 12:00-12:30. The mean lidar wind speed $U_l$ ranges from $7.3\,\mathrm{m\,s^{-1}}$ to $7.7\,\mathrm{m\,s^{-1}}$ and the lidar measured turbulence intensity $I_{T,l}$ ranges from 0.10 to 0.12, for different range gates.

of the prediction errors, defined as the deviation between the predicted and the observed values of targets, with respect to the values of the predictors. The histograms of the predictor values are plotted below the box plots correspondingly. The $x$-axes of the box plots correspond the upper bound of the respective bin in the histograms. For example, in Fig. 13 (a), the first box labelled with '4' means it is plotted with the prediction errors of the samples attributed to the mean wind speed range of $3\,\mathrm{m\,s^{-1}}$ to $4\,\mathrm{m\,s^{-1}}$. To avoid accidental conclusions, there is a minimum sample size requirement of 50 for the box plots.

The box plots indicate data within the first and the third quartiles (i.e. $25^{\mathrm{th}}$ and $75^{\mathrm{th}}$ percentile) and represent the main part of the data, whereas whiskers show the tails of the distributions of the data indicating extreme values. In Fig. 13–16, it can be observed that the boxes of the prediction errors of $a$ and $b$ are all quite narrow and centered around 0, indicating small prediction errors for the majority of samples. That the boxes are centered around 0, as well as the median and mean values (indicated as red lines and yellow crosses, respectively), means that there is no systematic error with respect to predictor values. In the box plots for the prediction errors of $a$, the ranges of boxes and whiskers do not show obvious relevance to predictor values except for small travel time and measurement separation. The large range of the box and whiskers of the first box in Fig. 15 (b) and that of the first box in Fig. 16 (a)) implies that the prediction of $a$ is likely more uncertain for small travel time and measurement separation (both are related to some extent). The ranges of boxes and whiskers of the prediction errors of $b$ show some relevance to the values of standard deviation, skewness, travel time, and measurement separation. In Fig. 13 (b), a clear trend can be observed, that the ranges of the boxes and whiskers decrease with the values of standard deviation, indicating that the prediction of $b$ might be better for high turbulence. A similar trend can be observed in Fig. 14 (a), meaning that the prediction of $b$ might be better under the circumstance of negative skewness (longer left tail) than that of positive skewness

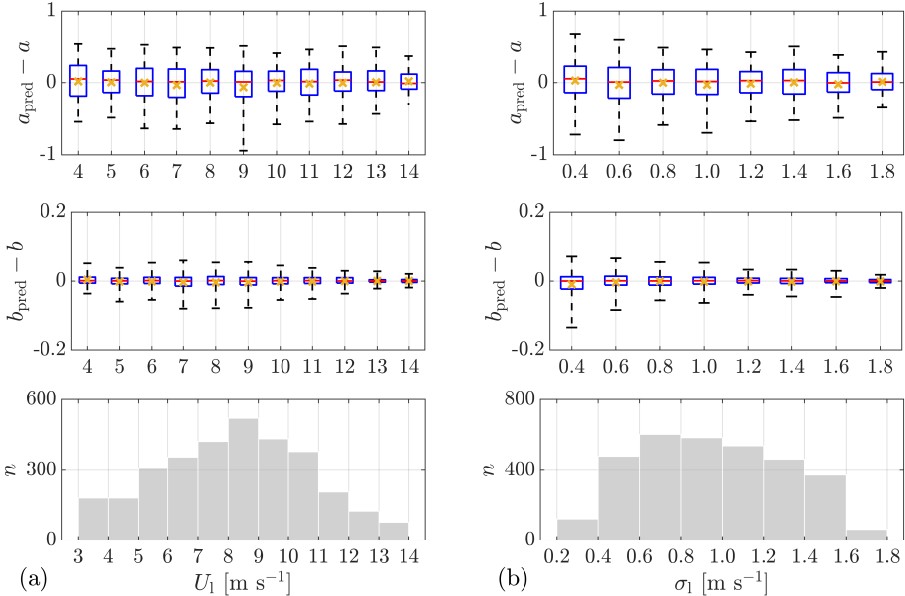

**Figure 13.** Prediction errors of $a$ (Case 6) and $b$ (Case 11) from LidarComplex with respect to the values of predictors. (a) Lidar measured mean wind speed $U_1$. (b) Standard deviation of lidar measured wind speed $\sigma_1$. $n$ is sample size. The bottom and top of the boxes indicate the first and the third quartiles, i.e. $25^{\text{th}}$ and $75^{\text{th}}$ percentile, respectively. The lower and upper whiskers show $5^{\text{th}}$ and $95^{\text{th}}$ percentiles. The red line and the yellow cross in the middle indicate the median and mean value, respectively.

(longer right tail). In Fig. 15 (b) and Fig. 16 (a), the ranges of boxes and whiskers get larger with travel time and measurement separation, implying that the prediction errors of $b$ increase with travel time and measurement separation.

It is worth emphasizing that the performance of any regression model can be only as good as the quality of the training data. No choice of regression model can eliminate noise from the training data. And the noisier the training data is, the more uncertainties the prediction of the regression model will contain. A good data source is always essential for training a good

regression model.

## 6 Conclusions and Outlook

This paper aims to investigate the potential of Gaussian process regression (GPR) in the parameterization of wind evolution. This research has been motivated by the need of lidar-assisted wind turbine control for accurate models to predict wind evolution, in order to avoid harmful and unnecessary control actions. In addition, the commonly used 3-dimensional stochastic wind

field simulation method can be extended to 4-dimensional by integrating wind evolution, to provide a more realistic simulation environment for this control concept.

In this research, data from two nacelle-mounted lidars in both onshore and offshore locations were used to estimate wind evolution. The estimated wind evolution was fitted to a two-parameter wind evolution model, modified from a model suggested

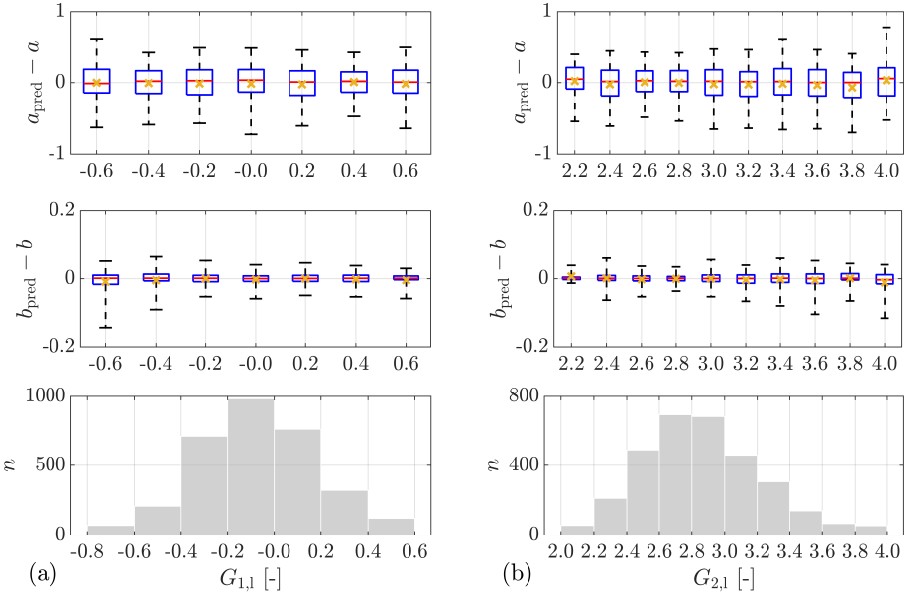

**Figure 14.** Prediction errors of $a$ (Case 6) and $b$ (Case 11) from LidarComplex with respect to the values of predictors. (a) Skewness of lidar measured wind speed $G_{1,1}$. (b) Kurtosis of lidar measured wind speed $G_{2,1}$. $n$ is sample size. The bottom and top of the boxes indicate the first and the third quartiles, i.e. 25th and 75th percentile, respectively. The lower and upper whiskers show 5th and 95th percentiles. The red line and the yellow cross in the middle indicate the median and mean value, respectively.

in the literature. To shed light on some characteristics of wind evolution, a statistical analysis was done for the wind evolution
model parameters.

In the statistical analysis, the distributions of the wind evolution model parameters of both measurements show some common characteristics, despite different wind-field–related variables and settings of the measurements. The value ranges of both wind evolution parameters $a$ (i.e. the decay parameter) and $b$ (i.e. the offset parameter) are very similar in both measurements. The distributions of $a$ and the $b$ seem to follow an inverse Gaussian distribution and a Gamma distribution, respectively. The
765 fitted parameters of the probability density functions are different in both measurements. We hypothesize that the parameters of the probability density functions might depend on the terrain type. Moreover, a strong dependence of wind evolution model parameters was observed on measurement separations. The decay parameter $a$ shows a decreasing trend with increasing measurement separation, while the offset parameter $b$ shows an increasing trend with increasing measurement separation.

An investigation was done to explore the potential of using GPR to achieve parameterization models for wind evolution.
GPR models were trained with the wind evolution model parameters (i.e. targets) and some wind-field–related variables (i.e. predictors) acquired from the lidars and a meteorological mast. The automatic relevance determination squared exponential kernel was applied to evaluate the relative importance of different predictors and to select the essential predictors for the models under different data availabilities. The performance of the GPR models was evaluated with the coefficient of determination $R^2$

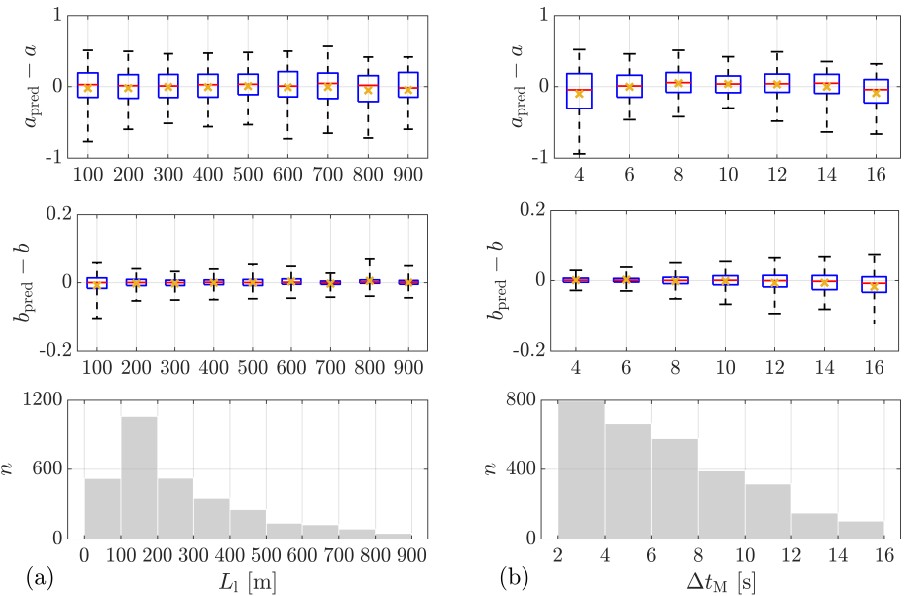

**Figure 15.** Prediction errors of $a$ (Case 6) and $b$ (Case 11) from LidarComplex with respect to the values of predictors. (a) Integral length scale of lidar measured wind speed $L_l$. (b) Time lag determined by the peak of maximum cross-correlation $\Delta t_M$. $n$ is sample size. The bottom and top of the boxes indicate the first and the third quartiles, i.e. $25^{th}$ and $75^{th}$ percentile, respectively. The lower and upper whiskers show $5^{th}$ and $95^{th}$ percentiles. The red line and the yellow cross in the middle indicate the median and mean value, respectively.

and root-mean-squared error (RMSE) using a 5-fold cross-validation. The $R^2$ of the models in the recommended cases for both targets, under different measurement campaigns and different data availabilities, range from $0.67$ to $0.83$.

A comparison between the models trained with different predictor combinations provides some interesting insights: 1) GPR models show preference to a fundamental variable than a derived variable when selecting between two related variables. 2) Introducing higher-order wind statistics (i.e. skewness and kurtosis) as predictors can improve the models. 3) When using travel time as a predictor, the approximation determined with the maximum cross-correlation is slightly preferred than Taylor's translation hypothesis, but the latter could still be an option for the sake of simplification. 4) Introducing one of the targets as a predictor for the other can also improve the models, but further research needs to be done to understand the propagation of the uncertainties introduced by the first predicted target. 5) Considering the misalignment angle as a predictor can properly account for its influence on the horizontal coherence. 6) Prediction using sonic data (not measured nearby) does not show any advantages given that it requires many more predictors to exceed the prediction using lidar data.

The predicted coherence is obtained by putting the two predicted parameters into the wind evolution model. To intuitively display how the prediction errors of $a$ and $b$ affect the shape and the position of the predicted coherence in the frequency domain, the predicted coherence and its $95\%$ confidence interval was visualized for a representative case-study example. The predicted coherence matches the coherence estimated from data very well, and the $95\%$ confidence interval is relatively narrow. In addition, the RMSE interval was also demonstrated to show the impact of the RMSE of $a$ and $b$ in a more general sense. The

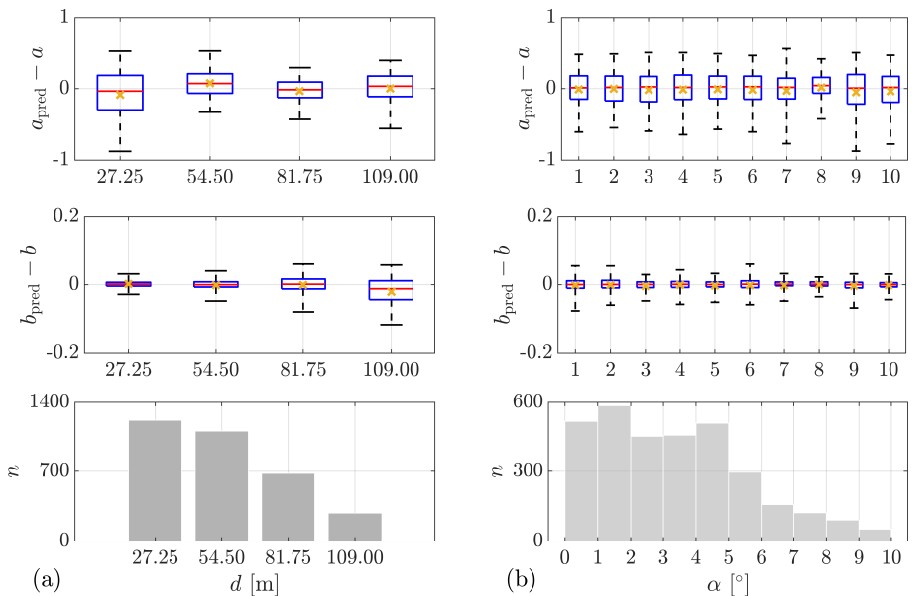

**Figure 16.** Prediction errors of $a$ (Case 6) and $b$ (Case 11) from LidarComplex with respect to the values of predictors. (a) Measurement separation $d$. (b) Misalignment angle of wind direction and lidar measurement $\alpha$. $n$ is sample size. The bottom and top of the boxes indicate the first and the third quartiles, i.e. $25^{\text{th}}$ and $75^{\text{th}}$ percentile, respectively. The lower and upper whiskers show $5^{\text{th}}$ and $95^{\text{th}}$ percentiles. The red line and the yellow cross in the middle indicate the median and mean value, respectively.

RMSE interval turns out to be quite narrow, indicating an overall goo model performance. Furthermore, the prediction errors of $a$ and $b$ were analyzed with respect to the values of each predictor, shown as box plots. The results show that, for both $a$ and $b$, there is no systematic error with respect to predictor values. The prediction of $a$ seems to be less accurate for small travel time and measurement separation. The prediction errors of $b$ show some relevance to the values of standard deviation and skewness of wind speed, travel time, and measurement separation.

There is still space to improve the performance of the parameterization model. Since the performance of any regression model can be only as good as the quality of the training data, reducing the uncertainty in the training data or increasing the data amount could improve the model performance. For example, methods to improve the estimation of the coherence and the wind statistics from lidar data are desirable. Moreover, the predictors discussed above do not cover all possibilities. Introducing new proper predictors could hence also improve the model performance. In fact, the model concept is very flexible. Any improvement of any part of the workflow can be easily integrated.

In the future, besides the ideas mentioned above, it would be interesting to involve more measurement data, especially from different terrain types, to further investigate whether the wind evolution characteristics found here occur commonly, and what physical principles stand behind them. Another question that needs answering is whether it is possible to achieve a generally applicable parameterization model, and how. Moreover, considering that the computational time of the model training could be an important issue for some applications, e.g. real-time model training, it is worth comparing GPR with some

alternative algorithms to develop insight into the trade-off between computation time and the prediction accuracy. Furthermore, considering the application of the parameterization model using real-time measurement data as predictors, an additional model will be needed to determine whether the current data meets the quality requirements to be input into the parameterization model.

Last but not least, as mentioned above, the our model concept is very flexible and its methodology can be applied in different situations. For example, for other lidar trajectories or even other measurement devices, the model concept can be modified by replacing the coherence estimation method. The wind evolution model and the regression model can also be changed. Basically, one can achieve a parameterization model to meet various specific requirements by following the concept and the methodology presented in this paper.

**Appendix A**

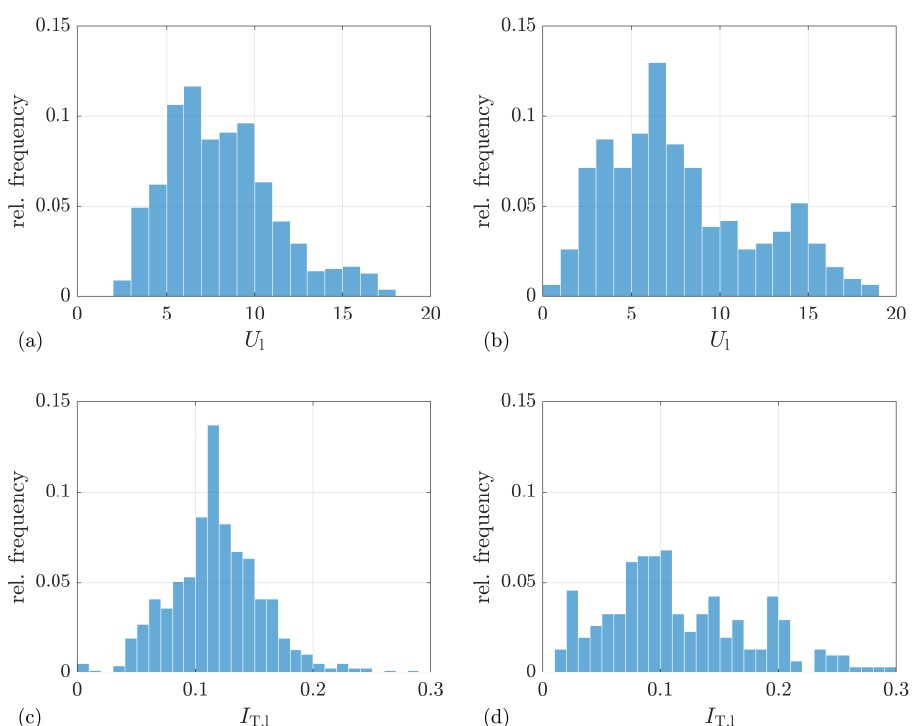

**Figure A1.** Distribution of lidar measured mean wind speed and turbulence intensity for the selected period. (a) and (c) are from LidarComplex, measurement point at $163.5\,\mathrm{m}$; (b) and (d) are from ParkCast, measurement point at $150\,\mathrm{m}$.

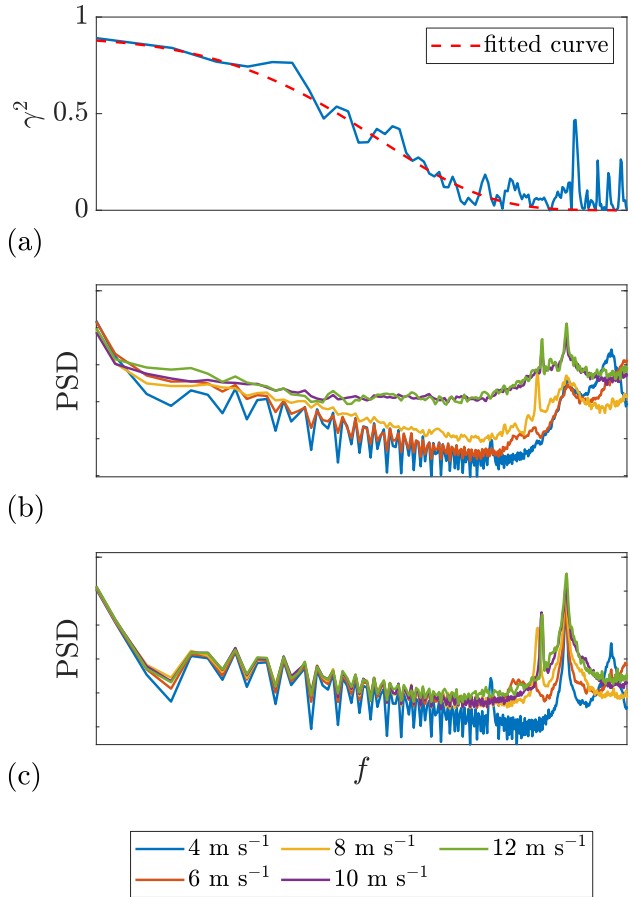

**Figure A2.** (a) Example coherence: $U_1 = 11.7\,\mathrm{m\,s^{-1}}$, $d = 81.75\,\mathrm{m}$, $R^2 = 0.95$. (b) and (c) are PSDs of the fore-aft and in-plane tower top acceleration, respectively. The $x$-axis is logarithmic. Date and time: 07 Dec. 2013, 12:00 - 12:30. Data Source: LidarComplex. Because of data protection it is not allowed to show any values concerning the turbine properties.

*Author contributions.* YC conceived the concept, developed the model codes, processed the data, created the figures, conducted the analysis, and prepared the manuscript. DS and PWC provided general guidance and essential suggestions throughout the process. PWC supervised the research.

*Competing interests.* The authors declare that they have no conflict of interest.

*Acknowledgements.* This research is carried out within the framework of Joint Graduate Research Training Group, Windy Cities supported by the State Ministry of Baden-Wuerttemberg for Sciences, Research and Arts. Some data used in this work was generated in the projects

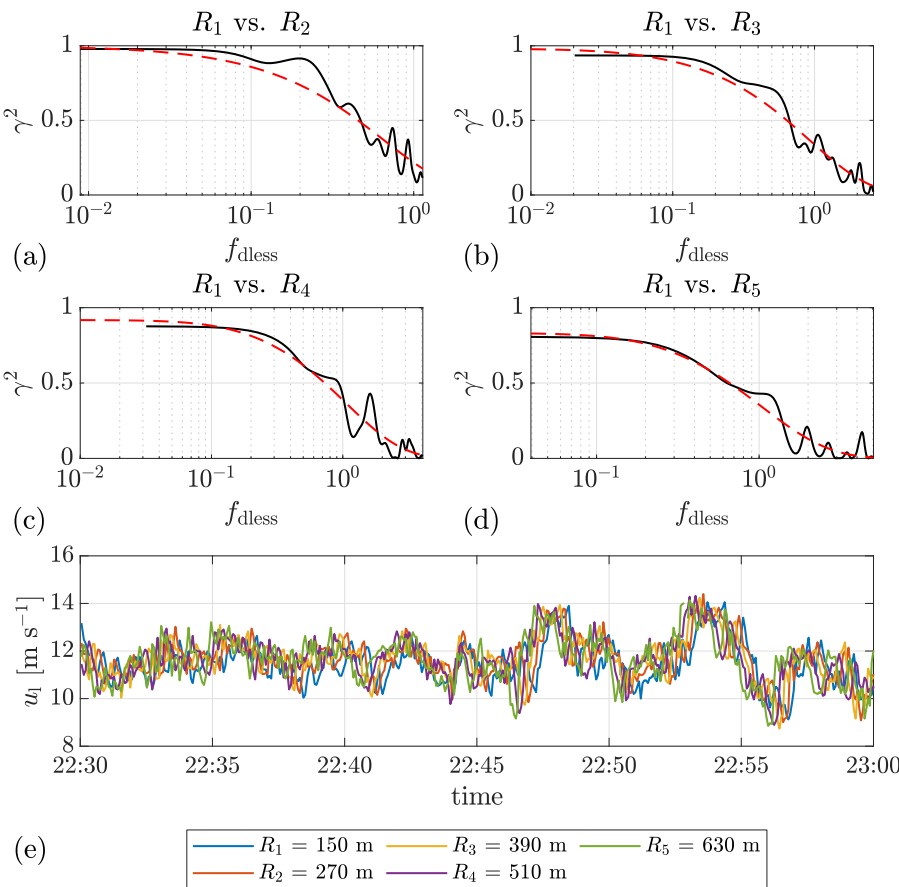

**Figure A3.** (a)–(d) Example plots of the estimated coherence between the selected range gates $R$ and the corresponding fitted curves. The corresponding measurement distances are $120\,\mathrm{m}$, $240\,\mathrm{m}$, $360\,\mathrm{m}$, and $480\,\mathrm{m}$, respectively. (e) Time series of the lidar wind speed. The mean lidar wind speed is $11.6\,\mathrm{m\,s^{-1}}$. Date: 12 June 2019. Data source: ParkCast.

LidarComplex (code number: 0325519A) and ParkCast (code number: 0324330A) which are funded by the German Federal Ministry for Economic Affairs and Energy (BMWi). This publication is supported by the Open Access Publications funding of University of Stuttgart.

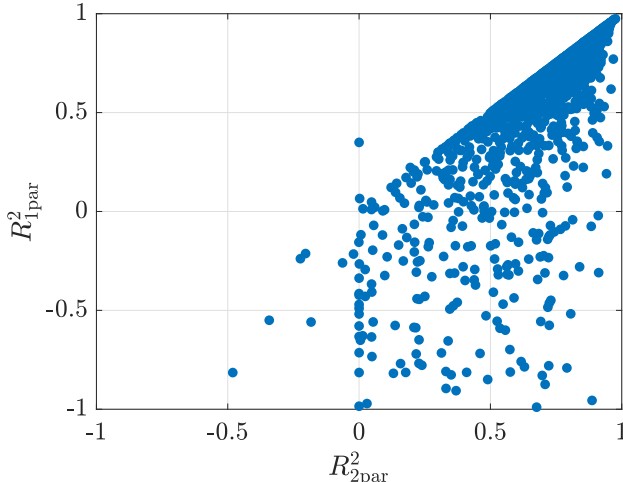

**Figure A4.** Comparison of the fitting quality ($R^2$) of the two-parameter wind evolution model (Eq. (8)) and that of the one-parameter wind evolution model (Eq. (5)). The subscripts '1par' and '2par' indicate one-parameter and two-parameter, respectively.

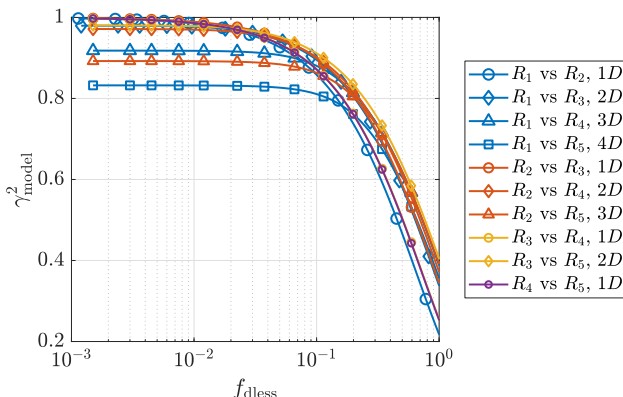

**Figure A5.** Fitted curves of the estimated coherence between the lidar wind speeds measured at different range gates. The range gate $R_1$ to $R_5$ are located at $150\,\mathrm{m}$, $270\,\mathrm{m}$, $390\,\mathrm{m}$, $510\,\mathrm{m}$, and $630\,\mathrm{m}$, respectively. $1D = 120\,\mathrm{m}$. The mean lidar wind speed $U_l = 11.6\,\mathrm{m\,s^{-1}}$. Date and time: 12 June 2019, 22:30-23:00. Data source: ParkCast.

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
