# Peer review of "Parameterization of Wind Evolution using Lidar"

_Wind Energy Science, 2020_

## Referee Comment (RC1) · Felix Kelberlau (Referee) · 19 Mar 2020

"General comments"

Chen et al. develop a method to predict the coherence of horizontal wind velocity fluctuations for mostly longitudinal separations. Their predictions are based on first to fourth order wind speed statistics that can be calculated from either nacelle-mounted lidar or mast-based in-situ anemometry. They use data from two measurement campaigns to test their approach and find good results that are especially relevant for lidar-assisted wind turbine control. The work lies therefore well in the scope of WES and is of broad international interest. The paper builds up on an existing wind evolution model and presents a novel approach to parameterise its two coefficients by means of machine learning. The manuscript explains the study thoroughly and reproducibly, presents all relevant results and discusses them critically.

[Figure]

Section 2.6 "Gaussian Process Regression" lies outside my field of expertise and I can therefore not evaluate if the chosen model is suitable for the task of parametrization the wind evolution model. The manuscript is overall understandable but would benefit greatly from being proofread by a native English speaker or similarly qualified person before publication. I recommend reconsideration for publication after major revisions.

"Specific comments"

l.2: I assume you mean "the mean flow" (also l.13 and all other occurrences).

l. 50: The introduction would benefit from references to research that support Taylor's frozen turbulence for very large turbulent structures but limit its applicability for long separation distances or small scale turbulence such as Willis and Deardorff (1976), Schlipf et al. (2010), and Kelberlau and Mann (2019).

l. 59: "the vertical intercept" It would be better to describe the second parameter without referring to the coherence-frequency plot that is not yet introduced here.

l. 61: "Mann spectral velocity tensor" Mann (1994) should be cited here.

l. 68: "If any data... is also available..." Please mention which data is available or would be of interest.

l. 94: Please introduce this travel time as a function of the mean wind speed here.

l. 97: Please explain why "it is not possible to predict every point of the coherence curve". For my understanding, the coherence curve is visible on a plot like in Fig. 1. Do you refer to not having not enough data to smoothen the curve or not having data for all separation distances?

l. 100: Do you mean "...according to measured wind velocity time series by a parame- terisation model"?

Fig. 1 and Fig. 2: In general, it is good to visualize the workflow like done here. But both figures show overlapping information and I recommend to merge them into one

figure. The numbering used in Fig. 2 with an explanation in the text and caption(!) is more informative than the keywords currently used in Fig. 1. A figure and its caption should be self explanatory whenever possible. Please try to improve the text l. 98-106 for better understanding.

l. 120: Please describe which frequency you are referring to. Probably the frequency of the horizontal wind velocity fluctuations. Maybe also introduce the wavenumber k here that is used as a measure of eddy size in many other publications.

Eq.(5): It is not clear where (5) comes from. If you do not want to include the complete deduction, I suggest to give a reference that shows it and uses the same form of the equation. In Simley and Pao (2015), a and b are defined a bit differently, I think.

l. 147: You should include the weighting here: e.g. "...but the weighted average of the wind speeds within the measurement volume"

l. 148: This is a bit ambiguous because spatial averaging does also refer to combining data from different measurement volumes in different lidar beam directions. Better write: "so-called line-of-sight averaging effect of lidar". (also l. 156)

l. 151: Please refer to more fundamental work (Nyquist-Shannon sampling theorem).

l. 152: You should mention the sampling rate of the lidars here (not only in the table) and compare it with the frequency of the eddies that you want to detect.

l. 158: The line-of-sight weighting of a pulsed lidar is usually approximated by a triangular function as in e.g. (Sathe and Mann (2012)) which is a $sinc^2$ function in the frequency domain.

l. 184: It should be considered that w(x) is approximately 0 for fluctuations that occur with a wavelength of twice the length of the illuminated section of the lidar beam (or length of the range gate). In this case the measurement signal would be determined by noise only. I suggest to estimate a range of critical frequencies based on the length of the range gates. This range of critical frequencies should be considered in the further

analysis, if it is relevant for the results. Your derivation assumes furthermore that the weighting function is identical for all range gates. This is only true if the laser beam is well collimated. Is this the case for the lidar devices used in this study?

l. 199: You could mention that a lidar with additional beams would help here and could also be used to avoid yaw-misalignment.

l. 208: What is the expected order of magnitude for the misalignment angle? How much "decorrelation" do you expect from a turbulence model (e.g. Mann (1994)) due to the resulting lateral separation? Can you quantify the order of magnitude of the resulting error approximately?

l. 213: Please always write which variable you are referring to when you mention standard deviation sigma.

l. 236: It would be good to introduce the variable alpha already in 2.4 (l. 199 and Fig. 4)if you refer to it here.

l. 263-266: This sentence is very long and difficult to understand.

l. 332-343: Please reassess which information should be given here: I miss: the measurement height of the lidar, length of each range gate, measurement distances... Some of these values are given in Table 2 but should also appear here. The information about the coordinate system will not be used again later in the text and do not need to be given at all then.

l. 342 and 350: Main wind direction refers usually to the direction from where the wind blows most frequently. Better write mean wind direction.

l. 389: I suggest a similar filtering against the line-of-sight averaging. See comment for l.184. Probably it is not worth it to re-run the computations. But check in the coherence plots, if the frequency range is relevant and if you see a random increase in coherence in it.

l. 397: Why do you not also filter the lowest percentile?

Fig. 6: Subfigure (e) would benefit from a zoom into just some few minutes of data with thin plot lines to show all velocities clearly and not on top of each other.

l. 445: That makes the difference caused by the reduced sampling rate in Fig. 7 (b) even more interesting. Earlier you write that "As long as the sampling rate of lidar is sufficiently high to acquire a complete coherence curve, it will not have a noticeable effect on the study of the coherence." but here you find that the influence on parameter b is big (logarithmic y-axis). What could be the reason? $f_s$=1/3Hz means, at least at high wind speeds, that it is difficult to measure exactly after the eddy travel time passed. That means you might miss the best moment to take your measurement. Could this maybe have to do with the results?

l. 462: This is a very interesting finding! Better write that the coherence is dependent on the separation distance but independent of the measurement distance, i.e., the position in the induction zone.

l. 471 ff.: You point out that "The decay parameter a shows a decreasing trend with increasing measuring separation." After reading the explanation given in lines 475–480 several times I still do not fully understand why longer travel time (or separation distance) leads to a less decay. And if I could follow the explanation, it would not explain why the value increases again for very long separations with ParkCast data. Maybe you can explain this better?

l. 489: You should explain the results shown in Fig. 11 at least briefly to justify your predictor selection shown in Table 5. Please mention the $\log(\sigma_m)$ thresholds.

Fig. 12: The scatter plots are not as informative as they could be. Please decrease the marker size and maybe add transparency to make it possible to see the density of the data points. Also a regression line would help to quantify the relation between $x$ and $y$.

Table 5: Please provide a better caption. What does for example bold font mean?

l. 564: The prediction accuracy is very good for this one example case but from Fig. 12 we know that the scatter throughout the whole dataset is quite high. Is this particular example (12.12.2013, 12:00–12:30) representative in any way? Maybe it is more interesting to show a plot for a case where the deviation between modelled and fitted curves equals the RMSE?

l. 599: The theory about atmospheric stability in section 2.5 can be removed then.

l. 625: The computation time was not mentioned before. Rather don't mention it only in the conclusions.

Fig. A2: Please add a grid and either add x-ticks or if that is not possible out of confidentiality concerns, remove [Hz] and [m/s$^2$] from the labels.

"Technical corrections"

l. 16: "taking values", "1..."

l. 45: "lidar is a remote sensing technology"

l. 67: Remove "Some"

l. 81: Remove "again"

Fig. 1: The text in the plot is too small. Maybe simply enlarge the whole plot a bit.

Fig. 2: The caption is a stub.

l. 109: "...with only a few simple parameters." or better: "... with as few parameters as possible."

l. 114: "a linear function or a more complicated term."

l. 127: Please check all occurrences of "the both" and use either "the two" or only "both" instead.

l. 131: "dimensionless frequency" should be written in roman script.

l. 141: "projected onto"

l. 150: Please check all occurrences of "starring" and change them to "staring".

l. 151: Please capitalize "Oppenheim".

l. 155: "...laser beam pointing into a fixed direction."

l. 200: "Since..." the sentence is not correct. Please rewrite.

l. 234: "So as the travel time Dt" is not a sentence. Please rewrite.

Table 1: Unusual table style. Consider three columns including a header for each column instead of lines separating the rows.

l. 351: "sampling rateS because of THEIR"

Table 2: The Matlab like interval syntax is maybe not the best way to present the measurement distances. Better write, e.g., 30,90,...,990, Dec is abbreviated, June is not.

l. 375: What does C stand for? Probably a formatting error here?

l. 379: and 381: No new paragraphs here.

l. 399: "referred to throughout"

l. 419: Remove both occurrences of "as"

l. 459: The idea for colours and markers is good but most of it is not visible in the plot. Try slightly thinner lines and different marker sizes for different colours (e.g. blue circle tiny, red circle small, yellow circle medium...).

Figure 9: The caption does not explain the difference between a) and c) and between b) and d).

l. 579: "And the more noisy..."

l. 586: "nacelle-based"

l. 611: "error is" or "errors are"

References:

Willis and Deardorff (1976): 10.1002/qj.49710243411

Schlipf et al. (2010): 10.18419/opus-3915

Kelberlau and Mann (2019): 10.5194/amt-12-1871-2019

Mann (1994): 10.1017/S0022112094001886

Sathe and Mann (2012): 10.1029/2011JD016786

––––––––––––––––––––––––––

---

## Referee Comment (RC2) · Anonymous Referee #2 · 20 Apr 2020

This manuscript presents a statistical model of longitudinal coherence describing the evolution of turbulent structures in the wind as they travel downstream. The topic is very relevant for lidar-assisted control applications (and other wind preview-based control applications), where a good understanding of the correlation between the wind at the measurement point and the turbine is needed. There has been previous work in the literature focusing on developing wind evolution coherence models with parameters describing atmospheric conditions as inputs. However, the existing models don't necessarily fit observed data well for all atmospheric conditions. This manuscript includes many additional atmospheric parameters as predictors to estimate the coherence and also applies a machine learning approach to model wind evolution. The advantage of the machine learning approach is that the set of parameters used to predict wind evolution can be adapted to the measurements available at a given location. The manuscript describes novel and relevant research, and overall is well written.

[Figure]

Despite the significance of the research, there are several areas that I believe should be addressed. First of all, it would be useful to understand how the accuracy of the developed model compares to existing wind evolution models (e.g., Kristensen, 1979; Simley and Pao, 2015; possibly Davoust and von Terzi, 2016). The manuscript claims that the developed model is sufficiently accurate to model wind evolution, but if possible, it would be interesting to know how much it improves over these simpler models.

Second, the manuscript is very well organized and easy to follow! But the English usage could be improved throughout the manuscript. For example, there are several sentence fragments, the word "the" is used in many places where it is not needed, and some of the language seems too casual (e.g., pg. 10, ln. 255: "Think of making a regression model from some data.").

My biggest concern with the manuscript is that the analysis assumes that the spatial averaging effect of the lidar can be ignored (discussed on pgs. 7 and 8). The authors correctly show that the lidar weighting function does not affect the measured coherence as long as it is assumed that wind evolution can be ignored within the probe volume (Taylor's hypothesis is applied). But this over-simplifies the problem. For example, the authors are estimating the wind evolution between the two adjacent range gates, with range gate spacing as low as 27.5 m. But pulsed lidars typically have a Full Width at Half Maximum width of ∼30 m. Therefore, it seems problematic to assume Taylor's hypothesis within the 30 m probe volume, but assume wind evolution between the two range gates separated by a similar distance. From my own analysis of the impact of spatial averaging on the measured coherence, when the wind evolution model is applied within the probe volume as well as between range gates, the presence of the weighting function significantly impacts the measured coherence. This has the effect of increasing the low frequency coherence but causing the high frequency coherence to decay much faster. Therefore, it seems likely that ignoring spatial averaging altogether in this work leads to incorrectly fitting the coherence model.

The authors should include some analysis comparing the modeled coherence with and

without wind evolution within the probe volume, using the results to either justify their approach or to show that Taylor's hypothesis cannot be ignored. A better approach would be to include the impact of spatial averaging and find the a and b parameters that best fit the measured coherence when the wind evolution model is combined with the spatial averaging model. In principle, this approach is similar to the method developed by Schlipf et al., 2015 (Meteorologische Zeitschrift), but much simpler since only a staring lidar mode is used.

Specific comments:

-Pg. 2, ln. 58: "adapted the Pielke and Panofsky's model by introducing a new parameter..." More accurately, the paper by Simley and Pao (2015) took the form of the coherence model for transverse and vertical separations suggested by the following paper, and adapted it to longitudinal coherence:

R. Thresher, W. Holley, C. Smith, N. Jafarey, and S.-R. Lin, "Modeling the response of wind turbines to atmospheric turbulence," Department of Mechanical Engineering, Oregon State University, RL0/2227-81/2, Corvallis, OR, Tech. Rep., Aug. 1981.

-Section 1: Introduction: Another very relevant paper should be discussed in the literature review section. The following paper discusses fitting lidar-measured coherence to the longitudinal coherence structure suggested in Simley and Pao (2015):

Analysis of wind coherence in the longitudinal direction using turbine mounted lidar S. Davoust and D. von Terzi 2016 J. Phys.: Conf. Ser. 753 072005

-Eq. 9: The last index "j" should be changed to "i".

-Pg. 8, ln. 188: This paragraph and Fig. 4 are hard to follow. I would suggest labeling the angles the text refers to in the figure, and also provide some equations to support what you are trying to explain.

-Pg. 8, ln. 196 - pg. 9, ln. 208: In this discussion, it is a little hard to tell if yaw misalignment is required by the coherence estimation method, or if it is optional. This

becomes obvious later, but I think here it would be good to explain that the final model allows different combinations of predictors (including yaw misalignment) depending on availability.

-Section 2.5: Can you compare the predictors you are using to the predictors used in previous longitudinal coherence models in the literature (e.g., Kristensen, 1979; Simley and Pao, 2015)? It would be insightful to understand which new parameters are included in this study.

-Pg. 9, ln. 213: It would be good to define turbulence intensity here.

-Pg. 10, ln. 233: "thus how likely or to what extent the local terrain changes" makes it sound like the terrain variations are the primary reason the coherence would depend on "d". But even if the terrain stays the same, I would still think there could be a dependence on "d".

-Pg. 10, ln. 234: "For prediction, it is not possible to obtain Delta t_maxcorr." Why can't it be determined? It can be calculated just like all the other predictors, right? Also, on pg. 6, ln. 130, you say that the d/U approximation is not used in this study and Delta t_maxcorr is used. Which of these statements is right?

-Pg. 10, ln. 251: (Chen, 2019). Can you describe how this current manuscript compares to the earlier work? Better yet would be to discuss this in the introduction.

-Pg. 11: Section: "Hyperparameters of GPR": In general, this section would be more clear if the specific variables discussed were connected to the wind evolution application.

-Pg. 11, ln. 271: "where x is the input vector of different parameters" Can you provide an example of what these input parameters are in your application?

-Pg. 12, ln. 278: "where X is the aggregation of all input vectors." Can you explain in more detail? What are the dimensions of X? # of parameters x # of observations?

-Eq. 19: I did not see these basis functions or the coefficients beta discussed anymore in the manuscript. Can you describe how you chose the basis functions and how the coefficients were estimated? And how do these values affect the final estimate in Eq. 22?

-Pg. 12, ln. 286: Please describe in more detail why you are using a kernel function (I assume because then you don't need to actually define the functions "phi(x)").

-Pg. 12, ln. 288-290: I don't think these sentences are needed, since in the next paragraph, you thoroughly introduce the ARD-SE kernel.

-Pg. 12, ln. 291: Why is the ARD-SE kernel chosen? And please provide a reference about this kernel.

-Eq. 21: Please define "D".

-Pg. 12, ln. 296: "A relatively large length scale indicates a relatively small variation along corresponding dimensions in the function" From Eq. 21, it seems more accurate to say that a large length scale relative to the amount of variability in the predictor indicates a smaller variation along the corresponding dimensions. For example, it seems the size of the length scale is only meaningful by itself if all of the predictors have been normalized to the same std. dev. Is this correct?

-Eq. 22: Can you state the difference between X and X_* here? Also, this equation seems to just be saying that the conditional distribution is normally distributed, so I don't think the right hand side of the equation adds anything. Perhaps it would be less confusing to just explain that the function values are estimated given input parameters X_* by conditioning f on the training parameters and observations, X and y, as well as X_*: f_*| X, y, X_*. Finally, how are the estimates formed from the resulting distribution? Is the mean value used?

-Table 2: The lidar weighting function width (e.g., Full Width at Half Maximum) would be a relevant parameter to list in this table.

-Pg. 15, ln. 367: "The threshold for both are 6 m/s and 3 m/s". How are the thresholds used? For example, are these the thresholds in terms of deviation from the mean value of the three-data point window? Also, how is the standard deviation defined here? How many data points are used to calculate the std. dev.?

-Pg. 18, ln. 436: "all the PDFs supported by MATLAB" Is there a particular MATLAB toolbox you are referring to here? Also please provide a reference for MATLAB.

-Pg. 20, ln. 465: "all the fitted curves of the coherence are grouped together proves it is reasonable to model the wind evolution based on dimensionless frequency". Do you mean that they are grouped together at high frequencies (f_dless > 0.1)? Additionally, "proves" seems like a strong statement here. Maybe "suggests"?

-Fig. 9: The caption should refer to the different subplots that are labeled (a-d).

-Pg. 22, ln. 486: "all the potential predictors are included... to determine the characteristic length scale". Please describe how the training to find the length scales is performed.

-Pg. 22, ln. 489: "Figure 11 illustrates a comparison among the log(sigma_m)." As mentioned earlier, it doesn't seem fair to compare the sigma_m magnitudes unless all of the predictor variables have been normalized to have the same std. dev. (or some other normalization). Is this done?

-Table 4: Please explain "standard deviation of observed responses" in more detail. It's not clear what the "observed responses" are.

-Fig. 13: I'm not sure how to interpret this figure. Are there errors in the plots or the legend? The legend lists separate solid lines and dotted lines, but I don't see both in the plots. Do they perfectly overlap? Additionally, the legend says that blue dotted is a fitted case and solid red is the predicted case. But these two lines are very far apart, which does not support the claim that the fitted and predicted curves are very close. Additionally, what is the significance of the particular period being shown here. Is this

one of the periods with the best match between the fitted and predicted coherence? Or is it representative of a typical case?

-Pg. 26, ln. 556: "R^2 at least over 0.65." What is the significance of 0.65 as an indication that the prediction accuracy is "satisfactory"?

-Pg. 26, ln. 573: "no obvious relevance between the error and values of any of the predictors is indicated in both figures." I don't quite agree. I think there are some interesting trends, like in Fig. 14 (b), the RMS prediction error decreases as sigma_l increases. Trends can also be observed in Fig. 15 (b) and (f).

-Pg. 29, ln. 612: "capable of achieving a parameterization model with sufficient accuracy for the prediction of wind evolution." This seems like a very strong statement to make. Please provide some more context for this statement. How is "sufficient accuracy" determined?

-Pg. 29, ln. 616: "methods to improve the estimation of the coherence and the wind statistics are desired." What are some of the shortcomings of your current approaches that you think could be improved?

-Fig. A1-A5: I think if there are appendix figures, they should be in a labeled appendix section.

-Pg. 30, ln. 623: Is the current computational time acceptable for real-time applications?

-Section 6: If possible, it would be nice to hear your thoughts on whether the chosen coherence formula structure (Eq. 5) could be improved. In other words, the paper mostly focuses on how to estimate the a and b parameters, assuming Eq. 5 is the right model. But is this model good enough? For example, Simley and Pao (2015) show that this kind of model did not fit stable atmospheric conditions very well.

---

## Referee Comment (RC3) · Mark Kelly (Referee) · 20 Apr 2020

review of "Parameterization of Wind Evolution using Lidar"

DRAFT WES 2020-50, SUBMITTED BY Y.CHEN, D.SCHLIPF, AND P.W.CHENG
*Reviewed by M.Kelly, 13 Apr. 2020*

**General comments**

This work examines evolution of advected turbulence, in terms of spectral coherence, with the motivation of wind turbine control assisted by inflow measured by lidar. The topic is quite relevant to wind energy, and fits well with the journal (WES).

There is some interesting content and potentially useful results, with the use of GPR

and Bayesian inference being quite nice.

Unfortunately the paper appears to be somewhat 'unfinished'; perhaps it is also due to the lack of English fluency or preparation time. For example, the abstract has simply copy-pasted a few sentences from the paper, and repeats in a cumbersome way: *'This paper aims to achieve parameterization model for the wind evolution model to predict the wind evolution model parameters'*. The paper needs to be proofread by somebody with English fluency, at any rate.

The abstract does not clearly provide an idea of the work done, and while the text has more detail, it is not clear throughout; I am not sure that readers could repeat what has been done.

More importantly, there are inconsistencies that have not been considered, and should be addressed/rectified; perhaps most significant are the form itself chosen for coherence (see derivations below in *Specific comments*), and the use of Taylor's hypothesis for some (but not other) parts of the model/parameterization. There are a number of details and also explanations which are missing, but which could hopefully be included, to make the work publishable. The results/performance are a bit overstated (in English), but this is not needed, as the numbenical results presented tell the story less subjectively—and are good to share with the wind energy community, provided that they are given with sufficient detail, replicability, and consideration.

**Specific comments**

Line 15 and elsewhere later in the paper: while the authors define 'wind evolution' as squared coherence, they imprecisely define such (e.g. *"Coherence is a dimensionless statistic in the frequency domain"*). Specifically, this should be written as temporal coherence (time shift, frequency domain), in contrast with spatial (wavenumber spectra) coherence. Further, coherence does not describe the "correlation between two signals", but rather the correlation between spectral components of two signals.

Do lines 19–20 not imply that use of Taylor's hypothesis means ignoring wind evolution? This may be relevant, for consistency later (line 235).

Line 35: The statement "*dependence of coherence on separation and atmospheric stability was not adequately researched*" lacks reference and/or explanation. It was not adequate, according to whom, or how?

Line 36–38: You write "*The longitudinal coherence differs from the lateral and vertical coherence because the former measures the correlation with respect to time lag while the latter with respect to spatial separation.*" This is not correct: longitudinal, lateral, and vertical coherences all depend on $f$, based on integration over time lags; they are depending on spatial separations in the respective directions. For the longitudinal coherence to give spectral correlations 'with respect to time lag' $\Delta t$, then the longitudinal separation is related to $\Delta t$ in some way, though you have stated before this point that you are not using Taylor's hypothesis.

Line 112/equation 2: how is this a function of frequency ($f$)? I.e., include the $f$-dependence on LHS, and also within $\tau$ on RHS.

There appears to be incompatibility between Eqs. 3–4 and Eq. 5; in particular (5) is missing $\sigma$ and $U$. Further, in Section 4.2 and Fig. 10, you analyze the behavior of $a$ with $\Delta t_{\mathrm{maxcorr}}$ (stating $a \propto \Delta t_{\mathrm{maxcorr}}^{-0.49}$), but do not consider that the equations already imply a $\Delta t$ dependence.
That is, eqns. 2 and 5 give

$$C\Delta t / \tau = \sqrt{(af\Delta t)^2 + b^2};$$

however (3)–(4) with (2) give

$$(CI_u f\Delta t)^2 = (af\Delta t)^2 + b^2$$

where the turbulence intensity is defined $I_u \equiv \sigma/U$. Thus one sees that

$$a = \sqrt{(CI_u)^2 - \left(b/f_{\text{dless}}\right)^2} = CI_u\sqrt{1 - \left(\frac{b}{C\Delta t/\tau}\right)^2}.$$

In the limit of high dimensionless frequency or large turbulence intensity, i.e. without the offset $b$, then $a \to CI_u$ like Pielke & Panofsky (1970). But in the limit of small turbulence intensity or low dimensionless frequency (small $f\Delta t$), i.e. a large offset, then we see $a$ become imaginary, implying a (nonphysical) coherence oscillating with $\Delta t$ or $\Delta x$.

The form (5) is the same as that of (4) in Simley & Pao (2015), with $d_\ell$ in the latter replaced here by $U\Delta t$, and $b$ here replacing their $abd_\ell$; this should be noted, and the text is not quite clear nor correct following your Eqn. 5. You do note the reason for keeping $\Delta t$ (instead of using $\Delta t = d_\ell/U$), but why did you drop the spatial separation dependence ($d_\ell$) from the '$b$' part of the Simley & Pao (2015) expression? From your logic for the '$a$' term, then instead of just $b$ you would have $bd_\ell$ (but not $bU\Delta t$).

In lines 145-9: your text is a bit imprecise here—the sonic anemometer has a measuring volume as well (not a point), it is just much smaller than the lidar's. Also, among the reasons why longitudinal coherence from lidar deviates from that calculated via sonic anemometers, one key possibility is missing: the validity of Taylor's hypothesis.

Line 153: what do you mean by "complete coherence curve"?

Lines 157–164: please include references, as this is not original.

Line 184: neglegt of the spatial averaging effect and $w(x)$ in $\gamma^2_{i,j}$ also demands use of Taylor's hypothesis. This should be noted (along with its potential inconsistency).

Lines 193–5: the sentence *"To retrieve the longitudinal coherence in this case, the above discussed spatial averaging effect must be coupled to a specific turbulence model (Schlipf, 2015; Mann et al., 2009), and thus the wind evolution model is included in the final model implicitly"* does not quite make sense. Could you clarify?

Line 217: by "*its definition*", do you mean the definition analogous to (13), where the time-lag $s$ is replaced by spatial separation $r$?

Lines 217–19: If you say $L_{int} = T_{int}U$, then aren't you just using $U$ as a potential predictor somehow? Also, isn't this inconsistent with the previous section, where you state that Taylor's hypothesis is to be avoided? Or, is Taylor's hypothesis avoided only for certain aspects? Please clarify.

Line 223: The statement "*atmospheric stability represents a global effect of the boundary layer on the wind field*" is not quite correct.
From what you measure, or via M-O similarity, it is a 'global' effect from the surface, and potentially only through part of the ABL (sometimes not even above the surface layer in stable conditions).

Line 234: "*So as the travel time $\Delta t$.*" is not a sentence. What are you trying to convey here?

Line 235: So you are meaning that distance $d$ is used instead as a predictor.

Line 250: by "*performs the best*", perhaps you should use 'performs well'—unless you explain what 'best' means (i.e. what other models).

Line 257: The phrase "*underlying functions of the data*" is not clear. Do you mean behavior conditioned on other variables, or relation to other variables?

Table 1/ line 255+ : Is it even possible to use the fourth or even third moment, given the large sampling uncertainty involved for highter-order moments? Please see and reference e.g. Lenschow, Mann & Kristensen (1994) and Ch.2 of Wyngaard's textbook (2010), to understand and defend use of $\mu_4$—let alone $\mu_3$.

Line 284: To be clear and consistent, can you not specify that $\beta$ is a weight, and the 'basis function' $h(\mathbf{x})$ maps the means into the new space?
What is meant by '*Basis function is one of the hyperparameters*'? I.e., how is a function a parameter, or is $h(\mathbf{x})$ already assumed to have some form, possibly related to the $\phi(\mathbf{x})$

forms?

lines 289–296: $\sigma_m$ is not a 'length' in the physical sense; it has units of whatever $x_m$ has. Thus it is a characteristic magnitude for the predictor having index $m$.

lines 299–302: To be explicit, the RHS of (22) does not contain a 'conditioning bar'. I.e. to help the reader and match the text, show in the math that the joint Gaussian prior is conditioned on $X_*$ (is the eqn. correct?). Most readers will not have read Duvenaud's PhD thesis, so it is useful to help them understand.

Lines 307–311: why $k = 5$? IF it is due to needing a large enough sample for verification, then this should be stated.

Line 335: was the lidar on the nacelle, at what height?

Line 350: please be more clear and specific, and also include references.

Line 362–366: Why are two different filtering types used?
State why -24dB for CNR; include reference.

Line 375: Using "$C_N^2$" to symbolically write '$N$ choose 2' unique pairs, is not standard practice. You can write $\binom{N}{2}$ or equivalently $N(N-1)/2$.

Line 479–480: need citation for Levenberg-Marquardt algorithm.

Fig.9 caption: mention which plots belong to which campaign.

Fig.11 / § 5.1 : why not plot $\sigma_m^{-2}$? This is what is actually used in the ARD-SE kernel shown in eqn.21, and its behavior more clearly demonstrates relevance.

Line 489–90: you state "*predictors are selected according to different preset limits of the log($\sigma_m$) considering different cases of application or data availability*", but what are these preset limits?

Line 494 / Table 4: how/why did you choose the initial $\sigma_m = 10$?

Line 509: I am not sure that $R^2$ of 0.65 is "*satisfactory*"; perhaps this could be just

written in terms of $R^2$ and RMSE without the subjective claim. Also, "*all situations*" is not quite consistent with just the recommended cases (i.e. it implies all cases).

Lines 519–520: This is a good point, and it would be useful to repeat this earlier, when introducing the potential predictor variables because some of them appear redundant.

Lines 524–527: It appears that you are conflating two things here, one of which you are missing—**applicability of Taylor's hypothesis** will also affect $L$ compared to $T$ via $U$, whereas this is not the case for the usage of $U$ to 'convert' $\sigma_u$ to $I_T$.

Lines 532–534: perhaps $\mu_3$ or $\mu_4$ could help prediction; but to be responsible, one needs to mention that [1] the uncertainty in these quantities are very large (Lenschow et al. 1994 reference), and [2] lidar may not be able to consistently measure these. Further, these higher-order moments are likely more affected by your filtering.

Lines 537–539: this is likely due to the implicit co-dependence I derived above, i.e. $a$ is a function of $CI_T$ and $b/f_{\mathrm{dless}}$. Your finding confirms also the need/utility to consider the behvior of the parameters involved.

Lines 542–554: what about cross-comparison using the sonic? Were the wind directions such that the sonic (at 270m upstream) could be compared to the lidar (e.g. at 163.5m upstrean)?

Line 555-6: isn't this '*satisfactory*' $R^2 \geq 0.65$ only true for certain cases and variables?

Figure 14 / Line 560 and afterward: these plots do not responsibly/transparently show prediction error, as they don't give an idea of the magnitude of $a$.
You should plot percentage error or similar; given that $a$ can be small depending on $b$ and $I_T$ (as derived above), the plotted differences in $a$ might be relatively significant.

Lines 574–5: if the whiskers are large because of sample size, then why not (also) account for this via $\sqrt{n}$?

Line 576: The claim "*it is proven that the Gaussian process regression is capable*

*of achieving an accurate parameterization model*" is an overstatement. It is DEMON-STRATED (not proven) that the GPR was able to predict two coherence model parameters with an $R^2 \geq 0.65$ in chosen cases (not simply 'accurate').

**Technical corrections**

There are many English usage/grammatical corrections and suggestions, which are included in the attached annotated PDF-file. I thus only include a sample of them here in this list. The sentence structure and writing is unclear or ambiguous in numerous places; the paper really should be reviewed and edited by somebody with adequate fluency.

I list some of the specific corrections below, but since there are >300, after the first few I include the line numbers, which refer to the annotated attached PDF. After page 16 I did not correct much English; this is left to the authors for the next draft.

- Abstract/line 1: One (generally) shouldn't copy sentences from the introduction into the abstract (further, this first sentence is a definition); 'turbulence' should be 'turbulent'; pluralize 'structure'; delete 'of the eddies'; replace 'while the eddies' with e.g. 'as they'; replace 'by the main flow over' with 'through'.

- L.2–4: Remove 'the'; change 'because' to ':'; remove 'only'...see annotated PDF for more details.

- L.5–7: These 2 sentences are quite unwieldy (cumbersome) and also somewhat tautological—particularly for an abstract, also with repeated phrases that need to be reduced/condensed. Please correct the English usage here.

- L.12–13: First sentence can be corrected from "*Wind evolution refers to the physical phenomenon that the turbulence structure of the eddies changes over time while the eddies are advected by the main flow over space.*" to something

like 'WIND EVOLUTION' REFERS TO THE PHYSICAL PHENOMENON OF TURBU-
LENCE STRUCTURES (EDDIES) CHANGING OVER TIME, WHILE THE EDDIES ARE
ADVECTED THROUGH SPACE BY THE MEAN FLOW..

- L.13-15: change "The mathematical" to 'A common statistical'; delete 'usually'; 'hereinafter for brefity, also' should be 'hereafter'; delete 'two time series data sets of the' and instead add 'measured at two different locations,' after 'velocity'; change 'with certain time shift' to 'calculated over varying time shifts'.

- L.17–19

- L.22

- L.25–27

- L.29

- L.32: run-on sentence; use parentheses as noted

- L.33–34

- L.36–38

- L.40, L.42–44

- L.47–48

- L.52 use of definite articles and parentheses

- L.55–59, L.61

- L.63 delete 'model'

- L.65, 67

- L.73–74

- L.79–81: delete a number of redundant words, add punctuation as noted

- L.122–123

- L.136

- L.151 Capitalize 'Oppenheim', here and elsewhere.

- L.156

- L.169

- L.177: have already introduced $U$ as mean wind speed (delete here).

- Page 8: L.186–7; 189; 196–9; 201–2

- Page 9: L.204; 210; 212; 220–2;

- L.224 and elsewhere: not 'Monin-Obukhov length', just use 'Obukhov length'

- Page 10: L.225–8; 233; 250–7

- Page 11: L.258–9; 261; 264; 269; 272

- Page 12: L.275–8; 288–9; 292; 295–6; 302; 304–5

- Page 13: L.308

- Page 14: L.335; 339; 346–7; 349–353; 363–5

- Page 15: L.367

- Page 20: L.470; 479

- Page 24, Table 5: Taylor is italicized under case 6, but should be Roman font.

- Page 25: L.537; 550

Please also note the supplement to this comment:
https://www.wind-energ-sci-discuss.net/wes-2020-50/wes-2020-50-RC3-supplement.pdf

**Supplement:**

[revised manuscript text omitted]

---

## Author Comment (AC1) · 24 Jun 2020

First of all, we would like to thank all the reviewers for their time taken to read our manuscript and their constructive comments. We have considered all the comments in detail and revised our paper accordingly. We believe that these comments have helped us to further improve the quality of our paper.

Please find below our responses to reviewer comments. The reviewer comments are repeated in black text, and our responses are provided in blue text.

[Figure]

**Response to comments of Felix Kelberlau**

**General comments**

Chen et al. develop a method to predict the coherence of horizontal wind velocity fluctu-
ations for mostly longitudinal separations. Their predictions are based on first to fourth
order wind speed statistics that can be calculated from either nacelle-mounted lidar or
mast-based in-situ anemometry. They use data from two measurement campaigns to
test their approach and find good results that are especially relevant for lidar-assisted
wind turbine control. The work lies therefore well in the scope of WES and is of broad
international interest. The paper builds up on an existing wind evolution model and
presents a novel approach to parameterise its two coefficients by means of machine
learning. The manuscript explains the study thoroughly and reproducibly, presents all
relevant results and discusses them critically.

We would like to thank the referee for the interest in this research.

Section 2.6 "Gaussian Process Regression" lies outside my field of expertise and I can
therefore not evaluate if the chosen model is suitable for the task of parametrization
the wind evolution model.

Gaussian process regression is a powerful modeling tool. One of the objectives of this
paper is to explore if this method can be applied to wind evolution modeling and the
results have demonstrated its potential.

The manuscript is overall understandable but would benefit greatly from being proof-
read by a native English speaker or similarly qualified person before publication. I
recommend reconsideration for publication after major revisions.

Thank you for your suggestion. The revised version will be proofread before submission.

**Specific comments**

l.2: I assume you mean "the mean flow" (also l.13 and all other occurrences).
Yes. This has now been corrected throughout the paper.

l. 50: The introduction would benefit from references to research that support Taylor's frozen turbulence for very large turbulent structures but limit its applicability for long separation distances or small scale turbulence such as Willis and Deardorff (1976), Schlipf et al. (2010), and Kelberlau and Mann (2019).
Thank you for your suggestion. We will mention the relevant research in the introduction.

l. 59: "the vertical intercept" It would be better to describe the second parameter without referring to the coherence-frequency plot that is not yet introduced here.
Thank you for your suggestion. This sentence has now been rephrased.

l. 61: "Mann spectral velocity tensor" Mann (1994) should be cited here.
Thank you for your suggestion. The citation has been added to the text here.

l. 68: "If any data... is also available..." Please mention which data is available or would be of interest.
We noted this sentence is not well formulated and have now rephrased it. And the data used in this research is introduced in Sect.3.

l. 94: Please introduce this travel time as a function of the mean wind speed

here.

In this context, the travel time $\Delta t$ is in a general sense, not specifically the travel time approximated by $d/U$, which is defined as $\Delta t_{\text{Taylor}}$ in l.130.

l. 97: Please explain why "it is not possible to predict every point of the coherence curve". For my understanding, the coherence curve is visible on a plot like in Fig.1. Do you refer to not having not enough data to smoothen the curve or not having data for all separation distances?

In this sentence, the *coherence curve* means the *estimated coherence curve*, which is the "reality" we aim to approach. And the smooth coherence curve is acquired from a *wind evolution model*. We think it is important to clearly distinguish the different meanings of *wind evolution* and *wind evolution model*. Wind evolution is a physical phenomenon (the "reality") and wind evolution model is a model used to approximate it (There are different wind evolution models). It is not possible to predict wind evolution but to predict the parameters of a wind evolution model and use this model to approximate it.

In fact, we'd like to explain our prediction concept at an abstract level in Sect. 2.2. We think that the key to using machine learning to build predictive models is to find suitable *predictors* and *targets* — This is the process of abstracting and condensing information. Essentially, using a wind evolution model is to condense the information in the estimated coherence into several model parameters, which are predictable. We will improve this section to make the logic more understandable.

l. 100: Do you mean "...according to measured wind velocity time series by a parameterisation model"?

"Wind field conditions" here refer to all variables related to a wind field, not limited to statistics of measured wind velocity time series. But we noted that this word might be not precise enough. We will explicitly write down the types of relevant variables instead.

Fig. 1 and Fig. 2: In general, it is good to visualize the workflow like done here. But both figures show overlapping information and I recommend to merge them into one figure. The numbering used in Fig.2 with an explanation in the text and caption(!) is more informative than the keywords currently used in Fig. 1. A figure and its caption should be self explanatory whenever possible. Please try to improve the text l. 98-106 for better understanding.

Thank you for your suggestion. We will merge Fig. 1 and Fig. 2 into one figure and modify Sect.2.2 accordingly.

l. 120: Please describe which frequency you are referring to. Probably the frequency of the horizontal wind velocity fluctuations.

Yes, it is the frequency of horizontal wind velocity fluctuations. We've made it clear in the text.

Maybe also introduce the wavenumber k here that is used as a measure of eddy size in many other publications.

Thank you for your suggestion. We agree that wavenumber is also a common measure in spectral analysis of turbulence. However, it is not used in this paper because our study is based on dimensionless frequency. But we think it makes sense to mention the relationship between wavenumber and dimensionless frequency.

Eq.(5): It is not clear where (5) comes from. If you do not want to include the complete deduction, I suggest to give a reference that shows it and uses the same form of the equation. In Simley and Pao (2015), a and b are defined a bit differently, I think.

We will include the complete deduction and clarify the reason for adapting the equation of Simley and Pao (2015) in the revised manuscript.

l. 147: You should include the weighting here: e.g. "...but the weighted average of the wind speeds within the measurement volume"

Thank you for your suggestion. The corresponding text has now been modified.

l. 148: This is a bit ambiguous because spatial averaging does also refer to combining data from different measurement volumes in different lidar beam directions. Better write: "so-called line-of-sight averaging effect of lidar". (also l. 156)
Thank you for your suggestion. The corresponding text has now been modified.

l. 151: Please refer to more fundamental work (Nyquist-Shannon sampling theorem).
Thank you for your suggestion. The following reference has been cited: C. E. Shannon, "Communication in the Presence of Noise," in Proceedings of the IRE, vol. 37, no. 1, pp. 10-21, Jan. 1949, doi: 10.1109/JRPROC.1949.232969.

l. 152: You should mention the sampling rate of the lidars here (not only in the table)and compare it with the frequency of the eddies that you want to detect.
According to the paper structure, Section 2 introduces the theoretical framework of this study and gives related discussions in a general way. Measurement-related content is first introduced in Section 3.

l. 158: The line-of-sight weighting of a pulsed lidar is usually approximated by a triangular function as in e.g. (Sathe and Mann (2012)) which is a sinc2function in the frequency domain.
Thank you for your suggestion. We agree that weighting functions for the volume averaging effect of pulsed lidars could have different functional forms. We've added the triangular function as an example of weighting functions.
Indeed, the functional form of the weighting function mainly depends on the shape of emitted pulses and the sampling of backscattered pulses. For example, a triangular function is used for the case where the pulse shape is assumed to be ideal rectangular (Sathe and Mann (2012)). For Leosphere pulsed lidar systems, a Gaussian weighting

function is usually used, see e.g. the following reference:

Carious, J.-P.: Pulsed Lidars, in: Remote Sensing for Wind Energy, DTU Wind Energy-E-Report-0029(EN), chap. 5, pp. 104–121, 2013.

l. 184: It should be considered that w(x) is approximately 0 for fluctuations that occur with a wavelength of twice the length of the illuminated section of the lidar beam (or length of the range gate). In this case the measurement signal would be determined by noise only. I suggest to estimate a range of critical frequencies based on the length of the range gates. This range of critical frequencies should be considered in the further analysis, if it is relevant for the results.

Thank you for your suggestion. That is a good point to check.

According to Schlipf (2015), the critical wavenumbers are $2\pi/W_L$ ($W_L$ is the full width at half maximum) and its harmonics. The relationship between wavenumber and dimensionless frequency is $kd/2\pi$. Thus, the first critical dimensionless frequency is $d/W_L$. For example, consider $W_L = 30$ m (for Leosphere systems) and $d = 27.25$ m (the smallest separation of LidarComplex, which is the most critical case), $d/W_L \approx 0.91$, which is located in the filtered part (the grey area). Therefore, the critical dimensionless frequency is not relevant for the results. This discussion will be briefly mentioned in the related part.

Schlipf, D.: Lidar-assisted control concepts for wind turbines, Dissertation, 2015.

Your derivation assumes furthermore that the weighting function is identical for all range gates. This is only true if the laser beam is well collimated. Is this the case for the lidar devices used in this study?

We agree that assuming identical weighting functions for all range gates is a simplification. As mentioned in the paper, the derivation is based on ideal assumptions.

l. 199: You could mention that a lidar with additional beams would help here and could also be used to avoid yaw-misalignment.

Thank you for your suggestion. This has now been added to the corresponding text.

l. 208: What is the expected order of magnitude for the misalignment angle?
First of all, we must emphasize that we don't have the data of turbine misalignment. What we have is the yaw position of the turbine and the wind direction measured on a met mast located 295 m away from the turbine. When calculating the misalignment angle $\alpha$, the mean wind direction at the turbine is approximated with the mean wind direction measured on the met mast (please note the possible uncertainty). $\alpha$ is approximately normal distributed, with $\sigma \approx 5°$.
How much "decorrelation" do you expect from a turbulence model (e.g. Mann (1994)) due to the resulting lateral separation? Can you quantify the order of magnitude of the resulting error approximately?
First of all, we want to emphasize that we did not ignore the influence of the misalignment angle on the horizontal coherence, but defined it as a predictor.
The resulting lateral separation depends on the separation between two range gates. Here, we make a simple comparison based on the coherence model of Kaimal spectrum:

$$\gamma(r, f) = \exp\left[-12 \cdot \sqrt{(\frac{f \cdot r}{V_{\text{hub}}})^2 + (0.12\frac{r}{L_{\text{c}}})^2}\right]. \tag{1}$$

We find one data block where $\alpha = 0°$ and compare the longitudinal coherence estimated from this data block with the theoretical lateral coherence calculated according to the above equation, assuming $\alpha = 5°$ (see the attached figure). Two measurement separations are included, $d_1$ = 27.25 m and $d_2$ = 81.75 m. The mean wind speed $V_{\text{hub}}$ = 11.7 m/s.
l. 213: Please always write which variable you are referring to when you mention standard deviation sigma.
Thank you for your suggestion. This has been improved.

l. 236: It would be good to introduce the variable alpha already in 2.4 (l. 199

and Fig.4) if you refer to it here.
Thank you for your suggestion. The introduction of alpha has now been moved to
Sect.2.4.

l. 263-266: This sentence is very long and difficult to understand.
Thank you for your suggestion. We will try to rephrase it to make it more comprehensible.

l. 332-343: Please reassess which information should be given here: I miss:
the measurement height of the lidar, length of each range gate, measurement distances...Some of these values are given in Table 2 but should also appear here.
Thank you for your suggestion. We've now added these information to the corresponding text.
The information about the coordinate system will not be used again later in the text
and do not need to be given at all then.
The wind coordinate system is defined for the processing of sonic data. Sonic data
from LidarComplex is used in the analysis (see Table 5).

l. 342 and 350: Main wind direction refers usually to the direction from where
the wind blows most frequently. Better write mean wind direction.
Thank you for your suggestion. This has been corrected throughout the paper.

l. 389: I suggest a similar filtering against the line-of-sight averaging. See comment for l.184. Probably it is not worth it to re-run the computations. But check in the
coherence plots, if the frequency range is relevant and if you see a random increase
in coherence in it.
Please find the response to the comment for l.184.

l. 397: Why do you not also filter the lowest percentile?

Because the value distributions of the parameters all have a long right tail. We've added this explanation to the text.

Fig. 6: Subfigure (e) would benefit from a zoom into just some few minutes of data with thin plot lines to show all velocities clearly and not on top of each other.
Thank you for your suggestion. We'd like to show the whole 30-min data block because the coherence is estimated from it. We will try to use thinner lines to make the curves more visible.

l. 445: That makes the difference caused by the reduced sampling rate in Fig. 7 (b)even more interesting. Earlier you write that "As long as the sampling rate of lidar is sufficiently high to acquire a complete coherence curve, it will not have a noticeable effect on the study of the coherence." but here you find that the influence on parameter b is big (logarithmic y-axis). What could be the reason? $f_s$ = 1/3 Hz means, at least at high wind speeds, that it is difficult to measure exactly after the eddy travel time passed. That means you might miss the best moment to take your measurement. Could this maybe have to do with the results?
Thank you for sharing your opinion about the possible influence of a low sampling rate. However, in this study, we found the difference caused by the reduced sampling rate too small to be observed in a plot with a normal y-axis. Both curves completely overlap. That is exactly the reason for using a logarithmic y-axis instead. Please note that a logarithmic axis will enlarge the difference between values lower than one.

l. 462: This is a very interesting finding! Better write that the coherence is dependent on the separation distance but independent of the measurement distance, i.e., the position in the induction zone.
Thank you for your suggestion. The sentence has now been improved.

l. 471 ff.: You point out that "The decay parameter a shows a decreasing trend

with increasing measuring separation." After reading the explanation given in lines 475–480 several times I still do not fully understand why longer travel time (or separation distance) leads to a less decay.

Because the modeled coherence curve is determined by $a$ and $b$ together. $b$ increases with separation between the two observed points, which means the coherence at zero frequency decreases with separation. In other words, "the starting point" of the coherence curve is lower, and thus the room for coherence decay becomes smaller.

And if I could follow the explanation, it would not explain why the value increases again for very long separations with ParkCast data. Maybe you can explain this better?

In our opinion, $a$ decreases with separation, and its decreasing trend gradually stops at a separation of around 300 m (see Fig.9 (c)). We are not sure why you think "the value increases again for very long separations with ParkCast data". If it is because of the upper whiskers, we think this indicates the value is more scattered for long separations. When we observe the trend, we mainly focus on the value range of 25th to 75th precentiles (the box) and the median value.

l. 489: You should explain the results shown in Fig. 11 at least briefly to justify your predictor selection shown in Table 5. Please mention the log($\sigma$m) thresholds.

Thank you for your suggestion. We will explain the predictor selection in more detail.

Fig. 12: The scatter plots are not as informative as they could be. Please decrease the marker size and maybe add transparency to make it possible to see the density of the data points. Also a regression line would help to quantify the relation between x and y.

Thank you for your suggestion. We will improve Fig.12 according to your suggestion.

Table 5: Please provide a better caption. What does for example bold font mean?

Thank you for your suggestion. The caption has been improved.

l. 564: The prediction accuracy is very good for this one example case but from Fig.12 we know that the scatter throughout the whole dataset is quite high. Is this particular example (12.12.2013, 12:00–12:30) representative in any way?

We chose this data block based on two principles: 1) data integrity and 2) representative wind statistics. In this example, the lidar measured mean wind speed is 7.3–7.7 m/s and the lidar measured turbulence intensity is 0.10–0.12, for different range gates. These values appeared frequently in the selected period according to Fig.A1. Hence, we think this example is representative for the data involved in this study. And we decided to choose a data block as an example because we thought neither the coherence itself nor its prediction are intuitive (esp. $a$ and $b$ are predicted respectively but must be combined to give the modeled coherence). We noted a mistake here: the date should be 07 Dec. 2013. Figure 6, 8, and 13 are all plotted with the same data block. The date has now been corrected.

Maybe it is more interesting to show a plot for a case where the deviation between modelled and fitted curves equals the RMSE?

Thank you for your suggestion. Maybe we could try to plot two additional curves of the modeled curve $\pm$RMSE in the example plot to indicate the range of RMSE.

l. 599: The theory about atmospheric stability in section 2.5 can be removed then.

We did analyze the influence of atmospheric stability. However, the stability happens to be mostly neutral in the period we chose, and thus we could not get any clear conclusions from the data. We will add an explanation about this issue in the text.

l. 625: The computation time was not mentioned before. Rather don't mention it only in the conclusions.

We have found out in this research that computational time could be a matter of concern when applying machine learning methods. Thus, we'd like to suggest it as one of the topics worth studying in the future in the outlook.

Fig. A2: Please add a grid and either add x-ticks or if that is not possible out of confidentiality concerns, remove [Hz] and [m/s2] from the labels.

Thank you for your suggestion. The grids and x-ticks in Fig.A2 were intentionally removed out of confidentiality concerns. We will remove the units as well based on your suggestion.

**Technical corrections**

l. 16: "taking values", "1..."
l. 45: "lidar is a remote sensing technology"
l. 67: Remove "Some"
l. 81: Remove "again"
Fig. 1: The text in the plot is too small. Maybe simply enlarge the whole plot a bit.
Fig. 2: The caption is a stub.
l. 109: "...with only a few simple parameters." or better: "... with as few parameters as possible."
l. 114: "a linear function or a more complicated term."
l. 127: Please check all occurrences of "the both" and use either "the two" or only "both" instead.
l. 131: "dimensionless frequency" should be written in roman script.
l. 141: "projected onto"
l. 150: Please check all occurrences of "starring" and change them to "staring".
l. 151: Please capitalize "Oppenheim".
l. 155: "...laser beam pointing into a fixed direction."
l. 200: "Since..." the sentence is not correct. Please rewrite.
l. 234: "So as the travel time Dt" is not a sentence. Please rewrite.
Table 1: Unusual table style. Consider three columns including a header for each column instead of lines separating the rows.

l. 351: "sampling rateS because of THEIR"

Table 2: The Matlab like interval syntax is maybe not the best way to present the measurement distances. Better write, e.g., 30,90,...,990, Dec is abbreviated, June is not.

l. 375: What does C stand for? Probably a formatting error here?

l. 379: and 381: No new paragraphs here.

l. 399: "referred to throughout"

l. 419: Remove both occurrences of "as"

l. 459: The idea for colours and markers is good but most of it is not visible in the plot. Try slightly thinner lines and different marker sizes for different colours (e.g. blue circle tiny, red circle small, yellow circle medium...).

Figure 9: The caption does not explain the difference between a) and c) and between b) and d).

l. 579: "And the more noisy..."

l. 586: "nacelle-based"

l. 611: "error is" or "errors are"

We'd like to thank the reviewer again for these suggestions for technical corrections. We have considered these comments in detail and made corresponding corrections and improvements.

**References**

Willis and Deardorff (1976): 10.1002/qj.49710243411

Schlipf et al. (2010): 10.18419/opus-3915

Kelberlau and Mann (2019): 10.5194/amt-12-1871-2019

Mann (1994): 10.1017/S0022112094001886

Sathe and Mann (2012): 10.1029/2011JD016786

**Response to comments of Anonymous Referee 2**

**General comments**

This manuscript presents a statistical model of longitudinal coherence describing the evolution of turbulent structures in the wind as they travel downstream. The topic is very relevant for lidar-assisted control applications (and other wind preview-based control applications), where a good understanding of the correlation between the wind at the measurement point and the turbine is needed. There has been previous work in the literature focusing on developing wind evolution coherence models with parameters describing atmospheric conditions as inputs. However, the existing models don't necessarily fit observed data well for all atmospheric conditions. This manuscript includes many additional atmospheric parameters as predictors to estimate the coherence and also applies a machine learning approach to model wind evolution. The advantage of the machine learning approach is that the set of parameters used to predict wind evolution can be adapted to the measurements available at a given location. The manuscript describes novel and relevant research, and overall is well written.

We would like to thank the referee for the interest in this research.

Despite the significance of the research, there are several areas that I believe should be addressed. First of all, it would be useful to understand how the accuracy of the developed model compares to existing wind evolution models (e.g., Kristensen, 1979;Simley and Pao, 2015; possibly Davoust and von Terzi, 2016). The manuscript claims that the developed model is sufficiently accurate to model wind evolution, but if possible, it would be interesting to know how much it improves over these simpler models.

Thank you for your suggestion. Indeed, we have also considered to compare the results of the GPR models and that of some existing wind evolution models. However, our

concern is if this kind of comparison would make sense given the different conditions in our study in comparison to the others. Here, we would like to take the results of Simley and Pao (2015) as example, because the wind evolution model used in this research is adapted from that one. Firstly, the curve fitting is done differently. In Simley and Pao's work, the objective function for fitting is the sum of the squared errors weighted by the corresponding power spectrum (See the equation (5) and the corresponding explanation in Simley and Pao (2015)). However, in our work, no weighting function is applied in the fitting. Therefore, the fitted coherence curves will be slightly different in both cases even for same data, and thus the corresponding model parameters will be slight different as well. Secondly, in Simley and Pao's work, the input variables used to determine the model parameters are supposed to be acquired from ideal point measurements because the model is developed from LES data. However, it is not possible for us to acquire equivalent input variables from the on-site measurements. Despite these difficulties, we will look into the possibility of making a comparison again.

Second, the manuscript is very well organized and easy to follow! But the English usage could be improved throughout the manuscript. For example, there are several sentence fragments, the word "the" is used in many places where it is not needed, and some of the language seems too casual (e.g., pg. 10, ln. 255: "Think of making a regression model from some data.").

Thank you for your suggestion. The revised version will be proofread before submission.

My biggest concern with the manuscript is that the analysis assumes that the spatial averaging effect of the lidar can be ignored (discussed on pgs. 7 and 8). The authors correctly show that the lidar weighting function does not affect the measured coherence as long as it is assumed that wind evolution can be ignored within the probe volume (Taylor's hypothesis is applied). But this over-simplifies the problem. For example, the

authors are estimating the wind evolution between the two adjacent range gates, with range gate spacing as low as 27.5 m. But pulsed lidars typically have a Full Width at Half Maximum width of $\approx$ 30 m. Therefore, it seems problematic to assume Taylor's hypothesis within the 30 m probe volume, but assume wind evolution between the two range gates separated by a similar distance. From my own analysis of the impact of spatial averaging on the measured coherence, when the wind evolution model is applied within the probe volume as well as between range gates, the presence of the weighting function significantly impacts the measured coherence. This has the effect of increasing the low frequency coherence but causing the high frequency coherence to decay much faster. Therefore, it seems likely that ignoring spatial averaging altogether in this work leads to incorrectly fitting the coherence model.

The authors should include some analysis comparing the modeled coherence with and without wind evolution within the probe volume, using the results to either justify their approach or to show that Taylor's hypothesis cannot be ignored. A better approach would be to include the impact of spatial averaging and find the a and b parameters that best fit the measured coherence when the wind evolution model is combined with the spatial averaging model. In principle, this approach is similar to the method developed by Schlipf et al., 2015 (Meteorologische Zeitschrift), but much simpler since only a staring lidar mode is used.

Thank you for pointing out the interesting question about wind evolution within the probe volume. We'd like to explain our consideration about this issue as following: In principle, wind evolution depends on the evolution time of turbulence (see equation (2) in the paper). Theoretically, Taylor's hypothesis is valid as the evolution time approaches zero. Although the probe volume seems to have a similar length as the distance between two adjacent range gates, the corresponding evolution time of both cases is totally different. The typical length of a laser pulse is in the order of magnitude of $10^{-7}$ s (e.g. 150-400 ns depending of devices). This temporal length corresponds to a spatial length in the order of magnitude of $10^1$ m considering the light speed. This

means it is not possible for lidar to distinguish the signals backscattered from the locations within this spatial range. This is the reason for a lidar having a probe volume. In this case, the evolution time of turbulence is in the order of magnitude of $10^{-7}$ s. But for the distance between two range gates, the evolution time corresponds to the travel time of the mean flow between two range gates, which is in the order of magnitude of $10^0 - 10^1$ s depending on wind speed and the distance between both range gates. Therefore, we think assuming Taylor's hypothesis within the probe volume is reasonable. We will add this explanation in the corresponding part of the paper to avoid misunderstanding.

Nevertheless, we agree that ignoring the spatial averaging effect of lidar is based on ideal assumptions (esp. the laser beam aligns with the mean direction, which is not always the case in practice) and is a kind of simplification. And in our paper, we also suggest the method developed by Schlipf et al., 2015 (Meteorologische Zeitschrift) for cases where the misalignment angle between the mean wind direction and the laser beam can be determined accurately (because this method requires the misalignment angle). In fact, our deduction follows the same approach but assumes no misalignment angle. The reason for that is, as discussed in the paper, determination of the misalignment angle is not always possible. For example, in our case, we can only use the wind direction measured on a met mast located at about 300 m away from the wind turbine to approximate the wind direction at the wind turbine. This approximation contains uncertainties. And sometimes even if it is possible to acquire the misalignment angle at turbines, the requirement for accuracy is very high because this variable is included in the most basic step — fitting the estimated coherence to the wind evolution model. Based on these considerations, we decided not to include the angle in the fitting but use it as a predictor, which makes this variable more standalone and prevents its errors from affecting "everything". And Gaussian process regression inherently assumes imperfect training data (containing noisy terms). Thus, it is better to keep uncertainties in predictors.
Moreover, in this research, our goal is to explore the potential of applying GPR in prediction of wind evolution. We wanted to examine it with different data avaiability. Thus, we could not include the misalignment angle in the fitting process, assuming this variable is always available. And we wanted to use a simple wind evolution model as a baseline case to demonstrate the prediction concept. As mentioned at the end of the paper, one could choose whichever wind evolution model suitable for own application scenario and obtain the corresponding parameterization model by following the methodology suggested in this work.

**Specific comments**

-Pg. 2, ln. 58: "adapted the Pielke and Panofsky's model by introducing a new parameter..." More accurately, the paper by Simley and Pao (2015) took the form of the coherence model for transverse and vertical separations suggested by the following paper, and adapted it to longitudinal coherence:
R. Thresher, W. Holley, C. Smith, N. Jafarey, and S.-R. Lin, "Modeling the response of wind turbines to atmospheric turbulence," Department of Mechanical Engineering, Oregon State University, RL0/2227-81/2, Corvallis, OR, Tech. Rep., Aug. 1981.
Thank you for your suggestion. This reference has now been cited to the corresponding text.

-Section 1: Introduction: Another very relevant paper should be discussed in the literature review section. The following paper discusses fitting lidar-measured coherence to the longitudinal coherence structure suggested in Simley and Pao (2015):
Analysis of wind coherence in the longitudinal direction using turbine mounted lidar
S.Davoust and D. von Terzi 2016 J. Phys.: Conf. Ser. 753 072005
Thank you for your suggestion. This paper is very relevant to our work. A short discussion about it has now been added to the literature review.

-Eq. 9: The last index "j" should be changed to "i".
Thank you for pointing out this mistake. It has now been corrected.

-Pg. 8, ln. 188: This paragraph and Fig. 4 are hard to follow. I would suggest labeling the angles the text refers to in the figure, and also provide some equations to support what you are trying to explain.
Thank you for your suggestion. We will improve the figure and the corresponding text.

-Pg. 8, ln. 196 - pg. 9, ln. 208: In this discussion, it is a little hard to tell if yaw misalignment is required by the coherence estimation method, or if it is optional. This becomes obvious later, but I think here it would be good to explain that the final model allows different combinations of predictors (including yaw misalignment) depending on availability.
Thank you for your suggestion. We will improve this part and mention that the misalignment angle could be used as a predictor if it is available.

-Section 2.5: Can you compare the predictors you are using to the predictors used in previous longitudinal coherence models in the literature (e.g., Kristensen, 1979; Simley and Pao, 2015)? It would be insightful to understand which new parameters are included in this study.
Thank you for your suggestion. We will add a simple comparison of predictors to this Section.

-Pg. 9, ln. 213: It would be good to define turbulence intensity here.
Thank you for your suggestion. The definition of turbulence intensity has been now added to this part.

-Pg. 10, ln. 233: "thus how likely or to what extent the local terrain changes" makes it sound like the terrain variations are the primary reason the coherence would

depend on "d". But even if the terrain stays the same, I would still think there could be a dependence on "d".

Thank you for pointing out this issue. Actually, we don't want to imply the terrain variations are the primary reason. But we have to admit the expression in that paragraph is not good enough and thus causes this misunderstanding. We will modify this paragraph to make it clearer.

-Pg. 10, ln. 234: "For prediction, it is not possible to obtain $\Delta t_{\text{maxcorr}}$." Why can't it be determined? It can be calculated just like all the other predictors, right?

Yes, you're right. Thank you for pointing out this mistake. We will include $\Delta t_{\text{maxcorr}}$ as a potential predictor and add a short discussion about selecting $\Delta t_{\text{maxcorr}}$ or $\Delta t_{\text{Taylor}}$ in Sect.5.2.

Also, on pg. 6, ln. 130, you say that the d/U approximation is not used in this study and $\Delta t_{\text{maxcorr}}$ is used. Which of these statements is right?

Thank you for pointing it out. This is a mistake in writing. The sentence in l.130 should be "... this approximation is not applied in estimation of coherence." When estimating the coherence, the velocity time series measured at the downstream is shifted by $\Delta t_{\text{maxcorr}}$ (l.377), while $\Delta t_{\text{Taylor}}$ could be used as a predictor considering it is easy to calculate.

-Pg. 10, ln. 251: (Chen, 2019). Can you describe how this current manuscript compares to the earlier work? Better yet would be to discuss this in the introduction.

Thank you for your suggestion. A brief introduction of the preliminary work has been added to the introduction.

-Pg. 11: Section: "Hyperparameters of GPR": In general, this section would be more clear if the specific variables discussed were connected to the wind evolution application.

This part aims to introduce the hyperparameters of GPR in a general sense so that readers could more or less understand the functions of these hyperparameters. In general, the logic of machine learning is to find statistical relationships among data (if we say it in a simple way). It is not possible to associate its algorithm to specific physical quantities except predictors (inputs of models) and target variables (outputs of models), which are introduced in other sections. Thus, we think it would be better to keep the explanation here abstract.

-Pg. 11, ln. 271: "where x is the input vector of different parameters" Can you provide an example of what these input parameters are in your application?
The input vector $x$ is a set of predictors for a single observation, which has the dimension of $D \times 1$. We've added the dimension of predictors $D$ to the text.

-Pg. 12, ln. 278: "where X is the aggregation of all input vectors." Can you explain in more detail? What are the dimensions of X? of parameters x of observations?
$X$ has the dimension of $D \times n$. $D$ is the number of predictors and $n$ is the number of observations. We've added the number of observations $n$ to the text.

-Eq. 19: I did not see these basis functions or the coefficients beta discussed any more in the manuscript. Can you describe how you chose the basis functions and how the coefficients were estimated? And how do these values affect the final estimate in Eq.22?
There are four types of basis functions provided in MATLAB: empty (assuming no basis function), constant, linear, and pure quadratic. We tried all of them and found there is not much difference. Finally, we chose the constant basis function because it is commonly used and takes a little less time. As far as we unterstand, the coefficients are estimated in the fitting of a GPR model by an optimizer like LBFGS-based quasi-Newton approximation to the Hessian. The algorithm is implemented in MATLAB.

-Pg. 12, ln. 286: Please describe in more detail why you are using a kernel function (I assume because then you don't need to actually define the functions "phi(x)").

Yes, exactly. As far as we understand, a kernel function can be used to replace the calculation of inner productions or covariance of the outputs of two functions. More details please refer to e.g. Rasmussen and Williams (2006) p.12 and p.14, Duvenaud (2014) Chapter 2, etc.

-Pg. 12, ln. 288-290: I don't think these sentences are needed, since in the next paragraph, you thoroughly introduce the ARD-SE kernel.

We introduce the ARD-SE kernel in detail because we have chosen this kernel in our study. But in general, kernel function is one of the hyperparameters which should be chosen according to data. Therefore, we consider it is necessary to give a short overview about kernel functions.

-Pg. 12, ln. 291: Why is the ARD-SE kernel chosen? And please provide a reference about this kernel.

We've tried both types of kernels and found out that applying the ARD kernels can obtain much better model performance than applying the kernels with same characteristic length scale, but the results of different ARD kernels, e.g. ARD-SE or ARD exponential, don't show much different. The reference for the ARD-SE kernel is cited in l.298. The same citation has been added to the first mention of the ARD-SE kernel.

-Eq. 21: Please define "D".

Thank you for your suggestion. $D$ is the dimension of predictors and has now been defined in the corresponding text.

-Pg. 12, ln. 296: "A relatively large length scale indicates a relatively small

variation along corresponding dimensions in the function" From Eq. 21, it seems more accurate to say that a large length scale relative to the amount of variability in the predictor indicates a smaller variation along the corresponding dimensions. For example, it seems the size of the length scale is only meaningful by itself if all of the predictors have been normalized to the same std. dev. Is this correct?

We think it depends on whether one decides to train the model with standardized data or not. In our study, the training data is standardized using z-scores, i.e. centering and scaling the data by its mean and standard deviation, respectively. This is explained in Sect.5.1 Model Training.

-Eq. 22: Can you state the difference between $X$ and $X_*$ here?

The meaning of * is stated below Eq.22. But if it is not clear, we can modify it.

Also, this equation seems to just be saying that the conditional distribution is normally distributed, so I don't think the right hand side of the equation adds anything. Perhaps it would be less confusing to just explain that the function values are estimated given input parameters $X_*$ by conditioning f on the training parameters and observations, X and y, as well as $X_*$: $f_*$| X, y, $X_*$.

We just intended to use Eq.(22) to explain that the predictive equation of GPR is a conditional distribution given the training data and the new input data, and this conditional distribution is normal distributed.

Finally, how are the estimates formed from the resulting distribution? Is the mean value used?

The mean value is the predicted value of the target variable, and the 95% confidence interval is determined by the variance of this distribution.

-Table 2: The lidar weighting function width (e.g., Full Width at Half Maximum) would be a relevant parameter to list in this table.

Thank you for your suggestion. Now, the information about the FWHM of both lidars has been added to the text as well as the table.

-Pg. 15, ln. 367: "The threshold for both are 6 m/s and 3 m/s". How are the thresholds used? For example, are these the thresholds in terms of deviation from the mean value of the three-data point window? Also, how is the standard deviation defined here? How many data points are used to calculate the std. dev.?
The range filter works in this way: 1) Calculate the value range within the window (range = max value - min value); 2) If the value range exceeds the preset threshold, this point will be filtered. The standard deviation is the standard deviation of all the values within the window. We noted the explanation about the filter is not well formulated. This will be improved.

-Pg. 18, ln. 436: "all the PDFs supported by MATLAB" Is there a particular MATLAB toolbox you are referring to here? Also please provide a reference for MATLAB.
We used a tool called *fitmethis* developed by Francisco de Castro. This tool requires Statistics and Machine Learning Toolbox of MATLAB. We apologize for missing the citation in the text! Citation: Francisco de Castro (2020). fitmethis (https://www.mathworks.com/matlabcentral/fileexchange/40167-fitmethis), MATLAB Central File Exchange. Retrieved Jan 13, 2020.

-Pg. 20, ln. 465: "all the fitted curves of the coherence are grouped together proves it is reasonable to model the wind evolution based on dimensionless frequency". Do you mean that they are grouped together at high frequencies ($f_{dless} > 0.1$)? Additionally,"proves" seems like a strong statement here. Maybe "suggests"?
Yes. And thank you for your suggestion for wording.

-Fig. 9: The caption should refer to the different subplots that are labeled (a-d).
Thank you for pointing out the missing information in the caption. This information has

now been added.

-Pg. 22, ln. 486: "all the potential predictors are included...to determine the characteristic length scale". Please describe how the training to find the length scales is performed.
As same as coefficients $\beta$, characteristic length scale(s) is estimated in the fitting of a GPR model by an optimizer like LBFGS-based quasi-Newton approximation to the Hessian.

-Pg. 22, ln. 489: "Figure 11 illustrates a comparison among the $\log(\sigma_m)$." As mentioned earlier, it doesn't seem fair to compare the $sigma_m$ magnitudes unless all of the predictor variables have been normalized to have the same std. dev. (or some other normalization). Is this done?
Yes. The training data is standardized by centering and scaling the data of each predictor by its mean and standard deviation, respectively, which gives the standard scores (also called z-scores) of the predictor data.

-Table 4: Please explain "standard deviation of observed responses" in more detail. It's not clear what the "observed responses" are.
The "observed response" generally means the model response observed from the data. In our case, it refers to the target variables, i.e. the fitted wind evolution model parameters $a$ and $b$. We will modify the corresponding text to make it clearer.

-Fig. 13: I'm not sure how to interpret this figure. Are there errors in the plots or the legend? The legend lists separate solid lines and dotted lines, but I don't see both in the plots. Do they perfectly overlap?
In this example, yes. We will modify the line colors or the line styles to make the curves distinguishable even though they overlap.
Additionally, the legend says that blue dotted is a fitted case and solid red is the

predicted case. But these two lines are very far apart,which does not support the claim
that the fitted and predicted curves are very close.

Different line styles indicate different types of curves: line with dot is fitted curve,
normal line is predicted curve, and dashed line is confidence interval. Different colors
indicate the results for different separations: blue is for $R_1 vs R_2$ and red is for $R_1 vs R_5$.
And there is indeed an error in the color of the predicted curves in the legend. It has
now been corrected.

Additionally, what is the significance of the particular period being shown here. Is this
one of the periods with the best match between the fitted and predicted coherence?
Or is it representative of a typical case?

We did not intend to show an example with the best match. We chose this data block
based on two principles: 1) data integrity and 2) representative wind statistics. In this
example, the lidar measured mean wind speed is 7.3–7.7 m/s and the lidar measured
turbulence intensity is 0.10–0.12, for different range gates. These values appeared
frequently in the selected period according to Fig.A1. Hence, we think this example
is representative for the data involved in this study. And we decided to choose a data
block as an example because we thought neither the coherence itself nor its prediction
are intuitive (esp. $a$ and $b$ are predicted respectively but must be combined to give the
modeled coherence). We noted a mistake here: the date should be 07 Dec. 2013.
Figure 6, 8, and 13 are all plotted with the same data block. The date has now been
corrected.

-Pg. 26, ln. 556: "RĚĘ2 at least over 0.65." What is the significance of 0.65 as
an indication that the prediction accuracy is "satisfactory"?

We have to admit that the wording here is not very appropriate. We will modify the
inappropriate expressions in the paper.

-Pg. 26, ln. 573: "no obvious relevance between the error and values of any of
the predictors is indicated in both figures." I don't quite agree. I think there are some

interesting trends, like in Fig. 14 (b), the RMS prediction error decreases as $sigma_l$ increases. Trends can also be observed in Fig. 15 (b) and (f).

Thank you for your suggestion. That is a very interesting finding. We will think about this part again.

-Pg. 29, ln. 612: "capable of achieving a parameterization model with sufficient accuracy for the prediction of wind evolution." This seems like a very strong statement to make. Please provide some more context for this statement. How is "sufficient accuracy" determined?

We noted the wording here is not very appropriate. We will modify this statement to make it more objective.

-Pg. 29, ln. 616: "methods to improve the estimation of the coherence and the wind statistics are desired." What are some of the shortcomings of your current approaches that you think could be improved?

For example, if the direction misalignment could be determined in a reliable way by e.g. using a lidar with multiple beams, it might be possible to use a more sophisticate wind evolution model to analytically account for more complicated effects. moreover, methods to improve the accuracy of turbulence intensity or high order wind statistics derived from lidar data would be of great interest (if it is possible).

-Fig. A1-A5: I think if there are appendix figures, they should be in a labeled appendix section.

Thank you for your suggestion. An appendix section has been added.

-Pg. 30, ln. 623: Is the current computational time acceptable for real-time applications?

According to our study, it is possible to do real-time prediction but not real-time model training.

-Section 6: If possible, it would be nice to hear your thoughts on whether the chosen coherence formula structure (Eq.5) could be improved. In other words, the paper mostly focuses on how to estimate the a and b parameters, assuming Eq. 5 is the right model. But is this model good enough? For example, Simley and Pao (2015) show that this kind of model did not fit stable atmospheric conditions very well.

We initially also wanted to study the influence of atmospheric stability on wind evolution. However, we found that the stability of the selected period of LidarComplex (where sonic data is available) happened to be mostly neutral. Therefore, unfortunately, we could not get any conclusions related to atmospheric stability.

According to our experience, we think this coherence formula structure is reasonable and can fit the estimated coherence well in most cases as long as noises in the high frequency range (if exist, e.g. the noises caused by motion of the nacelle) can be properly filtered. And we found that in comparison to fitting all coherence measured simultaneously (between different separations) at once (as Simley and Pao (2015) did), fitting the coherence individually could improve the fitting quality because this enable each coherence to find its best-fit parameters.

**Response to comments of Mark Kelly**

**General comments**

This work examines evolution of advected turbulence, in terms of spectral coherence,with the motivation of wind turbine control assisted by inflow measured by lidar. The topic is quite relevant to wind energy, and fits well with the journal (WES).

There is some interesting content and potentially useful results, with the use of GPR and Bayesian inference being quite nice.

We would like to thank the referee for the interest in this research and the positive feedback on our methodology.

Unfortunately the paper appears to be somewhat 'unfinished'; perhaps it is also due to the lack of English fluency or preparation time. For example, the abstract has simply copy-pasted a few sentences from the paper, and repeats in a cumbersome way:'This paper aims to achieve parameterization model for the wind evolution model to predict the wind evolution model parameters'. The paper needs to be proofread by somebody with English fluency, at any rate.

The abstract does not clearly provide an idea of the work done, and while the text has more detail, it is not clear throughout; I am not sure that readers could repeat what has been done.

Thank you for your suggestion. We will improve the presentation of the whole paper, especially the abstract. The revised manuscript will be proofread before submission.

More importantly, there are inconsistencies that have not been considered, and should be addressed/rectified; perhaps most significant are the form itself chosen for coherence (see derivations below in Specific comments), and the use of Taylor's hypothesis for some (but not other) parts of the model/parameterization.

Thank you for your suggestion. Please find below our responses to the related comments.

There are a number of details and also explanations which are missing, but which could hopefully be included, to make the work publishable. The results/performance are a bit overstated (in English), but this is not needed, as the numberical results presented tell the story less subjectively—and are good to share with the wind energy community, provided that they are given with sufficient detail, replicability, and consideration.

Thank you for your suggestion. We will include the missing information and modify the wording of the paper as suggested by the reviewer.

**Specific comments**

Line 15 and elsewhere later in the paper: while the authors define 'wind evolution' as squared coherence, they imprecisely define such (e.g."Coherence is a dimensionless statistic in the frequency domain").
When we reviewed the relevant literature (see the following references as examples), we noted that squared coherence is commonly used instead of magnitude coherence (Although it is not clearly stated as "squared coherence" in the text, but "coherence", the formulas show squared coherence.). Hence, we decided to follow their definition to keep our work consistent with the previous studies. But we agree that the expression "Coherence is a dimensionless statistic..." is not precise enough, and thus we've now rephrased the corresponding text.
Davenport, A. G. (1961). The spectrum of horizontal gustiness near the ground in high winds. Quarterly Journal of the Royal Meteorological Society, 87(372), 194–211. https://doi.org/10.1002/qj.49708737208
Panofsky, H. A., & McCormick, R. A. (1954). Properties of spectra of atmospheric turbulence at 100 metres. Quarterly Journal of the Royal Meteorological Society, 80(346), 546–564. https://doi.org/10.1002/qj.49708034604

Specifically, this should be written as temporal coherence (time shift, frequency domain), in contrast with spatial (wavenumber spectra) coherence. Further, coherence does not describe the "correlation between two signals", but rather the correlation between spectral components of two signals.

Thank you for your suggestion. This paragraph has now been rephrased accordingly.

Do lines 19–20 not imply that use of Taylor's hypothesis means ignoring wind evolution? This may be relevant, for consistency later (line 235).

Yes. We think that the degree of wind evolution should depend on the evolution time, so Taylor's hypothesis should be valid when the evolution time approaches to zero. We noted that the text in l.232-235 is not well formulated, and thus we've now rephrased that part.

Line 35: The statement "dependence of coherence on separation and atmospheric stability was not adequately researched" lacks reference and/or explanation. It was not adequate, according to whom, or how?

The content in l.29-35 is a brief introduction of the work of Panofsky and Mizuno (1975). This statement is specifically for that study. To avoid misunderstanding, we've added "in that study" at the end of the sentence.

Line 36–38: You write "The longitudinal coherence differs from the lateral and vertical coherence because the former measures the correlation with respect to time lag while the latter with respect to spatial separation." This is not correct: longitudinal, lateral, and vertical coherences all depend on f, based on integration over time lags; they are depending on spatial separations in the respective directions. For the longitudinal coherence to give spectral correlations 'with respect to time lag' $\Delta t$, then the longitudinal separation is related to $\Delta t$ in some way, though you have stated before this point that you are not using Taylor's hypothesis.

We have to admit that this sentence is not well formulated, and maybe the use of "with

respect to" is not proper here. Of course, coherence is a function of frequency when it is estimated from time series data. What we wanted to express is that the lateral and vertical coherence depends on spatial separation in the respective directions, while the longitudinal coherence is coupled with the time-dependent variation of turbulence because the evolving eddies are moving in the longitudinal direction with the mean flow. So, the longitudinal separation is related to the travel time (corresponding to the evolution time) $\Delta t$. And we think this is independent of application of Taylor's hypothesis, but Taylor's hypothesis provides an approximation ($x/U$) for the travel time.

Line 112/equation 2: how is this a function of frequency (f)?I.e., include the f-dependence on LHS, and also within $\tau$ on RHS.
Thank you for your suggestion. We've now included the f-dependence on LHS. But regarding the f-dependence of $\tau$ on RHS, because it is derived from Eq.(3) and Eq.(4), which is "unknown" for Eq.(2), we would prefer not to include it in Eq.(2).

There appears to be incompatibility between Eqs. 3–4 and Eq. 5; in particular (5) is missing $\sigma$ and $U$.
We have to admit that here we may have written too briefly. The sentence "Combining Eq. (2)–(4) and introducing the second parameter in the model, as inspired by Simley's model (2015a)..." means that combining Eq. (2)–(4) gives the formula like Pielke & Panofsky (1970), and then we imitate the formula of Simley's model (2015a) to introduce a second parameter in the model. We initially wanted to avoid introducing too many formulas for brevity. But if that would cause confusion, we will add related formulas and explanations.
Further, in Section 4.2 and Fig. 10, you analyze the behavior of $a$ with $\Delta t_{\mathrm{maxcorr}}$ (stating $a \propto \Delta t_{\mathrm{maxcorr}}^{-0.49}$), but do not consider that the equations already imply a $\Delta t$ dependence. That is, eqns. 2 and 5 give

$$C\Delta t/\tau = \sqrt{(af\Delta t)^2 + b^2};\tag{2}$$

however (3)–(4) with (2) give

$$(CI_u f \Delta t)^2 = (af\Delta t)^2 + b^2 \tag{3}$$

where the turbulence intensity is defined $I_u \equiv \sigma/U$. Thus one sees that

$$a = \sqrt{(CI_u)^2 - (b/f_{\text{dless}})^2} = CI_u\sqrt{1 - (\frac{b}{C\Delta t/\tau})^2}. \tag{4}$$

In the limit of high dimensionless frequency or large turbulence intensity, i.e. without the offset $b$, then $a \to CI_u$ like Pielke & Panofsky (1970). But in the limit of small turbulence intensity or low dimensionless frequency (small $f\Delta t$), i.e. a large offset, then we see $a$ become imaginary, implying a (nonphysical) coherence oscillating with $\Delta t$ or $\Delta x$.

Thank you for pointing out this interesting question. In fact, we did notice it and have examined if the $\Delta t$ dependence of $a$ is due to the $\Delta t$ introduced in $f_{\text{dless}}$. As presented in the paper, $a \propto \Delta t_{\text{maxcorr}}^{-0.49}$. If this dependence originally did not exist but entirely comes from $\Delta t$ in $f_{\text{dless}}$, the exponent of $t_{\text{maxcorr}}$ should be around -1, and it could be canceled out with the $\Delta t$ in $f_{\text{dless}}$. Moreover, we also tried to fit the estimated coherence to the coherence model dependent of frequency or wavenumber (without introducing $\Delta t$ in the formula), and we found that $a' \propto \Delta t_{\text{maxcorr}}^{0.51}$ (here use $a'$ to distinguish from the former $a$). Hence, we think that $a$ (or $a'$) originally has a $\Delta t$ dependence, and the $\Delta t$ in $f_{\text{dless}}$ just changes the exponent of $\Delta t$.

The form (5) is the same as that of (4) in Simley & Pao (2015), with $d_l$ in the latter replaced here by $U\Delta t$, and $b$ here replacing their $abd_l$; this should be noted, and the text is not quite clear nor correct following your Eqn. 5. You do note the reason for keeping $\Delta t$ (instead of using $\Delta t = d_l/U$), but why did you drop the spatial separation dependence ($d_l$) from the '$b$' part of the Simley Pao (2015) expression? From your logic for the '$a$' term, then instead of just b you would have $bd_l$ (but not $bU\Delta t$).

The reasons for using $b$ to replace their $ab'd_l$ (use $b'$ to distinguish from our $b$) are: 1)

[Figure]

In terms of curve fitting, $ab'$ is essentially the fitted term, and thus $b'$ shows a strong dependence on $a$, which is generally undesirable for machine learning methods. 2) Using $ab'd_l$ (or $b''d_l$) implies that the unit of $b'$ is $m^{-1}$, while $a$ is dimensionless. We wanted to make both dimensionless to keep consistent. 3) We did try $b''d_l$, but we found that $d_l$ is still an important predictor for $b''$, which indicates $b''$ still depending on $d_l$. Then, it is not necessary to assume $b''d_l$, but simply use $b$ and take $d_l$ as a predictor.

In lines 145-9: your text is a bit imprecise here—the sonic anemometer has a measuring volume as well (not a point), it is just much smaller than the lidar's. Also, among the reasons why longitudinal coherence from lidar deviates from that calculated via sonic anemometers, one key possibility is missing: the validity of Taylor's hypothesis.
Thank you for your suggestion. We will modify the text accordingly.

Line 153: what do you mean by "complete coherence curve"?
In that context, a "complete coherence curve" means the coherence can more or less cover the range from the highest coherence (e.g. 0.9 – 1.0, sometimes could be lower depending on spatial separation) to lowest coherence (e.g. 0 – 0.1). We've now improved the expression to make it clearer.

Lines 157–164: please include references, as this is not original.
Thank you for your suggestion. The relevant reference has now been cited to the corresponding text.

Line 184: neglect of the spatial averaging effect and w(x) in $\gamma_{i,j}^2$ also demands use of Taylor's hypothesis. This should be noted (along with its potential inconsistency).
Here, Taylor's hypothesis is applied within the lidar probe volume. The lidar probe volume is resulted from the length of laser pulses, with typical length in the order

of magnitude of $10^{-7}$ s (e.g. 150-400 ns depending of devices). Considering the light speed, this temporal length corresponds to a spatial length in the order of magnitude of $10^1 - 10^2$ m. Within this range, the exact locations from which the signals are backscattered can not be distinguished. Because wind evolution depends on the evolution time of turbulence (see equation (2) in the paper). In this case, the corresponding evolution time is in the order of magnitude of $10^{-7}$ s. Therefore, wind evolution can be neglected within the probe volume. Referee 2 asked a similar question. Please find in p.11 the discussion if you are interested in more details. And we noted a mistake: in fact, it is not necessary to specifically assume t = x/U. There must be a correspondence between t and x as long as wind flows in x direction.

Lines 193–5: the sentence "To retrieve the longitudinal coherence in this case, the above discussed spatial averaging effect must be coupled to a specific turbulence model (Schlipf, 2015; Mann et al., 2009), and thus the wind evolution model is included in the final model implicitly" does not quite make sense. Could you clarify?

Schlipf et al. (2015) suggested an approach to consider different effects of lidars when detecting wind evolution. Here, we briefly mention the explicit expression of the horizontal coherence deduced in that study, based on the assumption of lidar point-measurement for simplification:

$$\gamma_{ij,losP} = \frac{\cos^2(\alpha_H)\gamma_{ij,ux}\gamma_{ij,uy}S_{ii,u}}{\cos^2(\alpha_H)S_{ii,u} + \sin^2(\alpha_H)S_{ii,v}}, \tag{5}$$

where $\gamma_{ij,losP}$ is the horizontal coherence of lidar point-measurements, $\gamma_{ij,ux}$ and $\gamma_{ij,uy}$ are the longitudinal and lateral coherence of the u-component, $S_{ii,u}$ and $S_{ii,v}$ are the auto-spectra of u and v components, $\alpha_H$ is the misalignment angle. From this equation, one can see the determination of the longitudinal coherence $\gamma_{ij,ux}$ is only possible given a specific turbulence model (knowing $S_{ii,u}$, $S_{ii,v}$ and $\gamma_{ij,uy}$) and knowing the misalignment angle $\alpha_H$.

The volume averaging effect of lidar is then taken into account with a Riemann sum

based on the theoretical consideration for the case of lidar point-measurement, and thus the equation is too complex to be explicitly expressed.
More details please see:
Schlipf, D., Haizmann, F., Cosack, N., Siebers, T., and Cheng, P. W.: Detection of Wind Evolution and Lidar Trajectory Optimization for Lidar-Assisted Wind Turbine Control, Meteorologische Zeitschrift, 24, 565–579,
https://doi.org/10.1127/metz/2015/0634, 2015.

Line 217: by "its definition", do you mean the definition analogous to (13), where the timelags is replaced by spatial separation r?
Yes, exactly.

Lines 217–19: If you say $L_{int} = T_{int}U$, then aren't you just using U as a potential predictor somehow?
Yes. $U$ itself is also included as a predictor.
Also, isn't this inconsistent with the previous section, where you state that Taylor's hypothesis is to be avoided? Or, is Taylor's hypothesis avoided only for certain aspects? Please clarify.
Unfortunately, for calculation of $L_{int}$ from measured data, we have no alternatives except this approximation. But in this part, we only discuss in general what could be the possible predictors. A prediction selection is done to select proper predictors (discussion see Sect.5.2).

Line 223: The statement "atmospheric stability represents a global effect of the boundary layer on the wind field" is not quite correct. From what you measure, or via M-O similarity, it is a 'global' effect from the surface, and potentially only through part of the ABL (sometimes not even above the surface layer in stable conditions).
Thank you for your suggestion. We will modify the text accordingly.

Line 234: "So as the travel time Δt." is not a sentence. What are you trying to convey here?
Thank you for pointing out this mistake in writing. We've now modified this paragraph to make it clearer.

Line 235: So you are meaning that distance d is used instead as a predictor.
Yes, exactly.

Line 250: by "performs the best", perhaps you should use 'performs well'—unless you explain what 'best' means (i.e. what other models).
Thank you for your suggestion. In a preliminary study, we explored different machine learning algorithms on a simple level, including stepwise linear regression, regression tree, support vector machine regression, and Gaussian process regression, and we found GPR performs the best among these methods. Now, we've added these details and moved this part to the introduction.

Line 257: The phrase "underlying functions of the data" is not clear. Do you mean behavior conditioned on other variables, or relation to other variables?
Here, we mean "one needs to initially guess what type of function(s) could exist among the relevant variables before choosing a specific regression model." We will rephrase this part to make it clearer.

Table 1/ line 255+ : Is it even possible to use the fourth or even third moment, given the large sampling uncertainty involved for highter-order moments? Please see and reference e.g. Lenschow, Mann Kristensen (1994) and Ch.2 of Wyngaard's text-book (2010), to understand and defend use of $\mu 4$—let alone $\mu 3$.
Thank you for the recommended references. We agree that the third and fourth moment determined from measured data would contain a large uncertainty. But our approach is first to find all variables which can be obtained from measured data and

then to use feature selection to select suitable variables as predictors for the GPR models. Feature selection can detect the statistical correlation between predictors and the target variable. Although some variables could contain uncertainties, these variables could still be useful for the prediction of GPR models as long as they have a strong statistical correlation with the target variable. From this perspective, machine learning algorithms generally have an error tolerance for data (of course, the more accurate the data is, the more accurate the prediction could be), which is also one of the advantages of machine learning.

Line 284: To be clear and consistent, can you not specify that $\beta$ is a weight, and the 'basis function' h(x) maps the means into the new space?
$\beta$ can also be understood as the weight vector of $h(x)$. But we've defined $w$ as a weight vector before, we wanted to avoid using the same word in case reader might confuse these two different processes: $\mathbf{h}(x)^\top \beta$ is used to model the mean function $m(x)$ of $g(x)$, and $g(\mathbf{x}) \sim \mathcal{GP}(m(x), k(x, x'))$ ($k(x, x')$ is a kernel); $\phi(x)^\top w$ is used to find the linear model of $f(x)$ in a higher dimensional space. Theoretically, for any function $f(x)$, it is always possible to find a linear model equivalent to $f(x)$ in a higher dimensional space. For example, for a quadratic function

$$f(x) = ax^2 + bx + c, \tag{6}$$

if define $p = x^2$, $q = x$ and $r = 1$, then

$$f(x) = g(p, q, r) = ap + bq + cr, \tag{7}$$

which is a linear model of $f(x)$ in the three-dimensional space of $(p, q, r)$. In the algorithm of GPR, $\phi(x)$ is not explicitly defined. The mapping is done through a kernel, which is the so-called kernel trick.
What is meant by 'Basis function is one of the hyperparameters'? I.e., how is a function a parameter, or is h(x) already assumed to have some form, possibly related

to the $\varphi$(x) forms?
A hyperparameter, different from a parameter, is not necessarily a value (or values), but more like a setting adjusting model behavior (specifically related to machine learning). There are four types of basis functions provided in MATLAB: empty (assuming no basis function), constant, linear, and pure quadratic.

lines 289–296:$\sigma_m$ is not a 'length' in the physical sense; it has units of whatever $x_m$ has. Thus it is a characteristic magnitude for the predictor having index m.
In the context of machine learning, this term is defined as "characteristic length scale". Please see references like:
Rasmussen, C. E. and Williams, C. K. I.: Gaussian processes for machine learning, Adaptive computation and machine learning, MIT, Cambridge, Mass. and London, 2006.
Duvenaud, D.: Automatic model construction with Gaussian processes, Apollo - university of cambridge repository, https://doi.org/10.17863/CAM.14087, 2014.

lines 299–302: To be explicit, the RHS of (22) does not contain a 'conditioning bar'. I.e.to help the reader and match the text, show in the math that the joint Gaussian prior is conditioned on $X_*$(is the eqn. correct?). Most readers will not have read Duvenaud's PhD thesis, so it is useful to help them understand.
We just intended to use Eq.(22) to explain that the predictive equation of GPR is a conditional distribution given the training data and the new input data, and this conditional distribution is normal distributed. We did not include the equations of $\overline{f}_*$ and $cov(f_*)$ because both are very complicated. We don't think it is necessary for this paper to go so deep into mathematics.

Lines 307–311: why k=5? If it is due to needing a large enough sample for verification, then this should be stated.
Theoretically, k can be any integer between two and the number of observations

(This is a special case which is called "Leave-one-out".). When k is very small, the sample size of training data ($\frac{k-1}{k}$ of the total observations) could be not large enough. However, the training process must be repeated k times. So, when k is too large, the training could take very long time. Therefore, k = 5-10 is common used in machine learning. We will add this explanation in the corresponding text.

Line 335: was the lidar on the nacelle, at what height?
At 95 m. We have also noted some information about the measurements is only listed in Table 2 but missing in the text. We will improve this part.

Line 350: please be more clear and specific, and also include references.
The research project ParkCast is an ongoing project led by our institute (Stuttgart Wind Energy, University of Stuttgart). Because no publications related to this project have been published so far, we cannot cite any references. However, we have communicated the project-related information with our colleague in charge of this project to ensure its correctness.

Line 362–366: Why are two different filtering types used?
Because the two lidars are different, and one of them is a long range lidar (the max range was set as 990 m for the data used in this paper). For a long range lidar, the backscattered signals from distant range gates could be very weak, and thus the CNR values could be low although the measured wind speed is plausible. In this case, filtering the data based on CNR values is not a good idea. Würth et al. (2018) suggested an approach to filter the data with a range filter, which can keep more valid data compared to the CNR filter.
Würth, I., Ellinghaus, S., Wigger, M., Niemeier, M. J., Clifton, A., and Cheng, P. W.: Forecasting wind ramps: Can long-range lidar increase accuracy?, Journal of Physics: Conference Series, 1102, 012 013, https://doi.org/10.1088/1742-6596/1102/1/012013, 2018

State why -24dB for CNR; include reference.
We determined the threshold for CNR by checking the plot of CNR values and wind
speed. The CNR threshold could be various under different measurement conditions.

Line 375: Using "C2N" to symbolically write 'N choose 2' unique pairs, is not
standard practice. You can write(N2)or equivalently N(N−1)/2.
Thank you for your suggestion. This has now been corrected.

Line 479–480: need citation for Levenberg-Marquardt algorithm.
Thank you for your suggestion. The corresponding citations have now been added to
the text here.

Fig.9 caption: mention which plots belong to which campaign.
Thank you for pointing out the missing information. The campaign names have been
added to the caption.

Fig.11 / §5.1 : why not plot $\sigma_m^{-2}$? This is what is actually used in the ARD-SE
kernel shown in eqn.21, and its behavior more clearly demonstrates relevance.
We can also plot $\sigma_m^{-2}$, but in principle, it will not be different because
$log(\sigma_m^{-2}) = -2log(\sigma_m)$. The plots will just be flipped up side down, with the y-
axis (in log) scaled by two. The benefit of using $log(\sigma_m)$ is that it shows the order of
magnitude of $\sigma_m$ directly.

Line 489–90: you state "predictors are selected according to different preset lim-
its of the $log(\sigma_m)$ considering different cases of application or data availability", but
what are these preset limits?
The limits are listed in Table 5 and discussed in Sect.5.2. We will modify the text to
make it clearer.

Line 494 / Table 4: how/why did you choose the initial $\sigma_m = 10$?
The initial values of $\sigma_m$ are randomly set. They will be estimated from training data and the GPR algorithm just need some initial values to start the training process.

Line 509: I am not sure that R2 of 0.65 is "satisfactory"; perhaps this could be just written in terms of R2 and RMSE without the subjective claim.
Thank you for your suggestion for wording. We will modify it (and other similar expressions).
Also, "all situations" is not quite consistent with just the recommended cases (i.e. it implies all cases).
"All situations" here refer to l.501: "two different situations of data availability are considered: only using variables calculated with lidar data as predictors (in both of LidarComplex and ParkCast available) and only using variables calculated with sonic data (only in LidarComplex available)." We intended to distinguish "situation" and "case", and to use "situation" to indicate different data availability. But maybe "situation" is not a suitable word. We will consider the wording here again.

Lines 519–520: This is a good point, and it would be useful to repeat this earlier, when introducing the potential predictor variables because some of them appear redundant.
It is mentioned in the penultimate paragraph in Sect.2.5, but maybe is is not clear enough. We will indicate this point more clearly.

Lines 524–527: It appears that you are conflating two things here, one of which you are missing—applicability of Taylor's hypothesis will also affect L compared to T via U, whereas this is not the case for the usage of U to 'convert' $\sigma_u$ to IT.
What we wanted to discuss here is the possible difference for the model between using the variables directly acquired from measured data like U and $\sigma$ and the variables derived from the other variables like $I_T = \sigma/U$. In principle, $I_T$ can be regarded as a

function of $\sigma$ and $U$, and thus it is probably "useless" for the model. But we agree that we missed the point — the approximation of $L$ by $TU$ assuming Taylor's hypothesis is less accurate and thus probably less preferred by the model. Thank you for your suggestion.

Lines 532–534: perhaps $\mu_3$ or $\mu_4$ could help prediction; but to be responsible, one needs to mention that [1] the uncertainty in these quantities are very large (Lenschowet al. 1994 reference), and [2] lidar may not be able to consistently measure these.Further, these higher-order moments are likely more affected by your filtering. Thank you for your suggestion. That is a very good point.

Lines 537–539: this is likely due to the implicit co-dependence I derived above, i.e.a is a function of CIT and b/fdless. Your finding confirms also the need/utility to consider the behavior of the parameters involved. As mentioned above in the response to the comment on Eq.(5), we did consider this issue. The ideal case is $a$ and $b$ are completely uncorrelated, but the model form determines that the correlation between a and b cannot be completely eliminated. Indeed, we have reduced their dependence by adapting the form of the offset parameter $b$.

Lines 542–554: what about cross-comparison using the sonic? Were the wind directions such that the sonic (at 270m upstream) could be compared to the lidar (e.g. at 163.5m upstrean)? We are not quite sure what kind of comparison this "cross-comparison" refers to. We assume that it means the coherence estimated from the sonic data and the data measured at the farthest range gate of the lidar when the wind direction is aligned with the line between the met mast and the wind turbine. Firstly, to estimate coherence, this wind direction must exist long enough, which is less likely to happen in practice. Secondly, because the sampling rate of the ultrasonic anemometer is much higher than the lidar, the sampling rate of the sonic data must be artificially reduced, by e.g.

averaging, to match the sampling rate of the lidar data. But this leads to the fact that estimating coherence between the sonic data and the lidar data does not bring more benefits than estimating coherence between lidar data measured at different range gates.

Line 555-6: isn't this 'satisfactory' $R^2 \geq 0.65$ only true for certain cases and variables?
This statement is specific for the cases using lidar data. Although we discussed different variable combinations and compared the performance of the corresponding models, all the variables are essentially derived from line-of-sight wind speed. In other words, as long as there is a lidar measuring wind speed properly, one would be able to derive all the discussed variables. The question is which of them are necessary. In general, we want as many as necessary and as few as possible.

Figure 14 / Line 560 and afterward: these plots do not responsibly/transparently show prediction error, as they don't give an idea of the magnitude of a. You should plot percentage error or similar; given that a can be small depending on band IT (as derived above), the plotted differences in a might be relatively significant.
We have been thinking for a long time if it would be better to show relative error or absolute error. Our concern is that values of $a$ and $b$ are very abstract and completely not intuitive. In fact, the shape and position of the predicted coherence determined by both parameters together is the final prediction goal. And the prediction error is the shift of the predicted curve from its estimated curve due to the error of $a$ and $b$.
Assume a prediction error for $a$ is 0.5. Its relative error will be 50% given $a = 1$ and 25% given $a = 2$. But is the difference between the curve of $a = 1$ and $a = 1.5$ somehow related to 50% of something? Or is the difference between the curve of $a = 1$ and $a = 1.5$ somehow as twice large as the difference between the curve of $a = 2$ and $a = 2.5$ ? It is not the case. In the end, we chose to show the absolute error because knowing the absolute error, one could more or less imagine how much

the predicted curve would be shifted due to the error by observing Fig.3, which is not possible for showing the relative error.

Moreover, we are considering to plot two additional curves on the plot of example predicted curves (Fig.13) to indicate the range of predicted curves due to the RMSE of both parameters. That could hopefully give an intuitive feeling for the prediction error of the coherence curve.

Lines 574–5: if the whiskers are large because of sample size, then why not (also) account for this via $\sqrt{n}$ ?

Here, what we wanted to express is that large whiskers indicate large variances of predicted errors. The reason for that could be insufficient training data in the corresponding value range resulting in less accurate predictions for that value range by the model. We noted the expression here is not clear enough and will improve it.

Line 576: The claim "it is proven that the Gaussian process regression is capable of achieving an accurate parameterization model" is an overstatement. It is DEMONSTRATED(not proven) that the GPR was able to predict two coherence model parameters with an $R^2 \geq$ in chosen cases (not simply 'accurate').

Thank you for your suggestion for wording. We will rephrase this part.

**Technical corrections**

There are many English usage/grammatical corrections and suggestions, which are included in the attached annotated PDF-file. I thus only include a sample of them here in this list. The sentence structure and writing is unclear or ambiguous in numerous places; the paper really should be reviewed and edited by somebody with adequate fluency.

Thank you for your suggestion. The revised manuscript will be proofread before submission.

I list some of the specific corrections below, but since there are >300, after the first few I include the line numbers, which refer to the annotated attached PDF. After page 16 I did not correct much English; this is left to the authors for the next draft.

• Abstract/line 1: One (generally) shouldn't copy sentences from the introduction into the abstract (further, this first sentence is a definition); 'turbulence' should be 'turbulent'; pluralize 'structure'; delete 'of the eddies'; replace 'while the eddies' with e.g. 'as they'; replace 'by the main flow over' with 'through'.
• L.2–4: Remove 'the'; change 'because' to ':'; remove 'only'...see annotated PDF for more details.
• L.5–7: These 2 sentences are quite unwieldy (cumbersome) and also somewhat tautological—particularly for an abstract, also with repeated phrases that need to be reduced/condensed. Please correct the English usage here.
• L.12–13: First sentence can be corrected from "Wind evolution refers to the physical phenomenon that the turbulence structure of the eddies changes over time while the eddies are advected by the main flow over space." to something like 'WIND EVOLUTION' REFERS TO THE PHYSICAL PHENOMENON OF TURBULENCE STRUCTURES (EDDIES) CHANGING OVER TIME, WHILE THE EDDIES ARE ADVECTED THROUGH SPACE BY THE MEAN FLOW.
• L.13-15: change "The mathematical" to 'A common statistical'; delete 'usually'; 'hereinafter for brefity, also' should be 'hereafter'; delete 'two time series data sets of the' and instead add 'measured at two different locations,' after 'velocity'; change 'with certain time shift' to 'calculated over varying time shifts'.
• L.17–19
• L.22
• L.25–27
• L.29
• L.32: run-on sentence; use parentheses as noted

[Figure]

• L.33–34
• L.36–38
• L.40, L.42–44
• L.47–48
• L.52 use of definite articles and parentheses
• L.55–59, L.61
• L.63 delete 'model'
• L.65, 67
• L.73–74
• L.79–81: delete a number of redundant words, add punctuation as noted
• L.122–123
• L.136
• L.151 Capitalize 'Oppenheim', here and elsewhere.
• L.156
• L.169
• L.177: have already introduced U as mean wind speed (delete here).
• Page 8: L.186–7; 189; 196–9; 201–2
• Page 9: L.204; 210; 212; 220–2;
• L.224 and elsewhere: not 'Monin-Obukhov length', just use 'Obukhov length'
• Page 10: L.225–8; 233; 250–7
• Page 11: L.258–9; 261; 264; 269; 272
• Page 12: L.275–8; 288–9; 292; 295–6; 302; 304–5
• Page 13: L.308
• Page 14: L.335; 339; 346–7; 349–353; 363–5
• Page 15: L.367
• Page 20: L.470; 479
• Page 24, Table 5: Taylor is italicized under case 6, but should be Roman font.
• Page 25: L.537; 550

Please also note the supplement to this comment: https://www.wind-energ-sci-discuss.net/wes-2020-50/wes-2020-50-RC3-supplement.pdf

We highly appreciate these suggestions for technical corrections. All of the comments have been considered, and corresponding corrections have been made. We'd like to thank the reviewer again for taking valuable time to help us to improve this manuscript. The revised manuscript will be proofread before submission.

Please also note the supplement to this comment:
https://wes.copernicus.org/preprints/wes-2020-50/wes-2020-50-AC1-supplement.pdf

[Figure]

**Supplement:**

---

## Author Response (AR2)

**Author's response for minor revision**
**for the manuscript: Parameterization of Wind**
**Evolution using Lidar**

Yiyin Chen, David Schlipf, and Po Wen Cheng

September 2020

We would like to thank Mark Kelly again for reading our revised manuscript and we highly appreciate his comments!

Please find below our responses to the comments. The reviewer comments are repeated in black text, and our responses are provided in blue text.

**Response to comments of Mark Kelly**

This draft is much improved, after many changes addressing the 3 reviewers' comments. A few comments remain:

l.39: mention that Ri is a stability measure, somehow–this is the first time that stability comes up.
Thank you for your suggestion. A short additional note has now been added to the corresponding text.

l.47 replace "adequately" with perhaps 'thoroughly' and/or 'directly'.
Thank you for your suggestion. This has now been modified.

Regarding this reviewer's comment about the $\Delta t$ dependence already inherent in a, from the equations you use, where the review includes a derivation of such dependence: your reply unfortunately appears to indicate some confusion around the implied dependence. You respond with an explanation that "removing" the $\Delta t$ in dimensionless frequency gives a new dependence of $(\Delta t)^{0.51}$. But the implied dependence is still like eqn.(4) in this reviewer's previous review; shouldn't such "removal" give a dependence that looks like $(\Delta t) * (\Delta t)^{-0.49} = (\Delta t)^{0.51}$, according to the review's eqn.(4)?. If you have an implied dependence which is clear, then why "guess" at the exponent later in section 4.2? At very least, a clearer restatement of your response seems to be needed, and included in the article.
**1) Regarding the derived $\Delta t$ dependence in $a$ from the reviewer:**
To explain it more clearly, we first repeat all relevant equations as follows:

i. The initial assumption about the possible form of wind evolution model based on physical considerations in Ropelewski et al. (1973):

$$\gamma^2_{\text{model}}(f) = \exp\left(-C \cdot \frac{\Delta t}{\tau}\right). \tag{1}$$

ii. Our wind evolution model, modified from the model suggested by Simley and Pao (2015):

$$\gamma^2_{\text{model}}(f) = \exp\left(-\sqrt{a^2 \cdot (f \cdot \Delta t)^2 + b^2}\right). \tag{2}$$

Perhaps because our first draft did not explain the logical relationship between Eq. (1) and Eq. (2) well enough, the reviewer mistakenly equated these two equations (maybe out of misunderstanding, we guess) and made some derivations on this basis. The derivations from the reviewer are repeated as follows:

$$C\Delta t/\tau = \sqrt{(af\Delta t)^2 + b^2}; \tag{3}$$

$$(CI_u f \Delta t)^2 = (af\Delta t)^2 + b^2 \tag{4}$$

where the turbulence intensity is defined $I_u \equiv \sigma/U$. Thus one sees that

$$a = \sqrt{(CI_u)^2 - (b/f_{\text{dless}})^2} = CI_u \sqrt{1 - (\frac{b}{C\Delta t/\tau})^2}. \tag{5}$$

However, Eq. (1) is a special case of Eq. (2) when $b = 0$. In fact, $C\Delta t/\tau$ corresponds to $af\Delta t$. Eq. (1) only considers the decorrelation of turbulence dependent on time, but not the fact that the coherence is not necessarily unity for low frequency in reality, whereas Eq. (2) considers both effects.

Moreover, the reviewer got the conclusions that: 'In the limit of high dimensionless frequency or large turbulence intensity, i.e. without the offset $b$, then $a \to CI_u$ like Pielke & Panofsky (1970). But in the limit of small turbulence intensity or low dimensionless frequency (small $f\Delta t$), i.e. a large offset, then we see $a$ become imaginary, implying a (nonphysical) coherence oscillating with $\Delta t$ or $\Delta x$.' Regarding this, we think the reviewer might misunderstand the different roles of $f_{dless}$ and the model parameters $a$ and $b$: in the wind evolution model, $f_{dless}$ is the independent variable, $\gamma^2_{\text{model}}$ is the dependent variable, but the both model parameters are constants given a specific turbulent state. In other words, $a$ and $b$ are not functions of $f_{dless}$ but other variables (which are discussed in the paper).

In the major revision, we've completely restructured this part and also added sufficient detail. Hopefully, it won't cause misunderstandings anymore.

**2) Regarding our explanation about the $\Delta t$ dependence in $a$ due to**

**introducing $f_{dless}$:**
Theoretically, the independent variable of the wind evolution model can be frequency, wavenumber, and dimensionless frequency (Of course, the corresponding model parameters will be different). Therefore, we actually fitted these three different model forms to the estimated coherence, to determine which one would be the most suitable independent variable. During this process, we noticed the $\Delta t$ dependence of $a$ in both cases of using frequency and dimensionless frequency. As we explained in the last authors' response, if the $\Delta t$ dependence in $a$ did not exist inherently but was entirely introduced by $f_{\text{dless}}$, the exponent of $t_{\text{maxcorr}}$ should be around -1, and it could be canceled out with the $\Delta t$ in $f_{\text{dless}}$. But as shown in the paper, $a \propto \Delta t_{\text{maxcorr}}^{-0.49}$. Moreover, we did observe the $a_f$ (the decay parameter acquired from the fitting of the wind evolution model dependent on frequency) showing a proportional relationship with $\Delta t$. Hence, we believe that the $\Delta t$ dependence inherently exists in the decay parameter.

l.299-302: eqn.(18) uses Taylor's hypothesis to replace the spatial correlation with temporal correlation. But since this is potentially equivalent to no evolution (the spatial and temporal spectra differ, no?), then you have a conundrum here...using eqn.18 implies ignoring evolution. I think you need a different argument for picking only one of L or T; at any rate your argument in these lines doesn't quite hold. Why not include the text from your response ("from measured data, we have no alternatives"), and an argument as to how L and T are similar?
l.305: "L might contain uncertainties" ... —your eqn.18 is not correct, if there is evolution; see above comment.
Thank you for bringing up this interesting discussion! In our opinion, using Eq.(18) to approximate the integral length scale does not necessarily imply ignoring evolution. Please note that the main part of Eq. (18) is the integration of the autocorrelation function. The autocorrelation function is a quantity used to characterize the multi-time properties of a random process (Pope, 2000). (Autocorrelation can be regarded as equivalent to autospectrum.) Given the same turbulent properties, different realizations will follow the same autocorrelation function. And wind evolution is actually the decorrelation of turbulence, which can be understood as the process that one realization evolves to another realization preserving the same statistical properties (autocorrelation/autospectrum). In fact, the use of Taylor's translation hypothesis here means only the assumption of the "turbulent box" moving at the mean wind speed, but the "turbulent box" is not necessarily "frozen". The multiplication of the mean wind speed can be understood as scaling the x-axis of the integration, translating the time lag to spatial separation as follows:

$$L = \int_0^\infty \rho(U \cdot s)\mathrm{d}(U \cdot s), \tag{6}$$

where $s$ is time lag and $U \cdot s$ is used to approximate spatial separation.
We will formulate the corresponding text again to make it clearer.

Reference: Pope, S. B. (2000). Turbulent flows. Cambridge, New York: Cambridge University Press.

l.313-316: if the amount of sampling noise in mu3 or mu4 is very large, how is it "still worth investigating"? ML can't magically connect things if there is too much noise per trend/pattern; perhaps you can rephrase in terms of expected uncertainty, or at least in terms of (re-)confirming the expectation found in e.g. Lenschow et al.(1994). In the end the results appear to show that mu3 and mu4 are maybe carrying useful information; perhaps that could somehow be leveraged (or compare to the estimated uncertainty).

Thank you for your suggestion. We agree that the formulation here might be not good enough and have now rephrased it. As the reviewer mentions, ML can't magically connect things. This is exactly why we wanted to include skewness and kurtosis as predictors and test them with the ML models in this work — ML can not only make predictions, but it is also a useful tool for data mining. If skewness and kurtosis turned out to show no relevance to wind evolution prediction, it might not necessarily mean that they are really not related to wind evolution, but it might be that the relevance cannot be revealed due to the large uncertainties — we would not rule out this possibility from the beginning. Now, the results show that including skewness and kurtosis can improve the prediction of wind evolution, despite the uncertainties, which means they probably carry useful information related to wind evolution. This might indicate e.g. the further research on wind evolution could go into the direction of non-Gaussian fields and so on.

If we understand correctly, the reviewer suggests us to quantify the uncertainties of the sample skewness and kurtosis following the method suggested in Lenschow et al.(1994) and compare the estimated uncertainties with the expected one. We totally agree that this is a research question worthy of further study. However, this will be very complicated and need a lot of extra work. Considering the length of the current paper, it is not realistic to go into more detail in this aspect within this paper. We think it would make more sense to consider this research question in a new publication, so that we could have enough space to systematically discuss issues related to high-order wind statistics.

l.378-380: in your response you clarify that "$\beta$ can also be understood as the weight vector of h(x). But we've defined w as a weight vector before, we wanted to avoid using the same word in case reader might confuse these two different processes". Why not incorporate this in the updated draft version, to help the reader understand?

Thank you for your suggestion. This explanation has now been added to the corresponding text.

l.385-386: I understand that within the machine-learning context you are using the ML term 'characteristic length scale', but this is not a ML journal. I reiterate my statement about lines 289-296 in v3; why not help the reader (with my something like my previous comment in parenthesis, or a footnote)?

Thank you for your suggestion. We've added the suggested explanation about the 'characteristic length scale' to lines 396-397, after introducing the equation of the kernel function (Eq. (27)). We think it would be easier for readers to understand it when they first see the equation.

Some of the references are still lacking information, such as conference name, journal, or year. Further, some in-text citations are also lacking the year, and in sum cases initials are included in these citations; i.e., they do not conform to the referencing standard used here, and are not consistent throughout the paper.

Thank you for pointing out possible missing information in the references and possible errors in the in-text citations. We've carefully checked them again and corrected the errors.

[revised manuscript text omitted]